# On the tuning of plateaus in atmospheric and oceanic $^{14}$C records to derive calendar chronologies of deep-sea cores and records of $^{14}$C marine reservoir age changes

Edouard Bard[1] and Timothy J. Heaton[2]

1 CEREGE, Aix-Marseille University, CNRS, IRD, INRAE, Collège de France, Technopôle de l'Arbois, Aix-en-Provence, France

2 School of Mathematics and Statistics, University of Sheffield, Sheffield S3 7RH, UK

**Revised version (June 15, 2021)**

**Abstract:** We assess the methodology of the so-called $^{14}$C plateau tuning (PT) technique used to date marine sediment records and determine $^{14}$C marine reservoir ages (MRA) as recently reviewed by Sarnthein et al. (2020).

The main identified problems are linked to: the assumption of constant MRA during $^{14}$C-age plateaus; the lack of consideration of foraminifera abundance changes coupled to bioturbation that can create spurious plateaus in marine sediments; the assumption that plateaus have the same shapes and durations in atmospheric and oceanic records; the implication that atmospheric $^{14}$C/$^{12}$C peaked instantaneously from one plateau to the next; that the $^{14}$C plateaus represent 82% of the total time spent between 14,000 and 29,000 cal yr BP, whereas during the remaining 18% of the time, the radiocarbon clock was running almost 5 times faster than the radioactive decay; that the sparsity, combined with the level of analytical uncertainties and additional noise, in both atmospheric and marine data do not currently allow one to reliably or robustly identify plateaus (should they exist) beyond 15,000 cal yr BP; and that the determination and identification of plateaus in the deep-sea cores is reliant upon significant changes in sedimentation rate within those marine sediments which are, a priori, unknown and are not verified with an independent method.

The concerns we raise are supported and strengthened with carbon cycle box-model experiments and statistical simulations of pseudo-atmospheric and pseudo-marine records, allowing us to question the ability to identify and tune $^{14}$C-age plateaus, in the context of noisy and sparse data.

## 1/ Introduction

Sarnthein et al. (2020) review the results of a technique based on tuning hypothesized [14]C-age plateaus they inferred in deep-sea sediment cores with those that they have proposed exist in atmospheric radiocarbon archives, notably using the Lake Suigetsu record (Bronk Ramsey et al. 2012, 2020). The proposed outcomes of the so-called 'Plateau Tuning' (PT) are to establish accurate and precise calendar age scales of the marine sediments and, at the same

time, to determine the [14]C marine reservoir ages (MRAs) at the sea surface (for [14]C measured on planktonic foraminifera) and ventilation ages of deeper water masses (using [14]C measured on benthic foraminifera).

Sarnthein et al. (2020) reviews the results obtained by PT published over the last 13 years by the Kiel group (Sarnthein et al. 2007, 2011, 2013, 2015, Balmer et al. 2016, Sarnthein

& Werner 2017, Balmer & Sarnthein 2018, Küssner et al. 2018). By comparing the records from many locations, the authors conclude in section 2.2 that the [14]C-age plateaus beyond 15,000 cal yr BP *"show little coherence"* with independent [10]Be records based on polar ice cores and therefore that the cause of these [14]C anomalies may not be linked to cosmogenic production changes.

The authors thus propose that extremely large and variable ventilation ages may be the causes of these [14]C-age plateaus, constituting the fingerprint of abrupt reversals of deep ocean circulation and abrupt release or drawdown of $CO_2$ into or from the atmosphere. Nevertheless, the authors admit that *"ocean models still poorly reproduce"* their reconstruction of deep ocean circulation and carbon cycle changes (end of abstract). They claim that this mismatch is due to

model deficiencies in spatial resolution and tuning with reference data.

We have strong reservations about the appropriateness of the PT technique and consequently also the reliability of the results obtained in Sarnthein et al. (2020). The PT technique, proposed and used by the same authors from Kiel for more than 13 years, has not been checked and replicated independently by other groups. Outside the Kiel group, only

Umling & Thunell (2017) have used the PT technique, but found rather puzzling results (see more in section 2.3). PT has been presented on several occasions during International Radiocarbon Conferences and workshops of the IntCal group, but has never been adopted as a viable technique to reconstruct past [14]C variations (Reimer et al. 2020, Heaton et al. 2020a and previous IntCal iterations by Reimer et al. 2009, 2013).

The review paper by Sarnthein et al. (2020) only compiles previous papers by the same group. The risk is to mislead readers into thinking that the PT technique is now firmly

established. Indeed, the compiled records based on PT leads to perplexing outcomes (no coherence with either production changes or with ocean modeling results). This failure is linked to the inherent pitfalls listed below, which are not treated adequately in Sarnthein et al. (2020), nor in former papers by the same group.

With this extended comment, our objective is to expose and discuss openly some of the inherent problems linked to PT. We split our discussion into two sections. Firstly, we present our concerns from a geoscientific perspective. Secondly, we provide our statistical concerns with the proposed PT method and provide illustrative examples that highlight its intrinsic difficulties.

### 2/ Paleoclimatic & paleoceanographic perspective

**2.1/** The PT principle is reminiscent of the '[14]C wiggle matching technique' used to refine the dating of large pieces of wood with multiple [14]C analyses over a tree-ring sequence of at least a few decades (Bronk Ramsey et al. 2001). However, Sarnthein et al. (2020) propose to do this with ocean sediments, that are not annually laminated, and to obtain calendar chronologies accurate and precise at the "decadal-to-centennial" level mentioned in their section 1.1. However, there is no independent constraint on the sedimentation rate variations in these ocean cores (without annual varves). Indeed, sedimentation rate changes linked to climatic-oceanographic events (e.g. Dansgaard-Oeschger and Heinrich events) or more local sedimentological causes could also create [14]C-age plateaus.

PT is not always restricted to the tuning of a single plateau but often used to tune a suite of [14]C age-plateaus. However, one cannot reliably PT if one cannot reliably identify and define an individual [14]C plateau. A suite of plateaus does not necessarily add strength – in fact it potentially makes it more challenging should one miss, or simply mismatch, plateaus in either the atmospheric target or the sediment record. Moreover, the possible existence of spurious plateaus further complicates PT (see more below).

**2.2/** In addition to determining the calendar chronology of ocean sediments, PT is also used to calculate, at the same time, a variable offset with the atmospheric [14]C curve. The offset for planktonic foraminifera is often very large (1000 – 2000 [14]C yrs), and is interpreted as being due to [14]C reservoir age changes at the sea surface. However, in order to perform PT the authors are required to assume that the marine [14]C reservoir age (MRA) is strictly constant during the age plateaus, which represent 82% of the total time spent between 14,000 and 29,000 cal yr BP.

This assumption of reservoir age stability during $^{14}$C-age plateaus is antithetical with the conclusion that these plateaus are linked to carbon cycle changes. Such significant carbon cycle changes would have left their imprint in $^{14}$C records (Bard 1988), maybe even as $^{14}$C-age plateaus solely recorded in marine sediments. Hence, the $^{14}$C structures identified by Sarnthein et al. (2020) in pelagic sediments are severely under-constrained in $^{14}$C and calendar ages.

Indeed, changes in marine $^{14}$C reservoir age may either mask (or create false) $^{14}$C-age plateaus in the marine core causing issues with tuning to the atmospheric plateaus. On the one hand, a decrease in MRA coinciding with an atmospheric $^{14}$C-age plateau (i.e., a decrease in atmospheric $\Delta^{14}$C) may create a set of marine $^{14}$C observations lacking any plateau. On the other, an increase in MRA may make a period where the atmospheric $^{14}$C record does not plateau (e.g., constant atmospheric $\Delta^{14}$C) appear as a plateau in marine $^{14}$C observations. In both these instances, PT will fail. Unless MRAs only change at the boundary times of their chosen plateau, identifying whether a potential plateau in a set of marine observations should be tuned to an atmospheric plateau is potentially confounded by the very changes to MRA the authors report.

**2.3/** Sarnthein et al. (2020) refers to their hypothesized Lake Suigetsu-based $^{14}$C calibration curve as a 'rung ladder' that provides the basis of PT. Actually, the series of $^{14}$C-age plateaus hypothesized by Sarnthein et al. (2020) resembles a 'staircase' more than a 'rung ladder'. In Figure 1, we have created a Lake Suigetsu-only calibration curve using the same Bayesian statistical method used for IntCal20 (Heaton et al., 2020b, Reimer et al. 2020) but constructed based only upon the observations from Lake Suigetsu with its updated calendar age timescale (Bronk Ramsey et al. 2020). Fig. 1 shows the Lake Suigetsu $^{14}$C data and the resulting Suigetsu-only radiocarbon calibration curve for the period beyond the last 14,000 cal yr BP (14 cal kyr BP), i.e., where continuous high-precision data on tree-rings are not currently available. Superimposed horizontal lines indicate the 15 hypothesized atmospheric $^{14}$C-age plateaus of Sarnthein et al. (2020), with their numbering as listed in their Table 1. Fig. 2 is equivalent to Fig. 1, but compares the Sarnthein $^{14}$C-age plateaus directly with the IntCal20 calibration curve (Reimer et al. 2020). In addition to Lake Suigetsu, the IntCal20 curve uses (atmosphere-adjusted) $^{14}$C determinations from speleothems, lacustrine and marine sediments, and corals; as well as some $^{14}$C determinations obtained from floating tree-ring sequences. Besides the well-known plateau #1 corresponding to the beginning of the Bölling period, evidence for many of the older plateaus hypothesized by Sarnthein at al. (2020) is dubious. They are not replicated in either our statistically-robust Lake Suigetsu-only curve (Fig. 1) or the IntCal20 curve (Fig.

2). The weak evidence for many of these hypothesized $^{14}$C-age plateaus is further detailed in section 3.2.

By focusing only on the plateaus, Sarnthein et al. (2020) overlook the implication that, in their model, $^{14}$C ages must jump, often instantaneously, from one plateau to the next (i.e., like in a staircase as shown in Figs. 1 and 2). This is particularly true for five steps between 10

plateaus (10b to 10a, 9 to 8, 6b to 6a, 5b to 5a and 2b to 2a) for which the calendar gaps correspond to **zero** cal yr, but the atmospheric $^{14}$C ages drops by between 340 and 750 $^{14}$C yrs. Five other steps (between 10 plateaus: 11 to 10b, 8 to 7, 7 to 6b, 6a to 5b and 5a to 4) also have minimal calendar durations (70 to 170 cal yr) but show large $^{14}$C drops ranging from 660 to 1380 $^{14}$C yr. Consequently, the total duration of $^{14}$C plateaus represent 82% of the time spent

between 14 and 29 cal kyr BP, whereas during the remaining 18% of the time, the radiocarbon clock (i.e., the pace at which the $^{14}$C age changes compared with true calendar time) was running almost 5 times faster than radioactive decay. The implication of the hypothetical staircase shape of Sarnthein et al.'s proposed atmospheric calibration curve, is that radiocarbon would have never behaved as a geochronometer driven by regular radioactive decay.

It is useful to convert Figs. 1 and 2 from $^{14}$C-age into $\Delta^{14}$C in order to assess the implications of these vertical steps. This is done in Fig. 3 which shows that $^{14}$C-age plateaus are transformed into triangular $\Delta^{14}$C wiggles. The consequence of abrupt $^{14}$C-age drops between $^{14}$C-age plateaus is that most of these $\Delta^{14}$C wiggles exhibit instantaneous rises ranging in size from 50 to 250 ‰.

There is no known mechanism that could be responsible for such abrupt and large asymmetric wiggles of the atmospheric $\Delta^{14}$C. Instantaneous $^{14}$C production increases, that result in about 4 times the average production in a year, were discovered recently (Miyake et al. 2012, Mekhaldi et al. 2015). However, the size of these spikes attributed to extreme solar particle events is an order of magnitude smaller in terms of $\Delta^{14}$C than that required to explain

the jumps between the hypothesized $^{14}$C-age plateaus of Sarnthein et al. (2020). Furthermore, there is no evidence of huge corresponding spikes in the ice core $^{10}$Be record (Adolphi et al. 2018).  In addition, the impacts of abrupt changes of the geomagnetic field were found to be negligible on the production of $^{14}$C (Fournier et al. 2015). Finally, it is unlikely that abrupt changes of the carbon cycle are responsible for such large, frequent, and very abrupt $\Delta^{14}$C

spikes. For example, switching down the deep ocean circulation instantaneously in a carbon cycle box-model, leads to a rather slow and limited $\Delta^{14}$C rise in the atmosphere over several centuries (e.g., see Fig. 4b by Goslar et al. 1995 or Fig. 5 by Hughen et al. 1998; see also simulations performed with more complex models by Marchal et al. 2001, Delaygue et al. 2003,

Ritz et al. 2008, Singarayer et al. 2008). Consequently, the assumption which is required to underpin PT, that the radiocarbon calibration curve has the shape of a staircase, is in conflict with our basic understanding of $^{14}$C as a tracer.

**2.4/** Unfortunately, the Sarnthein et al. (2020) review paper fails to show a single figure illustrating oceanic $^{14}$C records with their plateaus compared, and tuned, with the proposed atmospheric calibration $^{14}$C curve. It is thus necessary to dig into the literature to see the marine core records: Figs. 3, 4 in Sarnthein et al. (2007) for cores SO17940, MD01-2416, ODP893A and PS2644, Fig. 2 in Sarnthein et al. (2011) for core MD01-2378, Figs. 3 to 13 in Sarnthein et al. (2015) for cores GIK23074, PS2644, MD08-3180, ODP1002, MD01-2378, GIK17940, SO50-37, MD01-2416, MD02-2489, ODP893A, MD02-2503, Suppl. Figs. S1a,b,c,d in Balmer et al. (2016) for cores GeoB1711-4, GeoB3910-1, KNR-159-5-36GGC, MD07-3076), Fig. 4 in Sarnthein & Werner (2017) for cores GIK23258, MSM5/5-712, and T88-2, Fig. 2 in Balmer & Sarnthein (2018) for core MD08-3180, Fig. 4 in Küssner et al. (2018) for core PS75-104-1.

Looking at these graphs is absolutely crucial to assess the poor quality of the determination of $^{14}$C-age plateaus in these ocean cores and their tuning to the atmospheric $^{14}$C calibration curve. This is particularly important as the PT implies sedimentation rates that vary by up to a factor of 5 to 8 within a single core (e.g. cores PS2644 and MD08-3180 in Sarnthein et al. 2015; PS75/104-1 in Küssner et al. 2018) and even much more, by orders of magnitude, in other cores from the Nordic Seas (Sarnthein & Werner 2017).

The proposed PT criterion which considers only the average sedimentation rate over a complete core (e.g. > 10 cm/kyr) is not sufficient to ensure reliability in the PT approach. Rather it is the profile and range of sedimentation rate within a sediment core which is most critical as these internal relative changes determine the identification of the plateaus in the absence of calendar age information. This also applies to the impact of bioturbation since the smoothing and phasing effects are directly related to the ratio between the bioturbation depth and the sedimentation rate (this ratio being the average residence time of foraminifera in the bioturbation zone, e.g. Bard et al. 1987).

In Appendix B of their Supplemental Information, Sarnthein et al. (2020) provide summary Figures for 18 individual deep-sea cores, showing the final reconstructions of surface and deep reservoir ages versus time for each core. However, these graphs are not particularly useful to assess PT tuning because they do not show the raw $^{14}$C data versus depth compared to the Suigetsu $^{14}$C record.

In fact, the changes of sedimentation rates implied by PT are even larger than mentioned

above because remaining conflicts between atmospheric and marine $^{14}$C records are resolved by introducing ad-hoc discontinuities in the core stratigraphies. These periods, forced to have a sedimentation rate dropping down to zero, are assumed to be previously unnoticed sedimentological hiatuses (e.g., 9 hiatuses are inferred by PT in the 18 sediment cores presented in Fig. S2). Sarnthein et al. (2020) further claim that the *«plateau-based high-resolution chronology has led to the detection of numerous millennial-scale hiatuses overlooked by conventional methods of stratigraphic correlation. In turn, the hiatuses give intriguing new insights into past changes of bottom current dynamics linked to different millennial-scale geometries of overturning circulation and climate change»*. No independent sedimentological evidence is presented to verify these previously unnoticed hiatuses, which occur surprisingly at the $^{14}$C plateau boundaries for no obvious reason. These discontinuities may just be artefacts of the PT method. In their CC accompanying this paper, Sarnthein & Grootes further claim that there should be a positive correlation between hiatus frequency and sedimentation rate. This counterintuitive hypothesis remains speculative in the absence of evidence independent of PT.

Surprisingly, the summary Fig. S2c for three cores (MD07/3088, SO213-76 and PS97/137-1) has been changed between the submitted version (available in *Climate of the Past Discussions*) and the final published paper by Sarnthein et al. (2020). The changes are particularly important for the last two cores: for example, the LGM surface reservoir age around 20 cal kyr BP has now doubled for core PS97/137-1 (more than 2000 $^{14}$C yr in the final publication instead of 1000 $^{14}$C yr in the initial submitted version). Benthic reservoir ages have also changed by more than 1000 $^{14}$C yr (up or down) in the last two cores. In addition, both records now exhibit two hiatuses in the final publication as opposed to the single hiatus presented in the initial submission. In the submitted and published versions of Sarnthein et al. (2020), these new results were referred to a submitted paper by Küssner et al., which is still unpublished. Hence, no explanation is available to the reader to assess the reason for the drastic change of MRA reconstructions between the two versions.

In their CC, Lamy & Arz who studied cores from the same area (e.g., Lamy et al. 2015), confirm the serious problem linked to the application of PT to core PS97/137-1. They raise doubts about the highly variable sedimentation rates within this core, which make the identification of any $^{14}$C plateaus highly uncertain. They further note that the status of the laminations in these sediments is still debated, which implies that a rough count cannot be used to support the chronology based on PT. In their CC, Michel & Siani also express caution about PT results presented by Sarnthein et al. on a South-East Pacific core (MD07-3088) studied previously by Siani et al. (2013). Their concern is about the reconstructed variability of

sedimentation rate based on PT (up to a factor of 25 for that core) and Michel & Siani also underline that the MRA cannot be precisely defined for the glacial part of the core. In addition to South Pacific cores, Lamy & Arz extend their doubts to PT tuning applied to other cores studied by Sarnthein et al. (e.g., GeoB3910 off Brazil in a zone studied previously by Arz et al. 1999).

Outside the 18 records obtained by the Kiel group and compiled by Sarnthein et al. (2020), there is one further paper by other authors who have used PT for stratigraphical purposes and for reconstructing $^{14}$C reservoir ages. Umling & Thunell (2017) used eyeball PT to derive their chronology for a sediment core located at 2.7 km depth in the Eastern Equatorial Pacific. Their tuning of the $^{14}$C record of core TRI163-23 onto the Suigetsu $^{14}$C record implies the unexpected presence of hiatuses in this core at boundaries between $^{14}$C plateaus (i.e., gaps between plateaus #1a & YD, between plateaus 1 & 1a, and between plateaus #2a & 1, see Fig. 3c of Umling & Thunell, 2017). The first hiatus is particularly long (1200 cal yr), but no independent data is presented to confirm the presence of such a large unconformity in this core. In addition, the deglacial $^{14}$C reservoir ages reconstructed for core TRI163-23 exhibit discrepancies with the record obtained by de la Fuente et al. (2015) on another core from the Eastern Equatorial Pacific collected at a similar depth (2.9 km). In their CC, Sarnthein & Grootes even disagree on specific aspects of the PT performed by Umling & Thunell (2017) highlighting the subjectivity of the PT approach.

**2.5/** The PT technique is focused on the detection and use of $^{14}$C-age plateaus. This implies that a large part of the $^{14}$C record is left unused in the matching process. This is actually surprising because plateaus are used in the frame of PT to define the absolute chronology of the marine sediment core, while plateaus in the $^{14}$C calibration curve are generally viewed as poor periods for obtaining precise calibrated ages, in contrast to 'high-slope' parts of the calibration curve (e.g., Svetlik et al. 2019). The only potential justification we can identify for such an approach would be if one believed that changes to MRA could only occur at plateau boundaries (and remained otherwise constant). However, there is no special significance in the plateaus of the $^{14}$C calibration curve. After transformation to atmospheric $\Delta^{14}$C (Fig. 3), a $^{14}$C-age plateau is only the second part of a wiggle, during which the $\Delta^{14}$C decrease compensates the radioactive decay. If anything, the entire $\Delta^{14}$C wiggle is the feature that should be matched (equivalent to a 'high-slope' – 'low-slope' sequence in the plot of $^{14}$C age vs calendar age).

Without a clear justification for why plateau boundaries would coincide with all MRA changes, there is thus no reason to match the $^{14}$C-age plateaus only and discard the other parts

of the $^{14}$C sequence. Matching the entire $^{14}$C record with the target curve would provide a stronger analogy with the wiggle matching technique used for tree sequences (but without resolving the remaining pitfalls of having no independent calendar dating for most ocean sediments; and that no matching could be done unless one already knew the MRA). Matching the entire $^{14}$C sequence is also the method used to synchronize floating tree-ring sequences to a master chronology (e.g., Capano et al. 2020 used for the IntCal20 curve by Reimer et al. 2020).

Indeed, even if MRA changes were to occur only at plateau boundaries, the marine and atmospheric $^{14}$C-age records should show entirely the same shape, just with piecewise constant offsets during and between each plateau. This would provide an even stronger argument that the whole $^{14}$C record should be matched. Plateau tuning would then reduce to finding the change points in the piecewise constant MRA offset between atmospheric and marine $^{14}$C-ages.

**2.6/** In contrast to the atmospheric $^{14}$C calibration curve, there is indeed a special significance in a $^{14}$C-age plateau observed in ocean sediments. As mentioned above, within a sediment core, a $^{14}$C-age plateau could be a simple consequence of an abrupt sedimentation rate increase or even a slump that mixes sediments of the same age. Another potential source of $^{14}$C-age plateaus has been completely overlooked by Sarnthein et al. (2020): the coupling of continuous bioturbation with changes in the abundance of the $^{14}$C signal carrier. Indeed, the assemblages of foraminifera used for $^{14}$C analyses often varied in the past due to global or regional paleoceanographic conditions (these large and systematic faunal changes are the basis for the use of planktonic foraminifera in paleothermometry, e.g., Imbrie & Kipp 1971, Kucera et al. 2005).

In particular, an abrupt decrease or an abrupt increase of the foraminifera abundance will inevitably create a $^{14}$C-age plateau, as theorized by Broecker et al. (1984). The first demonstration was provided by Bard et al. (1987) who showed that $^{14}$C-age plateaus and $\partial^{18}$O phase lags measured on two planktonic species in a deep-sea core from the North Atlantic could be explained quantitatively by bioturbation modeling forced with the abundance records of both species. Since 1987 many other groups have made similar observations of $^{14}$C-age plateaus and discrepancies explained by bioturbation coupled with foraminifera abundance changes (e.g., Costa et al. 2017, Ausin et al. 2019).

In order to interpret $^{14}$C-age plateaus in ocean sediments, it is thus indisputably necessary to show the absolute abundance records of the different foraminifera used for $^{14}$C dating (counts expressed in number per gram of sediment). This has never been the case for any

of the PT papers by the Kiel group used for this new compilation (Sarnthein et al. 2007, 2011, 2013, 2015, Balmer et al. 2016, Sarnthein & Werner 2017, Balmer & Sarnthein 2018, Küssner et al. 2018, Sarnthein et al. 2020). It is possible that several marine $^{14}$C-age plateaus could be mere sedimentary artefacts.

    In their CC, Sarnthein & Grootes cite another paper by Ausin, Sarnthein and Haghipour

(2021) published after the submission of our paper. This recent work is based on a sediment core from the Iberian margin for which Sarnthein & Grootes provide a graph representing the foraminifera abundance counts, showing no obvious correlation between $^{14}$C plateaus and drops in abundances. This information is indeed useful and it is unfortunate that these data are not shown, nor provided in the paper by Ausin et al. (2021).

Nevertheless, caution should be taken with the work by Ausin et al. (2021) who obtain low MRA and benthic $^{14}$C values in this Iberian Margin core, notably during Heinrich stadial 1 (HS1) and the last Glacial Maximum (LGM), in stark contrast with records obtained on nearby cores (Skinner et al. 2014, 2021). The MRA drop (down to 300 yr) during HS1 based on PT is also in conflict with modeling results (Delaygue et al. 2003, Ritz et al. 2008, Franke et al. 2008,

Butzin et al. 2017). Although Sarnthein & Grootes present the new paper by Ausin et al. (2021) as a "nice test case" of PT, the strong disagreement with the literature is not reassuring. Furthermore, it also remains to be seen if the bioturbation-abundance couple could not be an adequate explanation for some $^{14}$C-age plateaus in the 20 other published records based on PT and for which the foraminifera counts are not available.

    **2.7/** Let us now assume that $^{14}$C-age plateaus in marine records do match those identified in the atmospheric record (i.e., the basic assumption of the PT technique). What remains problematic, if we are intending to use PT to create a precise calendar age time scale for the marine record, is that marine $^{14}$C-age plateaus are assumed to be as sharp and as long as their

corresponding atmospheric plateaus, even in the case of very large surface reservoir ages reconstructed by the method. The sections below show that this conflicts with our understanding of the carbon cycle.

    For the past 14 kyr, high resolution $^{14}$C data on tree-ring and $^{10}$Be on polar ice cores have shown that most centennial $\Delta^{14}$C wiggles in the atmosphere are due to cosmogenic

production changes (Beer et al. 1988, Adolphi et al. 2016, 2017, 2018) mainly linked to the solar variability as illustrated by studies covering the last millennium (e.g., Bard et al. 1997, Muscheler et al. 2000, Delaygue & Bard 2011). This $^{14}$C production signal is transferred from the atmosphere to the ocean surface before being slowly transported to the deep ocean. The

atmospheric $^{14}$C wiggle inevitably gets damped in the other reservoirs of the carbon cycle,
notably the surface ocean. In addition, the oceanic wiggle is not strictly in phase with the
atmospheric one. Both damping and phasing effects conflict with the main assumption of the
PT technique, namely that marine $^{14}$C-age plateaus can be matched directly to atmospheric
ones.

Using numerical models, it is possible to quantify the damping and phasing effects,
which depend directly on the duration of the $\Delta^{14}$C wiggle and on the carbon residence time in
the considered carbon cycle reservoir (or chain of reservoirs). A convenient way is to consider
a sinusoidal $^{14}$C production leading to attenuated and shifted $^{14}$C signals in the atmosphere and
the ocean surface (see Figure 4). The Indo-Pacific low-latitude surface box of the 12-box model
by Bard et al. (1997) has a $^{14}$C reservoir age of 320 $^{14}$C yrs at steady state. The relative
attenuation (i.e., the amplitude in the troposphere divided by the amplitude in the surface ocean)
is a factor of 1.8 for 500-cal-yr long $^{14}$C wiggles and a factor of 3.1 for 200-cal-yr long wiggles.
The phase lag between atmospheric and oceanic response also varies with the duration of the
$^{14}$C production signal: about 60 and 45 cal yr for 500 and 200-cal-yr long wiggles, respectively.

To illustrate the relative attenuation increase with a larger reservoir age it is useful to
consider the simulation for the Southern Ocean surface box which has a reservoir age of 890
$^{14}$C yrs in the 12-box model. For the same 500-cal-yr and 200-cal-yr long $\Delta^{14}$C wiggles, the
relative attenuation factors are 3.0 and 3.4, respectively. Fig. 4a shows the results of these
calculations for signal periods ranging from 2000 to 100 calendar yrs. The period is equivalent
to the duration of the $\Delta^{14}$C wiggle and to twice the duration of the $^{14}$C-age plateau, which
corresponds to the descending part of the $\Delta^{14}$C wiggle.

As illustrated in Fig. 4b, a 500-cal-yr long wiggle of cosmogenic production varying by
$\pm 15\%$ around its mean value is sufficient to produce an atmospheric $^{14}$C-age plateau. However,
in the surface ocean boxes, the slope of the $^{14}$C-age vs cal-age curves does not drop to zero, i.e.,
there is an absence of a $^{14}$C-age plateau in the modelled surface ocean environment despite the
atmospheric plateau. This is a direct consequence of the damping of $^{14}$C signals by the carbon
cycle.

The $^{14}$C bomb spike of the early 1960s provides further evidence of the smoothing and
phasing effects. The main $^{14}$C injection lasted only a few years and the bomb spike can be
viewed as the impulse response function of the carbon cycle. In detail, the bomb pulse is
complicated because the $\Delta^{14}$C decrease observed in the atmosphere over the last 50 years is also
partially due to the Suess effect linked to the input of anthropogenic $CO_2$ devoid of $^{14}$C (Levin
& Hesshaimer 2016). In any case, the ocean surface $\Delta^{14}$C increased in the 1970s by $150 - 200$

‰ above pre-bomb values, which is about 5 times less than the maximum anomaly in the atmospheric pool. This large damping effect remains true even for the shallowest lagoons (Grottoli & Eakin 2007), which illustrates the efficacy of ocean mixing to counterbalance air-sea gas exchange.

Today the ocean surface $\Delta^{14}C$ is still above these pre-bomb values (e.g., Andrews et al. 2016) and this anomaly will remain with us for many decades to come as the present level in the ocean surface is only about halfway through its long-term asymptotic decrease. The calculation of an average phase lag between atmosphere and ocean is difficult because the impulse response functions are completely asymmetric (i.e., the delay for the signal rise is totally different to that observed for the signal decrease). To sum up, the bomb spike in the surface ocean is also a century scale event, with an attenuation compatible with that calculated by considering sinusoidal signals (Fig. 4a), which remains the traditional way used in signal analysis (i.e., so-called Bode plots).

**2.8/** The considerations above apply to $\Delta^{14}C$ wiggles linked to $^{14}C$ production changes, but it has likewise been suggested that $^{14}C$-age plateaus may also correspond to carbon cycle changes, notably at the end of the Younger Dryas climatic event (Oeschger et al. 1980, Goslar et al. 1995, Hughen et al. 1998). This YD $^{14}C$-age plateau may have been caused by an abrupt resumption of the meridional overturning circulation as simulated by numerical models (Goslar et al. 1995, Stocker & Wright 1996,1998, Hughen et al. 1998; see Fig. 17 by Bard 1998 comparing the $^{14}C$-age plateau simulated by the 12-box model and the Bern 2.5D physical model).

In such a case, the $^{14}C$ perturbation originates from the ocean and the relative attenuation and phase relationships with the atmospheric pool are more complex, exhibiting regional differences. The surface ocean regions responsible for uptake or outgassing of $CO_2$ exhibit large effects with no delay, whereas other regions are only affected in a passive way through the atmosphere. For those widespread passive regions, we must refer back to the calculations shown in Figs. 4a, and 4b, which lead to large attenuations in the ocean.

The study of atmosphere and surface ocean $^{14}C$ wiggles linked to spatially variable ocean-atmosphere exchange is inherently more complex as it requires spatial resolution with 2D-3D models and consideration of regional $^{14}C$ data to identify active and passive regions. However, regional gradient changes of surface $^{14}C$ simulated by models are generally on the order of a few centuries (Stocker & Wright 1998, Delaygue et al. 2003, Butzin et al. 2005, Franke et al. 2008, Singarayer et al. 2008, Ritz et al. 2008), rather than the millennia advocated

by Sarnthein et al. (2020).

**2.9/** In the introductory section 1.2 of their paper, Sarnthein et al. (2020) mention a completely different method to reconstruct MRA based on $^{14}$C datings of the same volcanic ash layer (tephra) in land and marine sediments. This is indeed a precise and accurate method used in many studies including those cited by Sarnthein et al. (2020). However, they should have also cited the two seminal papers on the subject: Bard (1988) was the first to propose specifically the use of volcanic ash layers to reconstruct past MRA variations, and Bard et al. (1994) the first to reconstruct past MRA changes with this powerful method.

The further advantage of the MRA method based on tephra is that the downcore profile of volcanic shard counts provides a useful constraint on the bioturbation depth and intensity, which also affect $^{14}$C ages measured on foraminifera. Indeed, this shard profile is the impulse response function of the bioturbation filter and hence provides information that can be used to correct $^{14}$C ages used for the MRA calculation (Bard et al. 1994). Another route to detect and constrain bioturbation is to study the total scatter of $^{14}$C ages measured on single specimens of foraminifera and to compare it with the smaller dispersion linked to analytical errors only (e.g., Fagault al. 2019).

### 3/ Statistical perspective

**3.1/** Fundamental to the accuracy, robustness, and reliability in the estimates of the timescales and MRAs obtained by PT, is whether true atmospheric $^{14}$C-age plateaus can be consistently identified, and also matched, between sparsely sampled and noisy records. These plateaus must be reliably identified both in the atmospheric *master target* record, Lake Suigetsu in the case of Sarnthein et al. (2020); and in the marine sediment cores for which one wishes to infer a calendar chronology. In the case of marine sediment cores, identification of these atmospheric $^{14}$C-age plateaus is further confounded by potential MRA changes, bioturbation, and the lack of an independent timescale. One needs to have confidence that the plateaus identified in both the marine and atmospheric $^{14}$C records are not only genuine atmospheric features but have also been correctly paired together.

Sarnthein et al. (2020) do not appear to address this issue – instead concentrating on an argument as to whether the true underlying atmospheric $^{14}$C record contains plateaus. The presence of atmospheric plateaus is a necessary condition, and is discussed further in Section 3.8, but certainly not sufficient for the reliability of PT. Rather, the main statistical concern for

PT is as to whether any such genuine atmospheric [14]C-age plateaus can be identified in both the sparse, and uncertain, Lake Suigetsu [14]C record **and** the marine sediment [14]C record being plateau-tuned. If one either mislocates a true [14]C-age plateau in one or other of the records, or incorrectly pairs plateaus between the records, then the subsequent timescales of the sediment 450    core and MRA estimation will be unreliable. We raise specific statistical concerns regarding the reliable identification and pairing of potential plateaus.

**3.2/** The first statistical requirement for the PT technique is the ability to precisely identify hypothesized [14]C-age plateaus based upon limited sequences of noisy and sparsely 455    sampled [14]C observations. The limitations in the amount of data available in the underlying records used for PT, and their observational noise, lead to rather ambiguous estimations of the locations, and durations, of the proposed Sarnthein et al. [14]C-age plateaus. The challenge of plateau identification will be most significant in the ocean sediment cores to be tuned, since they are typically the most sparsely sampled and lack a timescale, as we discuss later. However, 460    such ambiguity is also present in the atmospheric records. This raises questions regarding the reliability of the atmospheric [14]C-age plateaus used as a target.

In the case of Sarnthein et al. (2020), the hypothesized atmospheric [14]C-age plateaus are selected based on the Lake Suigetsu and Hulu [14]C records (see Sarnthein et al.'s Fig. S1, showing [14]C error bars at 1σ). Sarnthein et al. (2020) invoke ad-hoc changes of the dead carbon 465    fraction (DCF) of the Hulu Cave speleothems (section 2.2), and an argument that filtering has removed the plateaus within the speleothem [14]C record, to explain the lack of correspondence between the [14]C-age plateaus they outline in the Lake Suigetsu target curves compared with the Hulu Cave target (section 1.2). There is no obvious reason that would explain systematic changes in the speleothem DCF occurring only **during** the atmospheric [14]C-age plateaus in 470    such a way as to both mask these atmospheric plateaus in the speleothem [14]C record, and to ensure that the speleothem record does not generate additional spurious, DCF-driven, [14]C-age plateaus between the genuine atmospheric ones. As already underlined in our section 2.2, the opposite hypothesis is made by Sarnthein et al. (2020) for MRA which is assumed to remain constant during the atmospheric [14]C-age plateaus, changing only at boundary times between 475    plateaus.

Based on the comparison with the precise IntCal13 record over the past 14 cal kyr BP, the DCF for Hulu is in the order of $450 \pm 70$ [14]C yr (1σ, Cheng at al., 2018). This mean value and standard deviation have been further tested and refined in the frame of IntCal20 by comparing the Hulu data for individual speleothems with the tree-ring [14]C record over the past

14 cal kyr BP. This IntCal20 testing gives an estimate of $480 \pm 55$ [14]C yr (Reimer et al. 2020, Heaton et al. 2020b). The low value and stability of the Hulu DCF are attributed to the characteristics of the Hulu cave with its sandstone ceiling and open system conditions with the soil above it. In a similar way to the ocean modeling in section 2.7, the carbon transport and mixing processes leading to the DCF should have somewhat smoothed the atmospheric [14]C

variations. Although a specific modeling should be performed for a particular cave system, the DCF value for Hulu cave (ca. 480 [14]C yr) is equivalent to the average MRA of low-to-mid-latitude surface oceans (Heaton et al. 2020a). Consequently, the [14]C-age plateaus should be smoothed and delayed at a similar level as in surface ocean records, for which Sarnthein et al. (2020) assume that plateaus are of the same duration and timing as in the atmospheric record.

Further, one would expect that due to the time-directional nature of any speleothem filtering (i.e., that it averages over past atmospheric [14]C concentrations) [14]C-age plateaus in the Hulu record should either be seen with a time lag compared with the atmosphere, or at least towards the more recent end of the atmospheric plateau. This is not the case in Sarnthein et al. (2020) where Figure S1 predominantly proposes Hulu Cave [14]C-age plateaus that occur at the

beginning (i.e., older end) of their hypothesized atmospheric [14]C-age plateaus.

          When one considers the uncertainty in the [14]C determinations which are used to identify the hypothesized atmospheric [14]C-age plateaus (see Fig. S1 of Sarnthein et al. (2020) showing error bars plotted at $1\sigma$), it is unclear how strong the evidence for several of the plateaus in the atmospheric target suite is. When measurement uncertainties are large, one would expect

(simply due to the randomness of these uncertainties) to observe sequences of [14]C determinations that are non-monotonic even when the underlying atmospheric [14]C-age to calendar age is monotonically increasing. Based upon the Lake Suigetsu observations and uncertainties, it is therefore hard to assess whether some of the hypothesized atmospheric [14]C-age plateaus really exist or are rather just random artefacts due to measurement uncertainty.

This is particularly true for the upper panel of their Fig. S1 focused on the period between 21 and 27 cal kyr BP.

          To illustrate this concern about the reliability of the hypothesized atmospheric target plateaus more clearly, as mentioned in section 2.3, we created a Lake Suigetsu-only calibration curve. This used the same Bayesian statistical method as implemented for IntCal20 (Heaton et

al., 2020b, Reimer et al. 2020) but was constructed based only upon the [14]C observations from Lake Suigetsu (using the updated Lake Suigetsu calendar age timescale provided by Bronk Ramsey et al. 2020). Fig. 1 shows the Lake Suigetsu [14]C data with their $1\sigma$ analytical uncertainties (both in radiocarbon and calendar age) and the resulting Suigetsu-only

radiocarbon calibration curve with its 95% posterior predictive probability interval. Superimposed horizontal lines indicate the 15 hypothesized atmospheric plateaus with their numbering as listed in Table 1 of Sarnthein et al. (2020). Besides the well-known plateau #1 corresponding to the beginning of the Bölling period, it is dubious as to whether many of the older plateaus in particular are supported by the Lake Suigetsu data based on our statistical assessment.

Furthermore, most of the Sarnthein et al. (2020) hypothesized $^{14}$C-age plateaus have calendar durations exceeding the 95% posterior predictive probability interval around the Lake Suigetsu data (notably plateaus # 2a, 4, 8 and 10b). The two plateaus (# 2b and 6b) with the shortest duration (410 cal yrs) are compatible with the probability interval around the Lake Suigetsu data. However, these sections of the Suigetsu-only calibration curve are also compatible with straight oblique lines with no plateau at all. Such a conclusion is supported by Fig. 2 comparing the plateaus and the IntCal20 curve. Only a few plateaus could correspond to particular structures of the IntCal20 curve, notably plateau # 1 which is already known, and maybe plateaus # 7, 10a and 11, although the identified structures are much shorter, and thus smaller in $\Delta^{14}$C (Fig. 3), than the hypothesized plateaus would imply.

**3.3/** Current $^{14}$C sediment-based records do not have the resolution or precision in $^{14}$C measurement one might ideally desire – it is for this reason the community aims for the use of tree-rings to construct the internationally-ratified IntCal calibration curve. For example, the Lake Suigetsu record (Bronk Ramsey et al. 2012) on which PT is based contains only 76 observations from 12 – 13.9 cal kyr BP with $^{14}$C age uncertainties varying between 39 and 145 $^{14}$C yrs (1σ). The Cariaco marine record (Hughen et al. 2006) is one of the more highly-resolved collection of foraminifera yet contains only 24 observations from 14 – 15.9 cal kyr BP with $^{14}$C age measurement uncertainties of around 40 $^{14}$C yrs (1σ).

In light of such sparse sampling and measurement uncertainty, it is unclear how much confidence one can have that identified $^{14}$C-age plateaus, in the target Lake Suigetsu record and the marine record one intends to tune, are genuine atmospheric phenomena and also that they are correctly paired between the two records. One would need to be confident both that the Lake Suigetsu record had identified all genuine atmospheric $^{14}$C-age plateaus and further that one can then pair each identified marine $^{14}$C-age plateau correctly with its corresponding atmospheric $^{14}$C-age plateau. This latter step needs to take into account that the marine $^{14}$C-age plateau is offset by an unknown and potentially varying surface marine reservoir age.

If one fails to identify a genuine atmospheric $^{14}$C-age plateau using the Lake Suigetsu

record, then one cannot presumably be sure to be matching the same $^{14}$C-age plateaus between the records with the consequential risk of severe misalignment of the marine core. As a hypothetical example, suppose there were five true atmospheric $^{14}$C-age plateaus, but only three of these five plateaus were identifiable in the noisy Lake Suigetsu observations. The other two genuine plateaus would therefore remain unknown in our target. Further, suppose one then identified three plateaus in the marine core to be tuned. With there being five true underlying $^{14}$C-age plateaus, these three marine plateaus could correspond to different plateaus from the three identified in the Lake Suigetsu target. Despite this, PT would confidently align the three plateaus in both records. The resulting alignment could however be entirely incorrect, leading to errors in the marine core chronology and MRA reconstruction.

**3.4/** Further statistical difficulties arise in determining $^{14}$C-age plateaus in the marine core since, before PT is performed, one does not have a calendar time scale on which to provide a gradient (in terms of $^{14}$C yr/cal yr). Without an ability to work out the $^{14}$C gradient per cal yr, identifying a $^{14}$C-age plateau is considerably more challenging. A natural option might be to use the depth scale within the core. This would be equivalent to assuming a constant sedimentation rate. However, in then tying/matching subsequent plateau to the atmospheric $^{14}$C-age plateau this assumption of constant sedimentation would be over-ridden and potentially significantly violated. This introduces significant circularity and potential for contradiction into the PT approach. In fact, in previous work, the PT method appears to provide sedimentation rates which vary by up to a factor of 5 to 8 within a single core (e.g. cores PS2644 and MD08-3180 in Sarnthein et al. 2015; PS75/104-1 in Küssner et al. 2018); considerably more, by orders of magnitude, in other cores from the Nordic Seas (Sarnthein & Werner 2017); and even more in ad-hoc hiatuses mentioned above in section 2.4. The question of how one identifies marine $^{14}$C-age plateau in the context of a changing and unknown sedimentation rate/calendar age scale – for which estimates only become available after one has already been required to select plateaus and perform the tuning – does not have a straightforward solution and is prone to confirmatory bias.

**3.5/** The original PT method (Sarnthein et al. 2007, 2011) was based only on visual inspection of the observed $^{14}$C-ages vs depth to determine the $^{14}$C-age plateaus (both the duration of the plateaus and their constant $^{14}$C ages including an unknown marine reservoir age). In an attempt to reduce the subjectivity of eyeball evaluation, Sarnthein et al. (2015) calculated the first derivative of a locally-fitted $^{14}$C-age vs depth curve to identify the $^{14}$C-age

plateaus (i.e., times when the slope drops to near zero). This refinement goes in the right direction, but the authors admit that there is room for subjectivity when choosing the level of smoothing and the threshold for defining a plateau. In addition, their derivation technique does not consider explicitly the different analytical uncertainties of the individual $^{14}$C measurements on atmospheric and marine samples that are quite variable, notably for the Lake Suigetsu $^{14}$C data measured on small plant macrofossils of varying carbon masses (Bronk Ramsey et al. 2012). Such analytical uncertainties should be taken into account because a kink or a plateau in a $^{14}$C vs. depth relationship within a record could be a structure within the analytical errors rather than an atmospheric feature. PT to such random analytical error would be useless.

In any case, Sarnthein et al. (2015) chose to keep the visual inspection as their main tool: *"we continued to base our calculations of reservoir ages, our tuned calendar ages of plateau boundaries, and sedimentation rates on the boundary ages defined by visual inspection.".* In Sarnthein et al. (2020), it seems that eyeball inspection has been preferred again: indeed Table 1 provides $^{14}$C-age plateaus obtained *"by means of visual inspection"* in the target records (Lake Suigetsu varved sediments and Hulu cave stalagmites).

In his CC, Weninger advocates for the use of an alternative automated technique (a "summed probability distribution") proposed by Weninger & Edinborough (2020) for detecting plateaus in the $^{14}$C record. Unfortunately, this paper does not provide the necessary mathematical details to reproduce and test the proposed technique. It should also be noted that Weninger & Edinborough (2020) only claim the detection of 4 plateaus in the 24 to 14 kyr BP time window of IntCal20 (see their Fig. 1), in contrast with the 11 plateaus named by Sarnthein et al. (2020) over the same period.

**3.6/** To assess objectively the ability to reliably identify and tune $^{14}$C-age plateaus, in the context of the noisy and sparse $^{14}$C data currently available to us, we performed a simulation study. For this study, we aimed to investigate two aspects: firstly, can we reliably and robustly identify atmospheric $^{14}$C-age plateaus in data that are of comparable density and precision to those from Lake Suigetsu (Bronk Ramsey et al. 2020); and secondly, having simulated paired $^{14}$C age and depth data from a hypothetical marine core with comparable precision and density to the Cariaco Basin (Hughen et al. 2006), can we use PT to accurately reconstruct the marine record's underlying calendar age scale?

Readers should note that our simulated pseudo-Cariaco marine core is densely sampled compared with the marine cores studied by Sarnthein et al. (2020). Moreover, in our study, we have simplified the tuning problem by setting a constant MRA for our simulated marine record.

The additional complexities introduced to PT should MRAs change at any time other than plateau boundaries are not considered. Consequently, our simulation results should be considered a best-case scenario for the PT method.

To maximize objectivity, we aimed to implement the automated [14]C-age plateau identification approach presented in Sarnthein et al. (2015). The description of their automated approach lacks precise detail to be completely reproducible, however, we hope that our method follows the principles sufficiently closely. Having simulated our cores, to estimate the local [14]C yr/cal yr gradient at any depth, we fit a kernel weighted linear model using a $N(0, 50^2 \ cal \ yr^2)$ kernel (or an analogue in the case of the marine core for which we have no known calendar age scale) and based upon a fixed number of samples in the neighborhood of the depth under consideration. We also show the results using a $N(0, 100^2 \ cal \ yr^2)$ kernel to illustrate the effects on gradient estimation and plateau identification in our simulated pseudo-atmospheric cores. This wider kernel will extend the effective width of the window in which we calculate the local gradient, using more neighbouring observations and performing more smoothing. Further details are given in 3.7 and 3.8. Code is available on request.

Sarnthein et al. (2015) do not specify definite rules as to either the choice of bandwidth (the standard deviation of the weighting kernel) to use when estimating the local gradient within a record; or the subsequent gradient threshold which defines a [14]C plateau. The subjectivity in these selections, which as we show make a considerable difference to the number and locations of plateau, will always reduce the ability of PT to provide an objective approach. Indeed, as explained in 3.5, Sarnthein et al. (2015) admit to a final selection which is based upon agreement with their eye-balled choices.

Sarnthein et al. (2015) trialled two thresholds for the local-gradient to determine a [14]C-plateau. They state a value of 0 [14]C yr /cal yr generated too many short potential plateau periods. In increasing the threshold value, these disconnected short time periods will tend to merge with one another to create longer time periods. However, other disconnected time periods may simultaneously be introduced. Sarnthein et al. (2015) therefore also trialed increasing the threshold to 1 [14]C yr /cal yr which they suggest agreed better with their visual preferences. However, a time period with a gradient of 1 [14]C yr /cal yr is quite a distance from what most would describe as a plateau. Indeed, a gradient of 1 is what the slope ([14]C yr /cal yr) should be without any perturbation of the radioactive decay. Nevertheless, in order to test this issue, we now show our results alongside three potential gradient thresholds of 0, 0.5 and 1 [14]C yr/cal yr one might use to identify a plateau.

**3.7/** To further increase objectivity, we aim to have an underlying "*ground-truth*" atmospheric [14]C baseline for the simulated cores which accurately reflects the true size and scale of atmospheric [14]C-age variation and potential plateaus. Consequently, our study considers the period from 12 – 13.9 cal kyr BP and we use the IntCal20 curve (Reimer et al. 2020) as our ground-truth atmospheric [14]C baseline, shown as a black line in Figures 5a-c. Since

this period of IntCal20 is based upon densely-sampled [14]C determinations from highly-resolved tree-rings, we can be confident it represents genuine atmospheric [14]C-age variation and plateaus. Even here though, exactly what would be considered to constitute a [14]C-age plateau, and how many are present, is ill-defined.

      To this tree-ring based plateau baseline, we create a pseudo-Suigetsu atmospheric [14]C

record by randomly sampling (fairly evenly-spaced) calendar ages between 12 – 13.9 cal kyr BP and adding noise comparable to that seen in the [14]C determinations of Lake Suigetsu (Bronk Ramsey et al. 2020). For this pseudo-atmospheric record, we assume the calendar ages of the [14]C determinations are known precisely, which aids plateau identification.

      We also create a pseudo-Cariaco marine [14]C record by again sampling (fairly evenly-

spaced) calendar ages and adding noise comparable to the [14]C determinations in the Cariaco Basin record (Hughen et al. 2006). To make this analogous to a genuine marine record PT, the calendar age information within the pseudo-marine record was then dropped. We assumed the [14]C measurements were evenly-spaced in depth along the core. We sought to identify if we could then reconstruct this underlying calendar age information using PT.

      **3.8/** We initially simulated three hypothetical cores recording atmospheric [14]C. These aimed to represent data similar to Lake Suigetsu, both in terms of sampling density and [14]C age measurement uncertainties. There are 76 [14]C observations from Lake Suigetsu between 12 and 13.9 kcal BP. For each of our atmosphere-recording pseudo-cores, we sampled N = 76

observations as follows:

1.  Simulate calendar ages $\theta_i$:
   - Sample $U_k \sim Unif[12,13.9]$ for $k = 1, \dots, 2N$
   - Order sampled values to obtain $U_{(1)} < U_{(2)} < \cdots < U_{(2N)}$

- Set $\theta_i = U_{(2i)}$ for $i = 1, \dots, N$ (i.e., every 2nd ordered value)

This provides a set of random (but relatively-evenly sampled) calendar ages.

2.  Simulate $^{14}$C ages $X_i$, for $i = 1, \ldots, N$ as $X_i \sim N(\mu(\theta_i), \sigma_i{}^2)$, where $\mu(\theta_i)$ is the mean of the IntCal20 curve at calendar age $\theta_i$ and $\sigma_i{}^2$ are the variances of the $^{14}$C age measurements reported in the true Lake Suigetsu record (Bronk Ramsey et al. 2020).

To estimate the gradient ($^{14}$C yr per cal year) at any given calendar age $t$, following as best we could the approach described in Sarnthein et al. (2015), we fitted a linear model to the nearest, in terms of calendar age, 60 observations weighted according to their reported uncertainties $\sigma_i$ and a $N(0, 50^2 \, cal \, yr^2)$ kernel centered on $t$ – this gives a weighted moving window, where those observations more than 100-200 cal yr from $t$ have little weight. In our idealized scenario, we assumed the calendar ages of the observations were known exactly for weighting and gradient estimation – in the genuine Lake Suigetsu record these calendar ages are themselves uncertain making gradient estimate somewhat harder.

Figures 5a-c show, for each of the three atmosphere-recording pseudo-cores, the simulated data and estimated gradient (with 95% intervals for the estimate obtained by the weighted linear model). Gradient estimates at the calendar age extremes should be treated with caution and so are not considered below. In each of these plots 5a-c we overlay the three gradient thresholds (0, 0.5 and 1 $^{14}$C yr per cal year) which might be considered as potential plateau indicators as discussed in section 3.6. Shown as a rug at the bottom of each plot are the time periods (color coded according to threshold) for which the gradient falls below each threshold. Fig. 5d shows the gradient estimates for all three cores overlain with the recommended Sarnthein et al. (2015) gradient threshold of 1 $^{14}$C yr/cal yr. Here the rug shows the time periods in all the cores (color coded by core) where the locally estimated gradient falls below 1 $^{14}$C yr/cal yr. For the automated PT approach to be reliable it is necessary, yet not sufficient, for those periods/plateaus in the atmosphere-recording pseudo-cores that one intends to use as an atmospheric tuning target for the marine record to align.

We also repeated our gradient estimation technique using a wider $N(0, 100^2 \, cal \, yr^2)$ kernel. This creates a wider moving window and gives weight to a greater number of neighboring observations when estimating the local gradient. The results, on the same three simulated pseudo-atmospheric cores, with this alternative wider kernel which applies greater smoothing can be seen in Figure 6.

We observe that while the gradients obtained in each simulated core show some of the same main features, differences remain that are critical to PT. In particular, we see that choice of kernel and the gradient cut-off for identifying a plateau is key and selection is non-trivial if one wishes to maintain consistency in creating a robust atmospheric target between records.

Varying these subjective choices, even within the same core, can make a significant difference to the number and location of atmospheric plateaus which one would estimate. This has significant consequences upon the reliability of any subsequent PT of a marine core.

While we can seem to identify elements of the long YD $^{14}$C plateau from 12 – 12.4 cal kyr BP in all our simulated "*Suigetsu-quality*" atmospheric cores, beyond this it is hard to consistently and reliably identify what constitutes a plateau. Whatever the choice of threshold and kernel, the number and timing of plateau periods are not consistent between the three simulated records. Using a threshold of 1 $^{14}$C yr/cal yr and a $N(0, 50^2 \, cal \, yr^2)$ kernel, we may estimate either 5 or 6 plateau periods (Fig. 5d) dependent upon the simulated core we consider. Further, other than the YD plateau, these do not accurately align between the three cores. Even if one sought to merge the plateaus, it is unclear as to how one would know which to merge with others.

If one has different numbers of $^{14}$C-age plateaus in the atmospheric target, then one will tune the marine core very differently. A PT using the blue simulated record as an atmospheric target (Fig. 5a) would likely lead to very different results than tuning to the red simulated record (Fig. 5b). We see in Fig. 5d the lack of consistency in the alignment of the identified $^{14}$C-age plateau periods from 13 – 14 cal kyr BP making it very hard to know what plateaus one should aim to tune. In the time period we consider for our study, Sarnthein et al. (2020) identify two plateaus they aim to use as tuning target (at 11.9 – 12.48 kcal BP; and 13.66 – 14.04 cal kyr BP) plus potentially a further plateau which it is unclear if they always use at 12.78 – 13.08 cal kyr BP. However, in our simulation study we find just as strong support for a plateau around 13.4 cal kyr BP in some of our simulated cores. Further, looking at the underlying IntCal20 estimate which, as explained above, is tree-ring based and highly resolved in this period, the level of visual evidence for a plateau from 12.78 – 13.08 cal kyr BP is unclear.

In conclusion, we see that using different *Suigetsu-quality* atmospheric $^{14}$C records to initially identify $^{14}$C-age plateaus could lead to quite different target matchings for the marine records. Not only might the atmospheric target $^{14}$C-age plateaus identified not align with those one would have obtained using a different initial atmospheric record, but one might also be aiming to identify different number of $^{14}$C-age plateaus in the marine core. Further, if there are in fact more genuine atmospheric $^{14}$C-age plateaus than the atmospheric-recording core indicated, this could lead to erroneous pairings between the atmospheric and marine records.

**3.9/** We also created two pseudo-marine $^{14}$C records to span 12 – 13.9 cal kyr BP, again using the tree-ring based section of IntCal20 as our ground-truth atmospheric $^{14}$C baseline. For

these pseudo-marine records, we aimed to represent the relatively high density and precision of the Cariaco Basin unvarved [14]C record (Hughen et al. 2006). These simulated pseudo-marine cores are created similarly to the pseudo-atmospheric simulated records above but, after creation, we remove the calendar age information that aids [14]C yr/cal yr gradient calculation. This makes them analogous to the marine cores used by Sarnthein et al. (2020). Instead [14]C-

age plateaus in the pseudo-marine must be identified using only their ordering (or simulated depth) within the core – a considerably more challenging and less robust task. We aim to compare our simulated marine [14]C records against our pseudo-atmospheric cores of section 3.8 to assess the ability to identify, and match, shared [14]C-age plateaus.

For this element of the study, we make the very strong simplifying assumption that there

are no MRA changes in the marine records. This simplifying assumption will aid us considerably in identifying the atmospheric component of the [14]C-age signal in our simulated marine records. As such, this part of our study only considers the effect on identifying plateaus in light of the increased sparsity in marine records and the lack of timescale on which to reliably infer the [14]C to calendar age gradient. MRA changes will add a very considerable further layer

of confounding and difficulty.

For IntCal13, in the 14 – 15.9 cal kyr time period, the Cariaco Basin unvarved [14]C record contained 24 observations (Hughen et al. 2006). Note that this 14 – 15.9 cal kyr period is used as representative of sampling density since the Cariaco unvarved record does not extend to 12 cal kyr. For each pseudo-marine core, mirroring the approach given in our pseudo-atmospheric

simulated records, we simulated N = 24 random observations with underlying calendar ages again sampled according to every $2^{nd}$ ordered value of a uniform distribution to create relatively equi-spaced ages ranging from 12-13.9 cal kyr BP. However, for these marine cores, we selected [14]C age measurement uncertainties that matched those in the Cariaco Basin ($1\sigma$ of approximately 40 [14]C yr) and applied an adjustment of a constant MRA of 400 [14]C yrs. The

resultant simulations are shown in Fig. 7 – with the simulated data shown in Fig. 7a. Since the Cariaco Basin [14]C record is quite densely sampled compared to marine records studied by Sarnthein et al. (2020), and we have applied a constant MRA, this set-up provides a best-case scenario for PT.

As discussed above and in section 3.4, estimating the gradient ([14]C yr/cal yr) and

identifying [14]C-age plateaus is made more complex here since, in a marine core, the calendar ages are unknown before PT. One is therefore required to select a prior, pseudo-calendar, scale on which to estimate a gradient and hence identify plateaus. We chose to estimate the gradient by assuming our [14]C observations are equi-spaced in depth (and hence equi-age-spaced along

our pseudo-calendar scale) along the core as shown in Fig. 7b. Since, by construction, our true

calendar ages, $\theta_i$, are relatively evenly-spaced, this pseudo-calendar scale should equate to a

sedimentation rate which is not unrealistically variable compared to the reconstructed estimates

inferred by the PT in Sarnthein et al. (2015), Sarnthein and Werner (2017), and Küssner et al.

(2018). The inferred sedimentation rate which would generate this equi-spacing in depth can

be seen at the foot of Fig. 7b. The maximum variation in sedimentation rate within the simulated

cores is of the order of 10. Such an equi-spacing can equivalently be interpreted as using the

ordering-information only to determine the gradient, i.e., the change we observe moving from

one $^{14}$C observation to the next. This is a natural approach as it requires no a priori assumptions

regarding the unknown true calendar ages.

The same linear model approach as used for the pseudo-atmospheric cores was then

applied with weightings determined according to our even-depth observational spacing (i.e.,

using the observational order only). We used the nearest 20 marine $^{14}$C observations and a

kernel on the depth scale analogous to the $N(0, 50^2 \; cal \; yr^2)$ used for the pseudo-atmospheric

observations. This depth kernel was applied so that 50 cal yrs corresponded to 50/1900 of the

pseudo-marine core's depth (i.e., so the 1.9 kyr period between 12-13.9 cal kyr period covered

the full pseudo-marine core). The obtained pseudo-gradient estimates for the cores are shown

in Fig. 7b on this even-spacing (or observational order) basis.

We would select $^{14}$C-age plateaus, and the observations belonging to them, based upon

Fig. 7b with its even-depth-spaced, pseudo-calendar timescale. We have overlain thresholds of

0, 0.5 and 1 $^{14}$C yr/cal yr on this pseudo-gradient scale. Shown at the top of this plot are the

sections of each simulated core (color-coded according to the core) for which the local pseudo-

gradient lies below a threshold of 1 $^{14}$C yr/pseudo-cal yr. For example, in the simulated marine

core A (shown in purple) we might identify an order-based plateau ending around the 6$^{th}$

observation (perhaps covering the 1$^{st}$ to the 6$^{th}$) since in this neighborhood the $^{14}$C gradient on

the pseudo-calendar scale is below 1.

We can infer where any order-based $^{14}$C-age plateaus we identify correspond to on the

true calendar age timescale by transforming back from the ordered/even-depth spacing to the

underlying calendar ages of our observations. Fig. 7c shows the order-based gradient estimates

(and the observations in each simulated marine core) when plotted against the true "*unknown*"

calendar age timescale. In our example, the $^{14}$C-age plateau identified between the 1$^{st}$ and 6$^{th}$

observation in simulated marine core A of Fig. 7b corresponds to a $^{14}$C-age plateau covering

the interval 12-12.5 cal kyr BP. Shown as a rug are the underlying time periods for the sections

of each core (color-coded) we would assess as corresponding to a $^{14}$C-age plateau applying a threshold of 1 $^{14}$C yr/pseudo-cal yr.

In Fig. 7d, we overlay the order-based marine-core $^{14}$C gradients, after they have been transformed back to their true underlying timescales, against the $^{14}$C gradient obtained from our first simulated *Suigetsu-quality* atmospheric record. The rug indicates the plateaus one would identify, based on a threshold of 1 $^{14}$C yr/cal yr in the simulated (blue) atmospheric record and 1 $^{14}$C yr/pseudo-cal yr in the two simulated (purple and orange) marine records. For PT tuning to provide reliable MRA estimates, the identified plateaus (or dips in gradient) in the simulated marine cores and the simulated atmospheric $^{14}$C record should align in Fig. 7d. This does not reliably occur.

The large amount of noise in the marine records, their sparsity, and critically their unknown true time scale make identification of any $^{14}$C plateaus very difficult with the potential to be unreliable and inconsistent. Highly different numbers and locations of $^{14}$C plateaus would be indicated in applying the method to the two simulated marine records; and for all gradient thresholds. Furthermore, the marine $^{14}$C-age plateaus may not correspond to the $^{14}$C-age plateaus in the atmospheric record. Should one attempt to align by PT the indicated $^{14}$C-age plateaus of either of the simulated marine cores to the atmospheric target, one would obtain significantly incorrect chronologies for the marine cores. Further, this misalignment would then result in significant inferred MRA changes – such changes in MRA would be incorrect since the underlying marine records were created with a constant MRA.

While simple, this simulation study illustrates some of the potential difficulties of identifying $^{14}$C-age plateaus in the presence of observational noise and a lack of sampling density. Were the simulated data to have much higher precision and be sampled much more densely, we would expect the $^{14}$C-age plateaus to align more consistently. Such a preliminary study with an exact and reproducible approach is needed to assess the robustness of the method. The fundamental questions of how to remove the potentially confounding effects of MRA changes, the lack of a timescale for the marine records on which to calculate the $^{14}$C-age gradient combined with significant sedimentation rate changes, and other potential geoscientific factors in identifying $^{14}$C-age plateaus will however still remain.

**3.10/** Finally, implicit in the Sarnthein et al. (2020) paper is a suggestion that using Lake Suigetsu alone provides a more precise reconstruction of atmospheric $^{14}$C levels from 55 – 13.9 cal kyr BP than the IntCal20 synthesis (Reimer et al. 2020). Specifically, that by combining $^{14}$C records from a diverse range of archives, the IntCal curves lose genuine atmospheric structure

that can be extracted from Suigetsu. This perspective is suggested by Sarnthein et al. (2020) through an argument that the IntCal20 curve has lesser variation, and fewer wiggles, from 55 – 13.9 cal kyr BP, where it is based upon a range of archives, than from 13.9-0 cal kyr BP, where it is based upon highly resolved tree-ring determinations.

Indeed, Reimer et al. (2020) recognize the limitations of the current archives on which they base the IntCal20 curves. The ideal is for a truly atmospheric $^{14}$C record extending back to 55,000 cal yr BP that also provides sufficient detail to reconstruct the high-frequency component of the $^{14}$C signal reliably and precisely. However, this characterization of IntCal20, as overly smooth and hence unreliable from 55 – 13.9 cal kyr BP, is to misunderstand what the

IntCal curve represents.

The values published as the IntCal curves aim to provide pointwise summaries of the $^{14}$C age, in terms of the mean and uncertainty, at any chosen calendar age. This is not the same as trying to represent the level of $^{14}$C variation from one year to the next. Critically, a smooth pointwise mean does not necessarily imply no variation in atmospheric $^{14}$C levels. More likely,

it represents that we do not yet know when, and with what magnitude, any such $^{14}$C variations occur.

Construction of the IntCal20 curve uses Bayesian splines (Heaton al. 2020b). The outputs of this Bayesian approach are a large set of posterior curve realisations that aim to find a trade-off between passing near the observed $^{14}$C determinations on which the curve is based,

while not being so variable as to overfit the data and introduce spurious features which are simply artefacts of the analytical sampling errors within the $^{14}$C observations. The pointwise summary is then based upon averaging over 2,500 of these curve realisations.

The IntCal approach does, in fact, assume there are similar levels of short-term atmospheric variability from 55 – 13.9 cal kyr as in the tree-ring based section from 13.9 – 0

cal kyr BP. This can be seen by looking at 5 individual posterior realisations, randomly selected from the 2500 used to create the final IntCal20 pointwise summary, shown in Figure 8. All these individual realisations exhibit significant short term $^{14}$C variability, although none on the scale of the hypothesized Sarnthein et al. (2020) plateaus as they must still agree with the pointwise IntCal uncertainties (shown at the 95% level as the dotted purple envelope).

However, as we do not yet have sufficiently detailed $^{14}$C measurements, the precise timing and magnitude of the fluctuations in the realisations is unknown. Consequently, when averaged to provide the pointwise estimates, the realisations generate an IntCal pointwise mean (shown as solid purple) that is smoother than any individual realisation.

**4/ Conclusion and Outlook**

Creating reliable calendar chronologies for deep-sea cores, and understanding the scale and timings of MRA changes ($^{14}$C depletion) as well as their variation across the globe, are key for the understanding of our past Earth system and carbon cycle. If the MRA is known, then

$^{14}$C calibration can be used to create the calendar chronology. Conversely, if the calendar chronology is known, $^{14}$C can be used to infer the MRA. When neither the MRA nor the calendar chronology are known in advance, this creates a challenging problem. However, in such circumstances when neither MRA nor calendar chronology is known, we have strong reservations regarding the ability of PT to robustly provide such chronologies, or accurately

infer MRA changes.

$^{14}$C changes in marine sediment cores do not accurately mirror atmospheric variability due to attenuation, phase shift of the atmospheric signal, as well as variability in sedimentation rate, MRA and foraminifera assemblages. Further, low resolution and noisy $^{14}$C datasets, particularly hard to avoid in marine sediment cores, severely limit the ability to reconstruct

fine-scale structure needed to tune to atmospheric $^{14}$C signal. Consequently, the identification and tuning of hypothesized $^{14}$C-age plateaus between sparse marine sediment and atmospheric records may lead to significant misalignment. The resultant calendar chronology for the marine sediment, and inferred MRA changes, may therefore be equally incorrect.

Techniques do exist to provide calendar chronologies for deep-sea cores independently

from $^{14}$C (and which can then be used to infer MRA changes). These utilise alternative age links between proxies in combination with age-depth modelling. One approach is to use climate proxies with the sediment cores, for example these have been tied to the oxygen isotopic ($\delta^{18}$O) profile of Hulu Cave stalagmites which have been accurately dated by U-Th (Bard et al. 2013, Heaton et al. 2013, Hughen and Heaton 2020). Potential age links are also provided by dated

tephra layers (e.g. Bard et al. 1994, Siani et al. 2013). Some of these approaches have been developed by the INTIMATE group and are discussed and implemented in Austin et al. (2011) and Waelbroeck et al. (2019). The latter includes some of the sediment cores in Sarnthein et al. (2020).

Even amongst these alternative approaches to create calendar chronologies for deep-sea

cores, those based upon climate proxies (i.e., all except the use of tephra) require significant assumptions about the global synchronicity of the climate changes used to tie the records together. Although evidence does support an assumption of globally-synchronous timing for certain rapid paleoclimatic changes (Corrick et al. 2020), it is important to consider the specific

climate changes used for tying, and to recognize the uncertainty which potentially non-exact
synchronicity introduces in the resulting marine core calendar chronology (Heaton et al., 2013). The use of these climate proxy approaches to tuning will also mask possible non-synchronous climate changes and precludes studying the climate dynamics on the centennial to pluri-decennial time scale (Mekhaldi et al., 2020).

Regarding the identification of atmospheric $^{14}$C variation, we recognise that a precise and
highly resolved, directly-atmospheric, set of $^{14}$C archives extending back to 55 kcal BP would provide a step-change in our ability to understand higher frequency variations in past $^{14}$C levels. While the Lake Suigetsu $^{14}$C record provides a first step towards this goal, it is not of sufficient resolution or density to provide this – and, as explained in Sarnthein et al. (2020) and Bronk Ramsey et al. (2020), its varve counted timescale needs adjustment using the Hulu Cave $^{14}$C
record. To accurately identify the higher frequency components of $^{14}$C variation from 55 – 14 cal kyr BP we must await recovery of new archives, notably floating tree-ring sequences.

IntCal20, compared to previous versions of the $^{14}$C calibration curve, profits from a multitude of improvements in all $^{14}$C archives beyond 14 cal kyr. These include new high-resolution $^{14}$C and U-Th data from Hulu Cave (Cheng et al. 2018); reanalysis of the Lake
Suigetsu timeline (Bronk Ramsey et al. 2020); updates in the calendar scale and MRA modelling of the Cariaco basin $^{14}$C data set (Hughen and Heaton, 2020); floating tree ring $^{14}$C sequences (Adolphi et al. 2017, Turney et al. 2010, 2016, Capano et al. 2018, 2020); and modelling MRA time variations to correct the marine based records (Butzin et al. 2020). These have then been combined using a more robust Bayesian spline statistical approach (Heaton et
al. 2020b) that aims to identify shared features seen in multiple records, without being overly influenced by individual outlying $^{14}$C measurements that may not provide an accurate representation of atmospheric $^{14}$C levels. However, as accepted by the IntCal20 authors, the sampling resolution and nature of the diverse records (including their uncertain calendar timescales as well as the indirect nature of $^{14}$C records taken from both stalagmites and marine
sediment cores) mean that centennial $^{14}$C variations are not able to be reliably resolved at present beyond 14 cal kyr BP. The discovery of further floating $^{14}$C tree ring sequences (e.g., Cooper et al. 2021) hold promise in addressing this.

The extreme variations of the PT inferred sedimentation, including frequent hiatuses, should be tested with independent techniques in order to prove that PT is reliable, and that
sedimentation rate variations and hiatuses are not artefacts of PT. For these crucial tests, PT should be performed completely independently from the results obtained with other techniques (e.g. tuning with tephra or with climate proxy records). In other words, these other time markers

should not be combined to PT if one wants to test the validity of this method.

In order to further understand the possible existence of long $^{14}$C plateaus, it may also be useful to plan new model simulations taking into account carbon cycle and oceanographic changes during Heinrich and Dansgaard-Oeschger events. Numerical experiments performed with box-models and 3D Earth system models with relevant changes (e.g., of the Meridional Overturning Circulation of the ocean), may be compared and would provide a stringent test in a wiggle-matching exercise.


**Acknowledgments**: We thank the Editor, André Paul, the Referees, Paula Reimer and anonymous, and the Commenters, Michael Sarnthein, Pieter Grootes, Frank Lamy, Helge Arz, Elisabeth Michel, Giuseppe Siani and Bernhard Weninger for comments which helped us in revising and strengthening our paper. TJH was funded by a Leverhulme Trust Fellowship RF-2019-140\9. EB is supported by EQUIPEX ASTER-CEREGE and ANR CARBOTRYDH.

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

**Figure captions:**

**Figure 1:** Blue dots show the Lake Suigetsu $^{14}$C data with their 1σ analytical uncertainties in both radiocarbon and calendar age (Bronk Ramsey et al. 2020). The thin red solid line shows the pointwise posterior mean of a radiocarbon calibration curve constructed using the same Bayesian statistical method as IntCal20 (Heaton et al., 2020b, Reimer et al. 2020), but based only on the Suigetsu $^{14}$C data. The accompanying shaded interval represents the 95% posterior predictive probability interval. Superimposed thick green lines indicate the 15 atmospheric plateaus with their numbering as listed in Table 1 of Sarnthein et al. (2020).

**Figure 2:** The thin purple solid line shows the pointwise posterior mean of the IntCal20 curve with the shaded interval representing the 95% posterior predictive probability interval (Reimer et al. 2020, Heaton et al., 2020b). Superimposed thick green lines indicate the 15 atmospheric plateaus with their numbering as listed in Table 1 of Sarnthein et al. (2020).

**Figure 3:** The two panels represent the same data shown in Figs. 1 and 2, but converted in $\Delta^{14}$C unit in ‰. Short (or zero) gaps between plateaus are transformed into abrupt rises. Age plateaus correspond to the second parts of these atmospheric $\Delta^{14}$C wiggles, during which the $\Delta^{14}$C decrease compensates the radioactive decay.

**Figure 4: 4a/** Amplitude ratio of $\Delta^{14}$C wiggles in the atmosphere and the surface ocean created by sinusoidal changes of the $^{14}$C production as an input to the 12-box model shown in the upper insert (Bard et al. 1997). This normalized attenuation factor is plotted versus the signal period of production variations (on a log scale). The greater the attenuation, the more the amplitude of the sinusoid in the surface ocean is reduced. The factor would be equal to 1 if atmospheric and oceanic amplitudes were the same. The light blue dashed curve shows the calculation results for the Indo-Pacific surface box, while the dark blue dotted line stands for the Southern Ocean surface box. These two boxes differ by their surface $^{14}$C reservoir age at steady state (320 and 890 yr for the Indo-Pacific and Southern Ocean surface boxes, respectively). The vertical dash-dot black line underlines the 500-yr period wiggle used to construct Fig.4b. The inset graph shows the geometry of the model by Bard et al. (1997) with numbers on boxes indicating their steady-state $\Delta^{14}$C (‰), numbers in parenthesis stand for halving the meridional overturning circulation (MOC). Note that the main goal of the simulations in Fig. 4 is to demonstrate that age plateaus almost disappear and are delayed in the surface ocean due to the smoothing effect

of the carbon cycle, even if it stays strictly constant. These simulations are not intended to simulate $^{14}$C and MRA changes due to variations of the ocean circulation. On this subject, it should be noted that part of the simulations presented by Goslar et al. (1995) and mentioned in sections 2.3 and 2.8, were obtained with the very same box-model used for our simulations, but with a variable MOC.

**4b:** $^{14}$C age versus calendar age plot computed for sinusoidal $^{14}$C production wiggles with a period of 500 years and an amplitude change of ± 15 % around the mean value (in order to produce oscillations around the 1:1 line, $^{14}$C ages are calculated with the true half-life of 5730 yr). The red solid curve shows the evolution for the tropospheric box, exhibiting $^{14}$C plateaus (marked with black arrows) when the slope of the relationship goes down to zero. The age plateau corresponds to the second part of the atmospheric $\Delta^{14}$C wiggle, during which the $\Delta^{14}$C decrease compensates the radioactive decay. The light blue dashed curve shows the calculation results for the Indo-Pacific surface box, while the dark blue dotted line stands for the Southern Ocean surface box. The blue curves are offset with respect to the atmospheric curve by their respective marine reservoir ages. In both cases, the slope of the relationship does not decrease to zero, implying the absence of true $^{14}$C age plateaus in the surface ocean.

**Figure 5:** Simulation study to identify the ability of a Suigetsu-style record to reliably identify atmospheric $^{14}$C-age plateaus. In panels a)-c) we present three simulated atmospheric records generated by sampling, subject to noise, from the high-precision tree-ring-based section of IntCal20 between 12-13.9 cal kyr BP (shown as a black line) with a sampling density matching that of Lake Suigetsu (Bronk Ramsey et al. 2020).. The level of noise added to create the simulated $^{14}$C observations (blue, red and green dots) is of an equivalent level to that present in the Lake Suigetsu $^{14}$C. For each simulated set of observations, we present an estimate of the local gradient (shown as blue, red and green curves with their 95% confidence intervals) according to a locally-weighted linear model as proposed by Sarnthein et al. (2015). These local gradient estimates are obtained using a $N(0, 50^2 \; cal \; yr^2)$ kernel to provide the weightings. We overlay three gradient thresholds (0, 0.5 and 1 $^{14}$C yr/cal yr) which might be used to identify a $^{14}$C-age plateau. Shown as a rug at the bottom of each plot are the time periods in each core which correspond to a local gradient below each threshold (color-coded by threshold). In panel d) we overlay the gradient estimates to assess consistency (or lack of) between the three simulated cores in terms of the number and location of $^{14}$C-age plateaus one might identify. As a rug, we plot the time periods in each core (color-coded by core) which correspond to a local gradient below the threshold of 1 $^{14}$C yr/cal yr proposed by Sarnthein et al. (2015).


**Figure 6:** As Fig. 5 but using a wider $N(0, 100^2 \; cal \; yr^2)$ kernel to provide the weightings for the local gradient estimate. The same three simulated atmospheric cores are used.

**Figure 7:** Simulation study to identify the ability to identify [14]C-age plateaus in marine records

for which calendar age scales are not initially known. In panel a), we simulate two marine records between 12-13.9 cal kyr BP, again using the tree-ring-based IntCal20 mean as the ground truth. These simulated marine cores are based upon the sampling density of the Cariaco Basin unvarved record (Hughen et al. 2006), with a similar level of observational [14]C noise, and have been created with a constant MRA of 400 [14]C yrs. To identify plateaus, we first

estimate the gradient on the basis of an unknown calendar age scale. We create a pseudo-calendar age scale by rescaling the observations so they are equi-spaced along the cores before applying our locally-weighted linear model approach, using an equivalent $N(0, 50^2 \; cal \; yr^2)$ kernel, to estimate the gradients on this pseudo (equi-spaced) scale. Estimates of pseudo-gradients, and implied relative sedimentation rates, on this equi-spaced pseudo-timescale are

provided in panel b). At the top, we plot (color-coded by core) the section of core for which we would obtain a local pseudo-gradient below 1 [14]C yr/pseudo-cal yr and which could be identified as a [14]C-age plateau. Scaling back to the true, underlying calendar age timescale, panel c) indicates where, in terms of the actual calendar age scale, one might classify plateaus. In panel d), these are overlain against the first of our simulated atmospheric records of Fig 5 to

assess synchroneity. As a rug we show the time periods, on the true underlying calendar age timescale, corresponding to local gradient threshold estimates below 1 [14]C yr/cal yr for the simulated atmospheric record and 1 [14]C yr/pseudo-cal yr for the two simulated marine records.

**Figure 8:** Plot of five individual spline realisations randomly selected from the Bayesian

posterior of the Markov Chain Monte Carlo approach used to create IntCal20. The published IntCal20 consists of pointwise means (shown as purple solid line) and 95% pointwise predictive intervals (purple dotted line) which are obtained by averaging over 2500 of these individual curve realisations (Heaton et al. 2020). Superimposed thick green lines indicate the 15 atmospheric plateaus with their numbering as listed in Table 1 of Sarnthein et al. (2020).


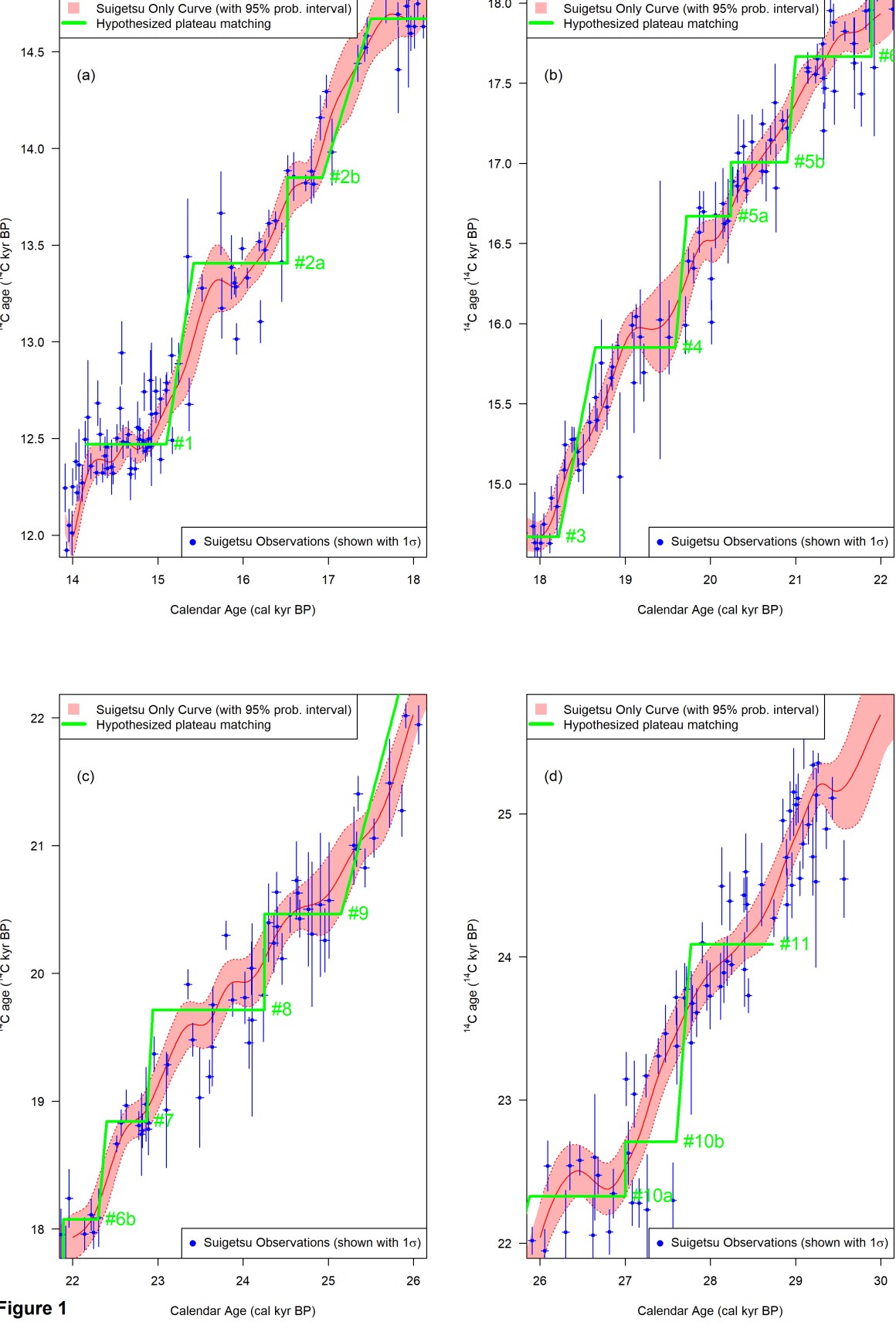

**Figure 1**

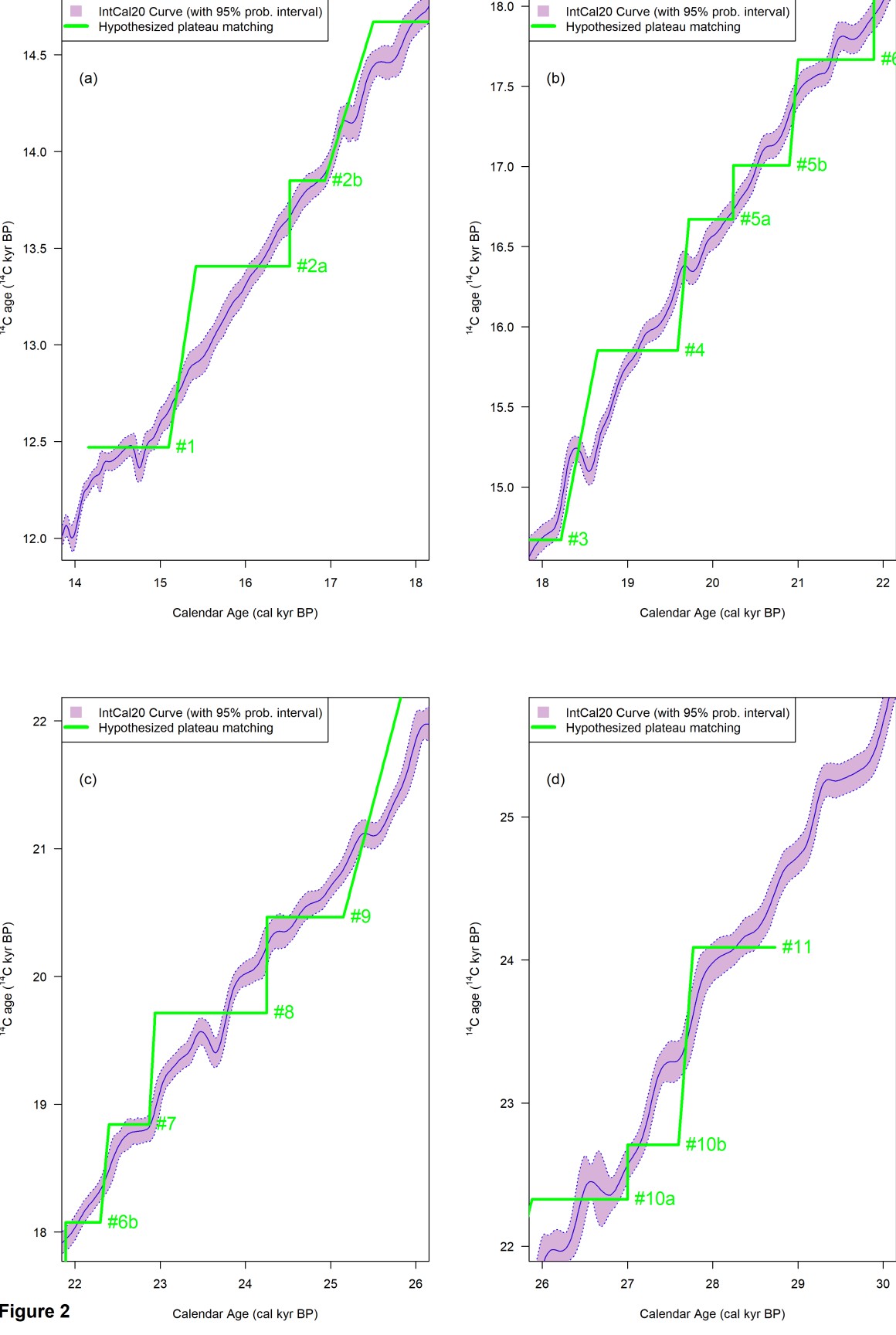

**Figure 2**


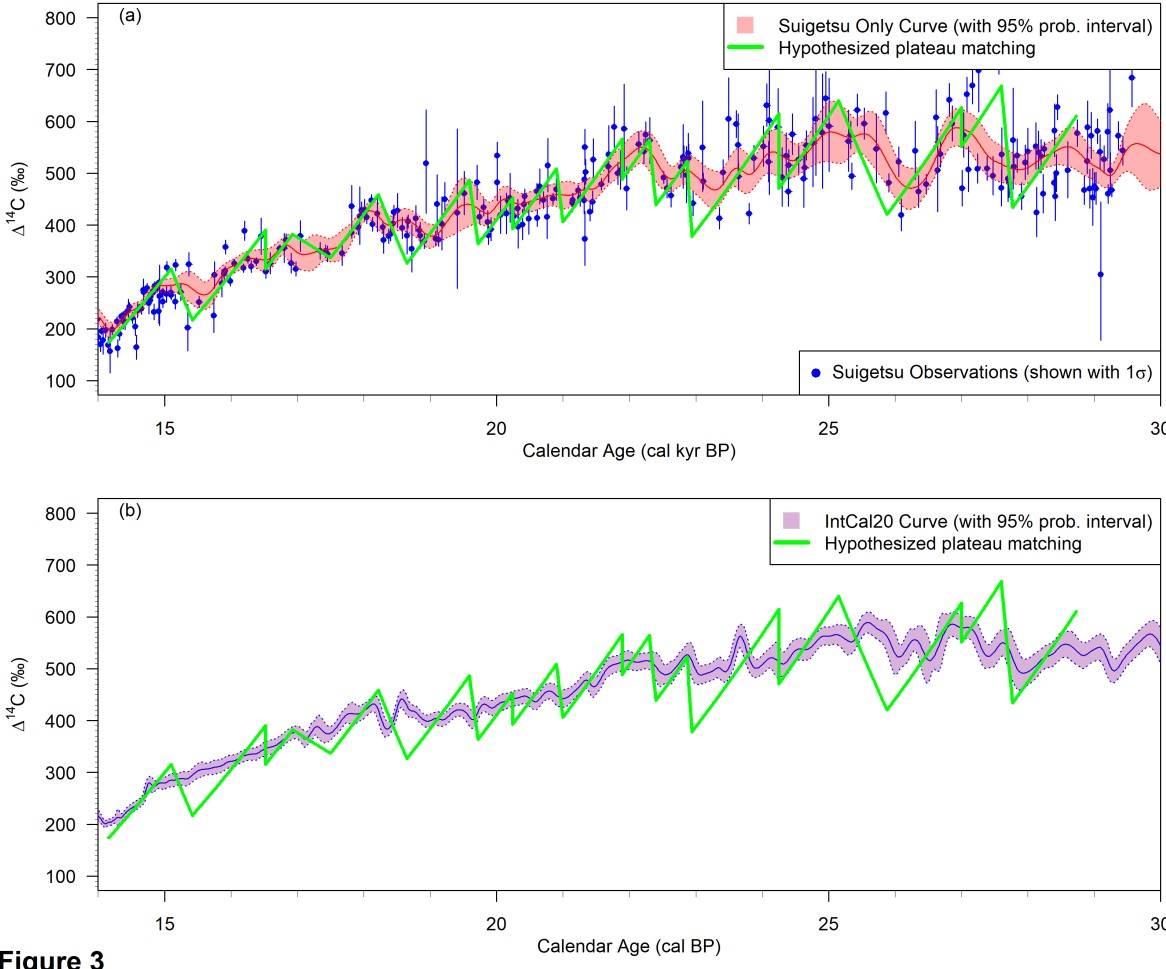

**Figure 3**

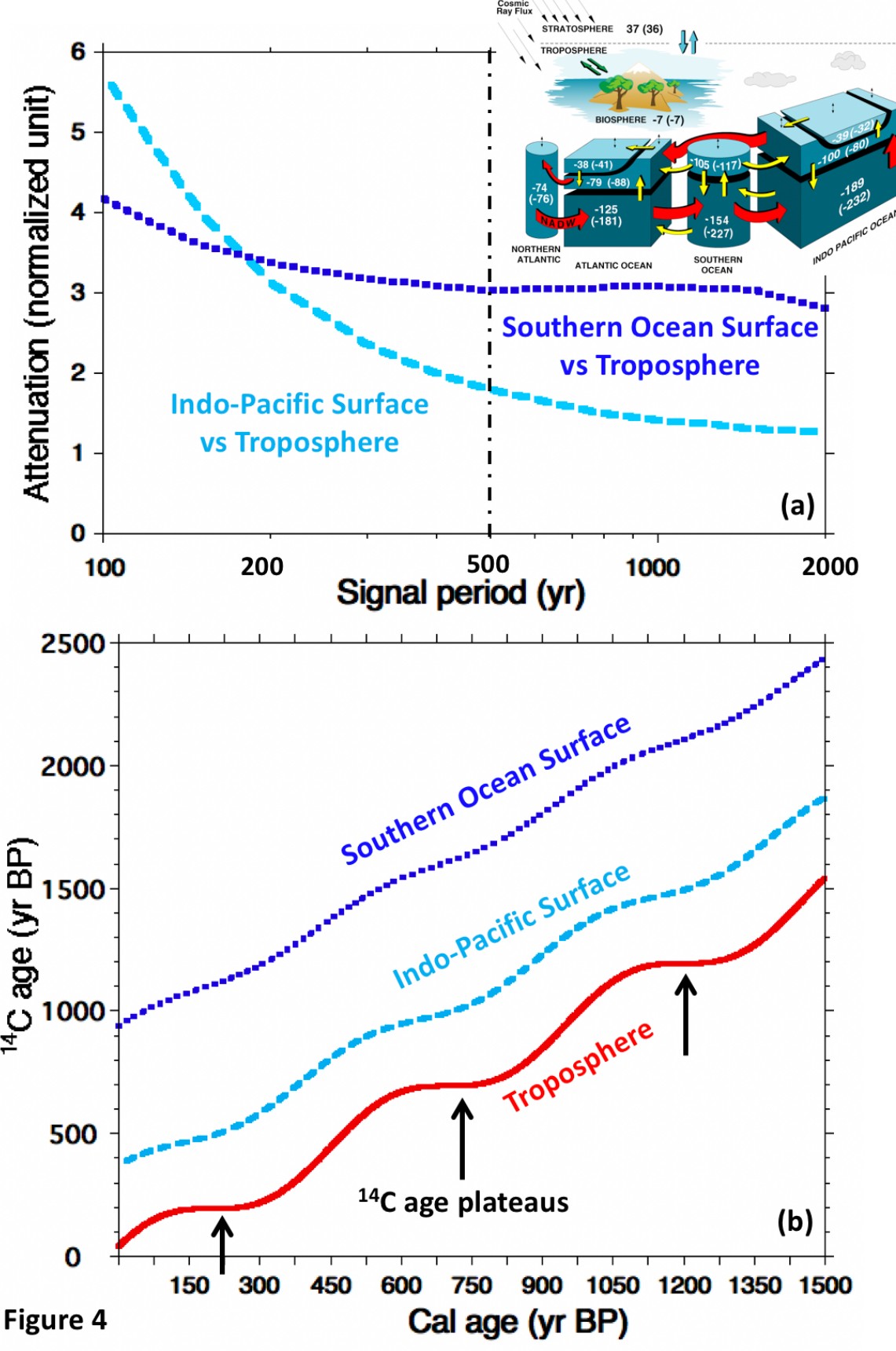

**Figure 4**

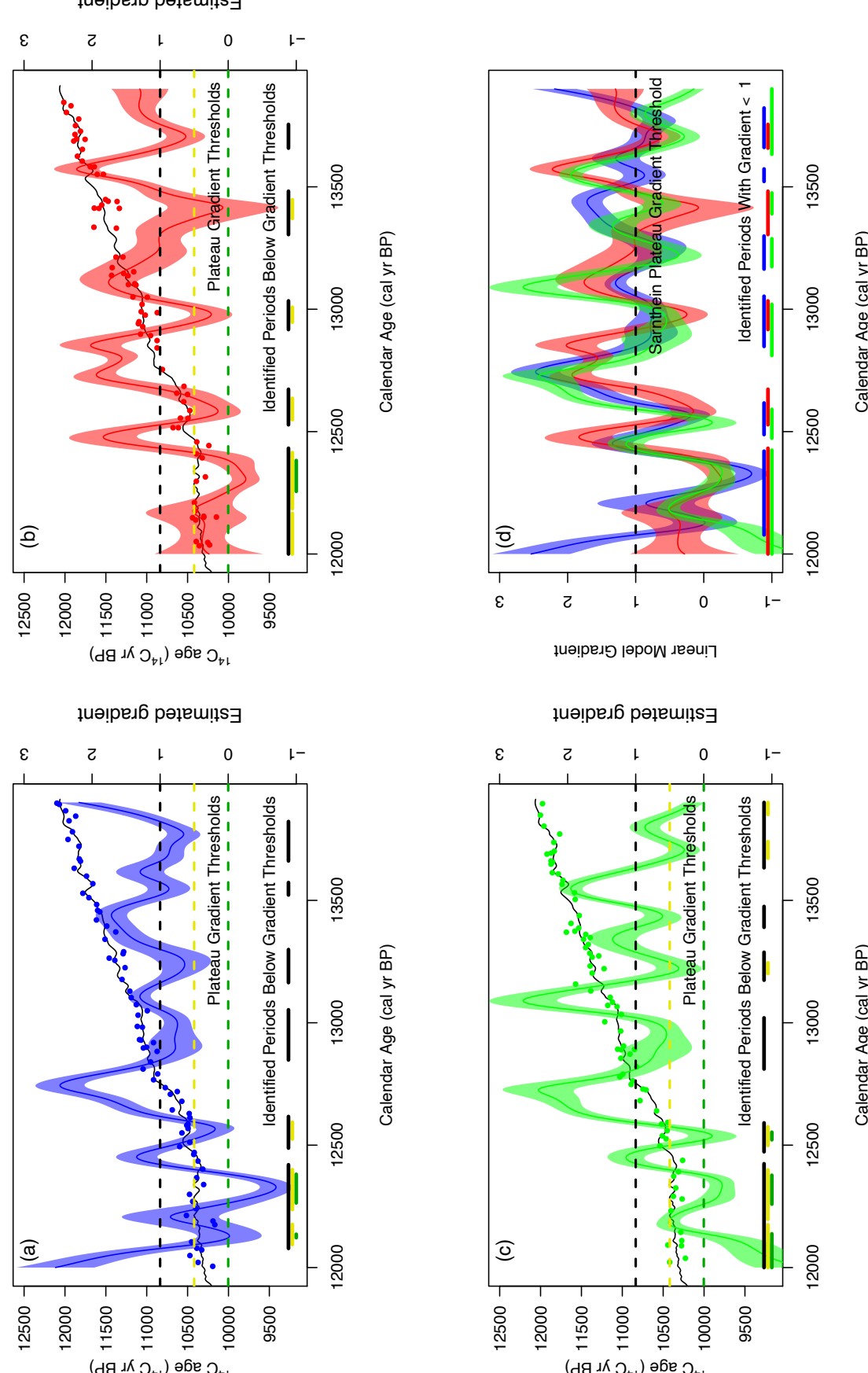

**Figure 5**    Bandwidth of 50 cal yr i.e. kernel of N(0, 50²)


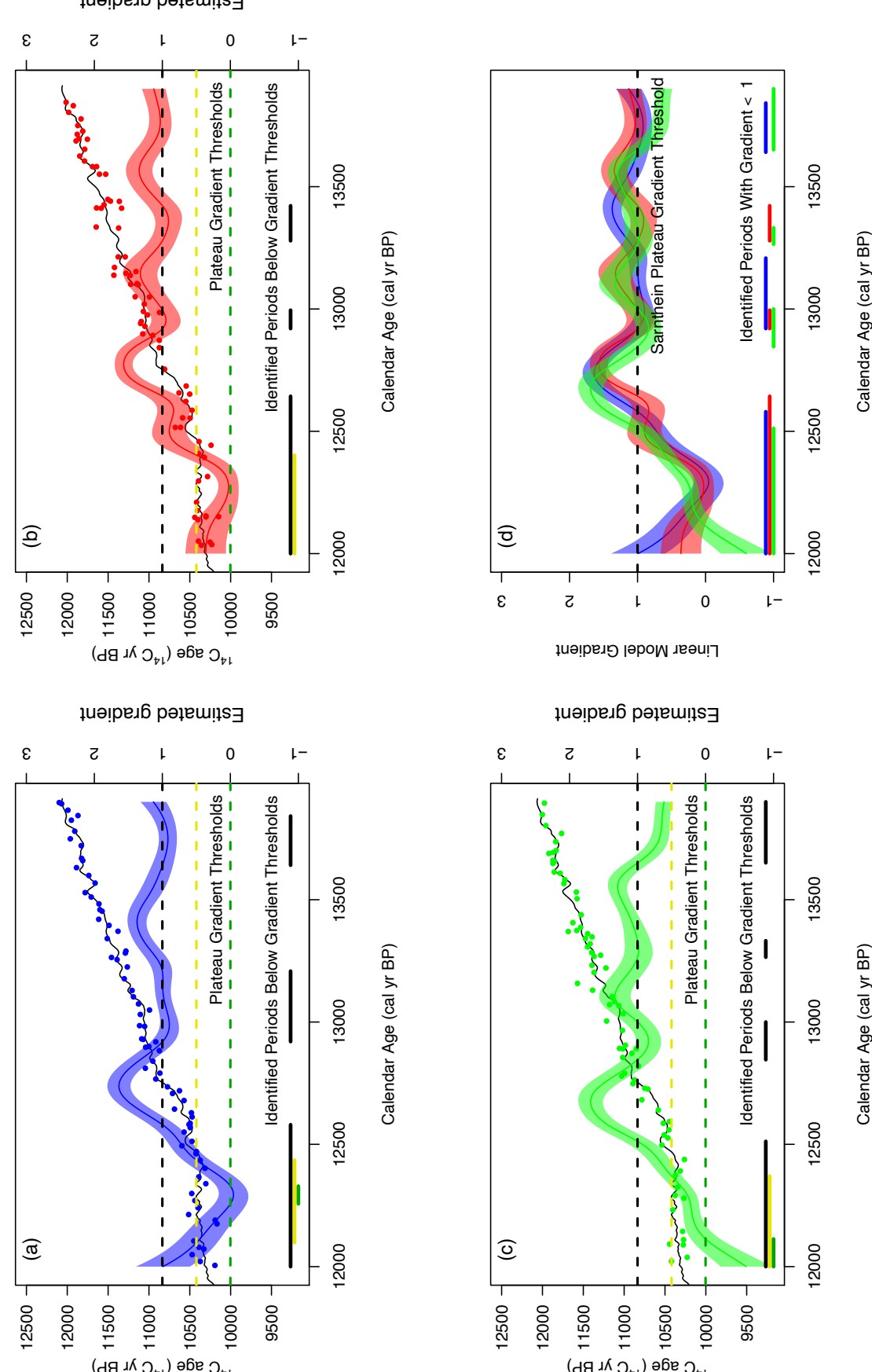

**Figure 6**   Bandwidth of 100 cal yr i.e. kernel of N(0, 100²)

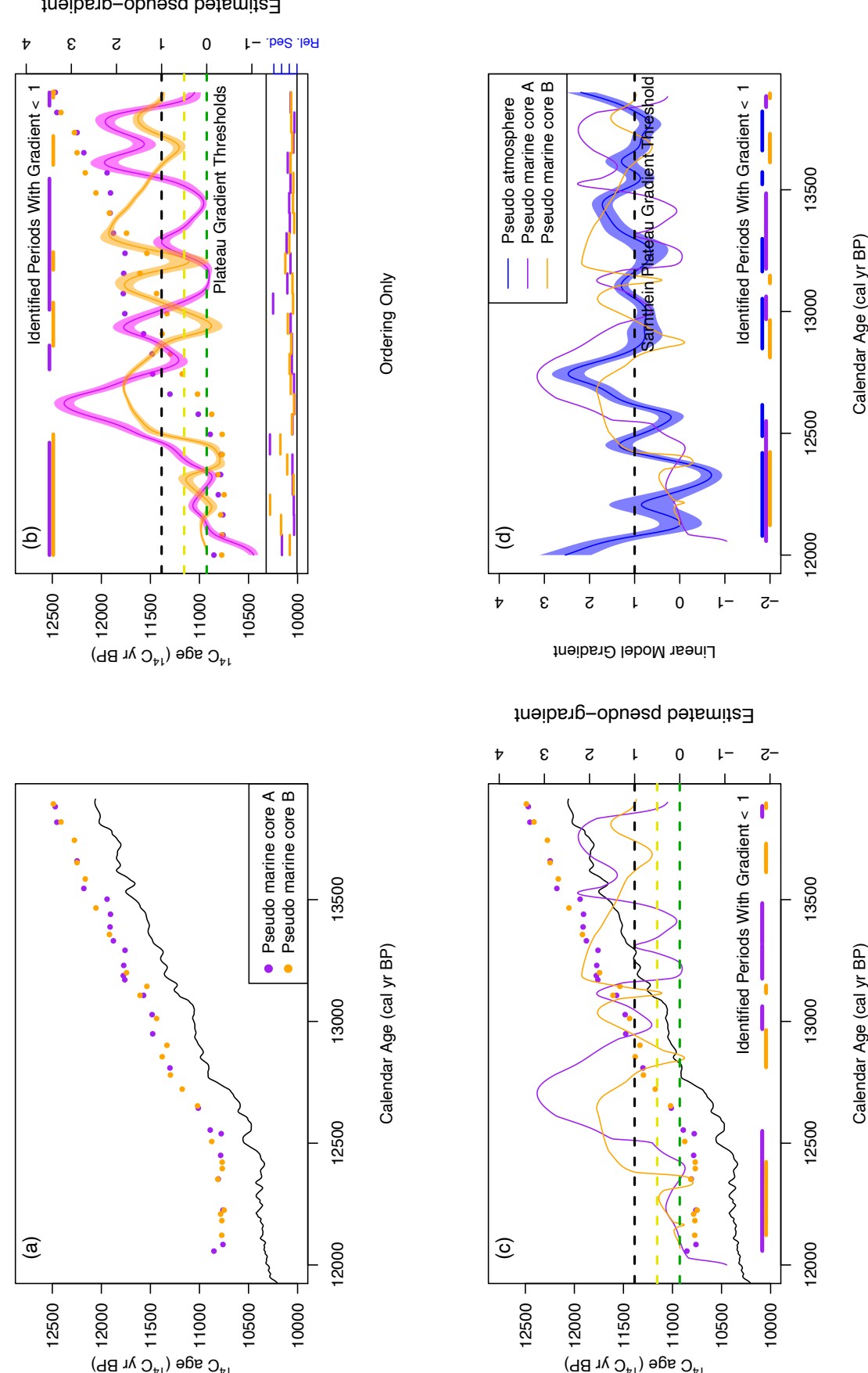

**Figure 7**    Bandwidth of 50 cal yr i.e. kernel of N(0, 50²)

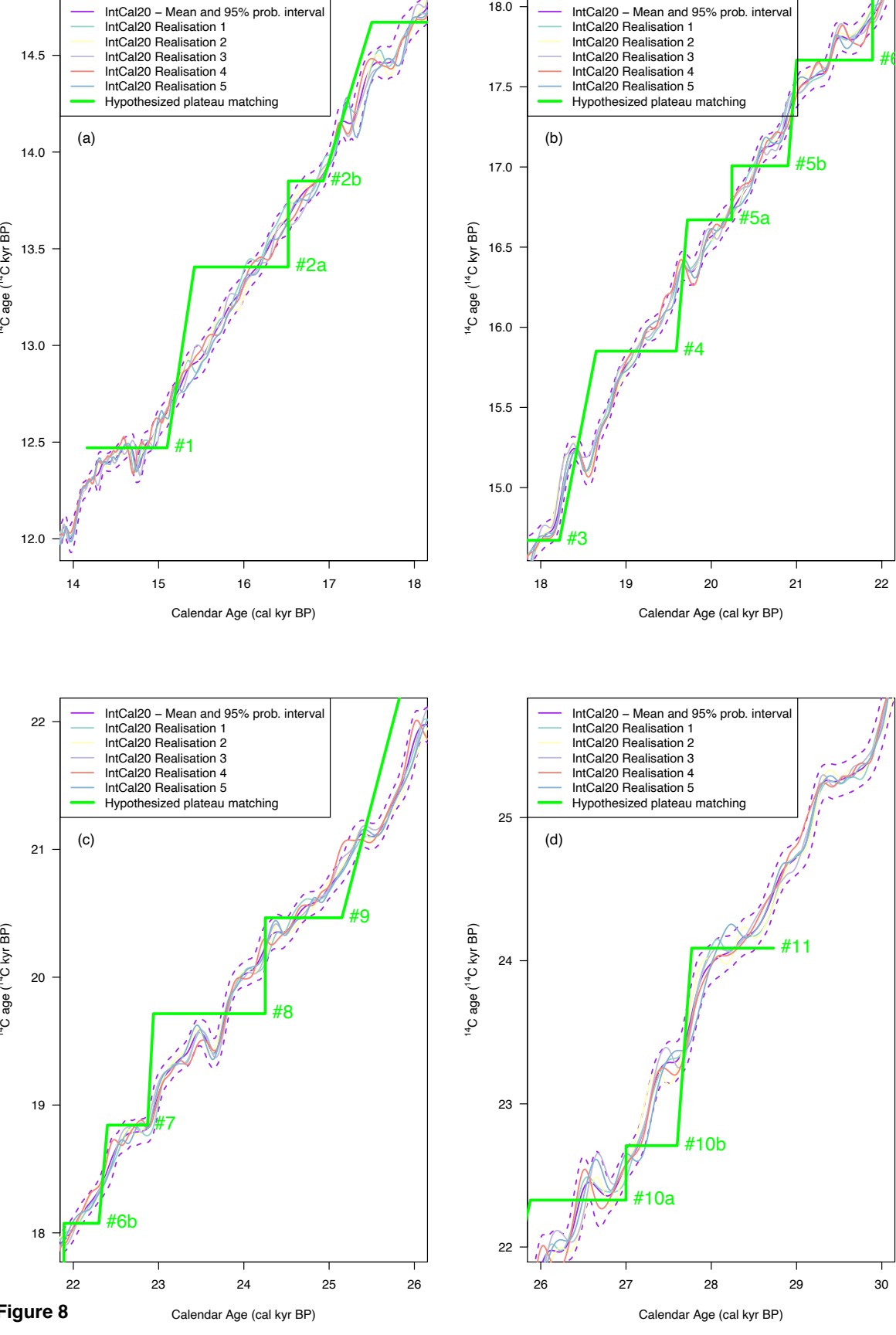

**Figure 8**