# Peer review of "On the tuning of plateaus in atmospheric and oceanic 14C records to derive calendar chronologies of deep-sea cores and records of 14C marine reservoir age changes"

_Climate of the Past, 2020_

## Community Comment (CC1)

**RESPONSE to the PREPRINT of E. Bard and T.J. HEATON (B&H)**

"On the tuning of plateaus in atmospheric and oceanic [14]C records to derive calendar chronologies of deep-sea cores and records of [14]C marine reservoir age changes"

CLIMATE OF THE PAST, DISCUSSIONS

Michael Sarnthein[1] and Pieter M. Grootes[2]

1 Institute of Geosciences, University of Kiel, 24118 Kiel, Germany

2 Institute of Ecosystem Research, University of Kiel, 24118 Kiel, Germany

**Abstract**

In response to an extended comment of Bard and Heaton (2021) (B&H) on the synthesis paper of Sarnthein et al. (2020) we counter their reservations both in the field of statistics and about the technique of [14]C plateau-tuning (PT), like in a manual, one-by-one, by means of telling lines of evidence. In particular, we single out the following points of view:

-- We show proof that results of PT of marine sediment records are hardly affected by bioturbational mixing and changes in foraminifera abundance, given the limitation of PT to cores with sedimentation rates >10 cm/kyr;

-- We illustrate the importance of initial guidelines of conventional stratigraphy to confine overall sedimentation rates as boundary condition and to derive alternative modes of PT for a whole suite of [14]C jumps and plateaus in a sediment record, [14]C structures to be compared to those of the paired atmospheric reference record of Lake Suigetsu.

-- Extended tests (Balmer & Sarnthein, 2016) revealed that changes in sedimentation rate *per*

*se* are unable to generate a complete suite of [14]C plateaus by now already defined in some 20

sediment cores and independently corroborated by various lines of local evidence.

-- Over the interval 10 - 15 cal. ka, the plateau structures of the Suigetsu atmospheric (atm) [14]C

record are clearly paired with well-defined tree ring- and floating tree ring-based [14]C structures (IntCal13; Adolphi et al., 2017). By comparison, we suggest that prior to 15 cal. ka the continuing [14]C fine structure of noisy Suigetsu with [14]C jumps and plateaus is by far more realistic than the admittedly smoothed [14]C trend of the Hulu speleothem and IntCal20, records that may also suffer from unknown but likely changes in the Hulu Dead Carbon Fraction (DCF).

-- By comparison to Holocene and late deglacial times, where PT may be constrained by tree ring records, glacial-to-early deglacial marine reservoir ages (MRA) can indeed be regarded as largely constant over time spans as long as $^{14}$C plateaus about 500-1000 yr. In turn, major MRA

changes are confined to more extended intervals of climate, sea ice cover, and ocean circulation similar to those of Heinrich events, Dansgaard Oeschger cycles, and their multiples.

-- Per analogy to the record of 10-15 cal. ka, overall $^{14}$C changes and shifts in the radiocarbon clock at 15-29 cal. ka are necessarily focused to inter-plateau times, just 18 % of the total time span as estimated by B&H. This concept indeed was first documented by means of PT.

-- We show that minor intra-plateau changes in MRA indeed exist, although they cannot be specified by our limited sampling resolution of ~50-150 yr. Careful inspection of the complete suite of plateaus in each core enabled us occasionally to identify distinct intra-plateau changes.

-- Concerns about low sampling density are unfounded. $^{14}$C structures in pelagic sediment records like boundaries of $^{14}$C plateaus, were not "under-constrained" by $^{14}$C ages but systematically documented by iterative sampling.

-- The box model discussion is scientifically correct. However, it only deals with *Pla,* the planktic $^{14}$C concentration of ocean surface waters, and not with MRA = (*Pla-Atm*).

In view of these findings the technique of PT cannot be regarded as 'result of inherent pitfalls'.

Rather PT is emerging as great opportunity to generate both a suite of narrow-standing and robust age tie points for marine sediment records and a record of short-term changes in MRA

and paleoceanography for last glacial-to-deglacial times in ocean sediment cores where independent high-resolution calendar age information is usually rare.

**1/ Introduction**

In the fall 2019, Edouard Bard and Tim Heaton (B&H) got access to the discussion version (opened in CPD at 25-10-2019) of our paper "Plateaus and jumps in the atmospheric radiocarbon record – Potential origin and value as global age markers for glacial-to-deglacial paleoceanography, a synthesis" (Sarnthein et al., 2020). A letter to the Editor of CPD on 31-01-

2020 stated they had written an extended comment to the paper but had submitted it as a research paper since it "includes substantial material of broad interest to the community using radiocarbon in marine sediments for geochronology and paleoceanography". This preprint is now subject to our discussion.

We thank B&H for their time and the efforts they spent for a synthesis to fight the 'misery' of

$^{14}$C plateau tuning (PT) that we, in turn, regard as accomplishment and 'blessing'. Their detailed arguments and reasoning nicely extend far down to basic processes that may control an atmospheric and sedimentary $^{14}$C record and are important when trying to specify a great number of major-to-minor potential pitfalls in our approach. Our response may help to clear various misconceptions and clarify crucial aspects of PT method, since all of us aim to find the best-possible technique to generate proper age control of ocean sediment records as now achieved by PT. In the following text-sections we try to summarize -- one by one -- the main concerns raised by B&H about PT in a brief initial statement. Subsequently, we give a detailed discussion and/or rebuttal of these concerns on the basis of our scientific reasoning and practice in PT.

**2/ Paleoclimatic and paleoceanographic perspective**

2.1/ *No independent constraint on sedimentation rate changes possibly creating $^{14}$C plateaus*

-- For each marine sediment core PT is constrained on the basis of stratigraphic guidelines derived from widely accepted marine stratigraphic techniques such as $\delta^{18}$O stratigraphy and/or sea surface temperature and climate records tuned to polar ice core stratigraphy (e.g.,

Voelker et al., 1988; Gebhardt et al., 2008; de la Fuente et al. 2015; Skinner et al., 2010;

Waelbroeck et al., 2019; Wang et al., 1999). PT does not abolish but refine conventional age control. It is a wiggle matching technique that compensates for the lack of lamination in most sediment cores by matching a suite of plateaus covering thousands of years.

-- PT requires permanent monitoring of changes in short-term sedimentation rate at plateau boundaries. As a rule, rates of change between one plateau and the next are low. If they exceed a factor of 1.5 the guideline for PT suggests detailed inspection of sediment parameters to trace the origin of the changing sedimentation regime.

-- Model tests of Balmer & Sarnthein (2016) show that sections with enhanced sedimentation rates *per se* are virtually unable to explain a long suite of $^{14}$C plateaus extending over sediment sections of up to tens of cm each.

Examples of independent PT confirmation are:

-- PS97-137 off Southern Chile (Küssner et al., 2020): A rough count of sediment laminations has fairly well confirmed the length of a PT-derived paired [14]C plateau for the LGM.

-- MD07-3088 off Central Chile (Küssner et al., 2020): Sedimentation rates and ages are confirmed by succession of four independent age values of ash layers.

-- MD07-3076, mid-ocean ridge South Atlantic (Balmer et al., 2016): Perfect match of PT-based

MRA with MRA deduced by correlation to Antarctic ice cores.

**2.2/** *Questionable claim for PT that Marine Reservoir Ages (MRA) have to be 'strictly constant'*

*over single plateau*s

This statement presents a misinterpretation. [14]C structures in pelagic sediment records were not "under-constrained" by [14]C ages but systematically documented by iterative sampling, in particular close to plateau boundaries.

-- As nicely shown by B&H in Fig. 3, the [14]C concentrations in the atmosphere have been, and still are, varying irregularly on centennial time scales. PT is based on a close comparison of the

*full suite* of planktic [14]C concentrations in a sediment core - averaged into one [14]C value for each low-slope section called 'plateau' - with their contemporaneous counterparts in the Suigetsu record of past atmospheric [14]C concentration.

-- PT gives (1) a suite of age tie points derived from translating U/Th-age based atmospheric plateau boundaries to the plankton-based [14]C record on a depth scale and (2) a record of average [14]C age differences between sedimented plankton and atmosphere, that is, of varying local Marine Reservoir Age (MRA). On the basis of conventional age control and internal [14]C

plateau structures PT, of course, needs to ponder the best-possible match between the full two curves for the glacial-to-deglacial period considered. Plotting the derived atmospheric age tie points against core depth allows for variable sedimentation rates. No further assumptions are needed!

-- Most [14]C plateaus cover time spans of 300-700 yr each, rarely reaching up to 1100 yr, in agreement with Fig. 3 of B&H. We see no problem in accepting that the ocean carbon cycle and

MRA have in most cases not been subject to major changes over these time spans and that changes were confined to short intervals in between. Major changes in MRA generally occur more rarely, at times of documented major change in ocean circulation such as in the context of

Dansgaard-Oeschger (DO) and Heinrich events.

-- Centennial-scale scatter of $^{14}$C values occurs *within* each single plateau defined as a $^{14}$C

scatter band with low or no slope with time/sediment depth. This scatter may indeed reflect real small-scale limited changes on decadal-centennial time scales besides sampling and measurement uncertainty. These minor variations have consciously been averaged for each plateau, since they can't be properly resolved by our sampling density and their small size makes them difficult to identify and separate from noise in the sediment and in the Suigetsu atmospheric $^{14}$C record. The offset between the averaged planktic and atmospheric plateau bands defines the MRA averaged over single plateaus as an approximation of the more complex and variable system.

-- In about a quarter of all cores one or more plateaus of the suite were distorted, in particular near DO event 1. Here major changes in MRA were indeed uncovered within the time span of a plateau by means of systematically testing (and rejecting) of various alternative models used to tune a complete suite of plateaus and careful comparison of the complete suite and internal structure of plateaus registered in the $^{14}$C record. In this way also "false" plateaus that B&H

warn against in line 105-114 were unmasked. In a number of cases (as listed in section 2.1) PT- based MRA values were confirmed by independent tephra-based ages and further lines of age control based on conventional correlation of paleoceanographic tracers that likewise are subject to multiple uncertainties.

-- Careful PT develops high-resolution age control and provides MRA as input to document sequential short-lived changes in the carbon cycle, often related to short-term ocean circulation changes as 'real' events more rarely occurring at millennial scales. Prior to 15 cal. ka such information was largely missing on the basis of conventional age control that assumes long-term MRA means. New developments in reservoir age modeling are also improving this situation (Heaton et al., 2020; Butzin et al., 2020), but generally lack the spatial resolution to match specific sediment cores.

**2.3/ *'Rung ladder' versus 'staircase' of age tie points - workable approximation vs. claim for**

*reality?***

-- In part, the different wording simply results from using more or less stretched Y-scales and differential intra-plateau slopes in a $^{14}$C age vs cal. age plot (B&H, Fig. 1). Different from this

Fig. 1, the slope of $^{14}$C scatter bands is not necessarily perfectly zero, as shown by tree ring- based records, e.g., near 10 cal. ka, and in 1st-derivative plots.

-- Figs. 1-3 of B&H form a useful base for discussion, although Fig. 1 unfortunately lacks a reconstruction of the crucial time section 10 to 14 ka. Here Suigetsu and tree ring data overlap giving the great opportunity to weigh Suigetsu $^{14}$C plateaus against a truly atmospheric and better-defined plateau record based on tree rings of IntCal13 (Sarnthein et al., 2020, Fig 2;

IntCal20 then not available yet; Reimer et al., 2020). The six plateaus defined for Suigetsu indeed match the seven of IntCal13, where Plateau 1a of Suigetsu may in reality depict two smaller ones in IntCal13. Vice versa, $^{14}$C structures based on Hulu Cave ages are matched by significantly more scatter. Accordingly, we trust the Suigetsu record more than IntCal20

dominated by Hulu as best possible indicator of atmospheric $^{14}$C beyond 14 cal. ka. This is corroborated by a comparison of Figs. 1, 2, and 3a and b of B&H that show the smoothed character of non-tree-ring IntCal20 relative to that of Suigetsu, thus a loss of fine structure needed for PT, as implied from the 10-to-14 ka intercomparison.

-- IntCal20 is the most reliable $^{14}$C calibration presently available, but integrates $^{14}$C ages from various carbon archives that in part filter the atmospheric $^{14}$C signal through surface water and groundwater reservoirs, hence dampen $^{14}$C fluctuations. In addition, local effects on the dead- carbon fraction by changes in rain fall, vegetation, soil cover, and $\delta^{13}$C that occur on millennial- scales may influence the Hulu cave $^{14}$C record (Kong et al., 2015). Also, coral- and foraminifera- based marine $^{14}$C records are subject to variable MRA assumptions. Prior to 15 cal. ka, floating tree ring sequences are rare. Following the principle "absence of evidence is no evidence of absence" the lack of structures in the IntCal20 curve is far more speculative and dubious than the suite of structures recorded in the only direct atmospheric $^{14}$C record of Suigetsu.

-- Since short-term plateau structures are distinct and common in the tree ring-based deglacial record 10-15 cal. ka (including plateau #1), moreover, all over the Holocene $^{14}$C record, though much shorter, any lack of pertinent atmospheric $^{14}$C structures over the preceding deglacial and peak glacial period would imply an assumption highly inconsistent, even if hypothesized by B&H

in section 3.1.

-- B&H are concerned about the large portion of atmospheric $^{14}$C drops at $^{14}$C jumps that encompass only 18% of the total time span studied. Thus, radiocarbon would have never behaved as a 'normal' geochronometer merely driven by regular radioactive decay, a claim our results of PT indeed confirm. (i) This concern overlooks considerable internal secondary $^{14}$C

variations within most plateaus, including possibly spurious drops and $^{14}$C reversals, that we necessarily eliminated by averaging the $^{14}$C age over a plateau.

-- (ii) Indeed, we may get accustomed to accept that centennial-scale jumps of atmospheric [14]C

age, similar to those found in the tree ring record for the last 15 cal. ka, paired with deep ocean circulation changes are real and form the rule rather than the exception. Plateaus during times of glacial-to-deglacial climate change appear far longer than most plateaus of the climatically 'quiet' Holocene, the last 8500 yr, which may reflect a different cause. We welcome B&Hs' notice of numerous, though unlikely abrupt rises in atmospheric [14]C. Also, we ourselves have discussed them internally already over years and regard them as valuable novel signals of short-term variations in ocean-atmospheric carbon exchange. As B&H say, processes controlling these partly fairly instantaneous processes are complex and difficult to model but highly challenging and worth to be traced by future studies.

**2.4/ B&H complain about a lack of a single figure illustrating all twenty marine [14]C records as**

*compared to their atmospheric [14]C curve used for calibration – Justification of hiatuses.*

In Sarnthein et al. (2020) we decided to avoid a repetition of basic data sets already published and documented elsewhere. Also, we were advised to reduce the length of our synthesis paper. Thus Figs. S2 did not intend to display the qualities of MRA derivation but the global spatio-temporal distribution of MRA results. Further below, Fig. 1 may serve as example to illustrate the technique of PT.

-- In most sediment cores the published lineups of records show that the alignment of the suite of planktic [14]C plateaus to the paired atmospheric [14]C calibration curve is fairly robust. In contrast to claims of B&H, most short-term changes in sedimentation rate between consecutive plateaus are low, hardly exceeding a factor of 1.5-2.0. Sporadic major shifts indeed mark climate tipping points (such as depicted in laminated sediments of SW Pacific core MD08-3180 near to Heinrich Event 2 and/or by reversals of the Denmark Street Overflow during early HS-1 in core PS2644).

-- Different from the claim of B&H, careful visual inspection of pertinent sediment sections and local proxy records have proven that the hiatuses contested by B&H also form distinct sediment unconformities, hence must not be discarded as artifact of PT. Several stratigraphic gaps are simply reflected by "mega-jumps" in the high-resolution [14]C record (e.g., cores

PS75/104-1, PS97/137-1, SO213-76, 17940 from South Pacific and South China Sea; synthesis

Figs. S2c, d). These lines of evidence were discussed at length in various source papers summarized in our synthesis paper. Though widely not appreciated by paleoceanographers, hiatuses appear to be a feature actually widespread at high-sedimentation rate sites in the deep sea – One may assume: The higher the rates the more extreme they may be subject to changes in depositional regime.

-- B&H are concerned about recent changes in our plateau assignment for two South Pacific cores. These changes are the result of a valuable discussion on alternative tuning modes ongoing after a first public display of data in CPD. Finally, we choose the mode better supported by various lines of sediment-based evidence.

**2.5/** B&H regret that *the focus of PT on $^{14}$C plateaus may leave large parts of $^{14}$C record unused in the process of matching a marine $^{14}$C record to the atm. $^{14}$C record of Suigetsu*. Conversely, Svetlik et al. (2019) just regard the high-slope parts of the $^{14}$C record as crucial for defining the absolute chronology. Under this topic B&H introduce to a discussion of basic objectives.

-- The concern of B&H is opposed to that discussed in TOPIC 2.2, where they calculated (and are concerned) that Suigetsu-based plateaus cover 82% of the total time. The remaining 18%, that are the $^{14}$C jumps in our Suigetsu record, may indeed confirm the conceptual model of Svetnik et al., hence form the crucial tie points for correlation to radioactive age control and to constrain past changes in MRA. Here it may be remembered that the distribution 82%/18% in part results from our choice of the length of the strictly horizontal plateaus and simplifies reality. Hence, we basically follow B&H in claiming that (most) "changes to MRA could only occur at plateau boundaries". Elsewhere changes may exist but cannot be resolved by the PT method. In summary, MRA derived from PT do form the best possible reconstruction available.

-- By now, the "special significance" required by B&H was only found for few plateaus of our $^{14}$C calibration curve (e.g., Plateau YD, 1, and lower 2a; see our synthesis Fig. 6). Here we propose a potential link to rare deglacial events of major ocean degassing similar to that on top of the YD and HE-1.

-- We agree with B&H that aligning the entire $^{14}$C record of a marine sediment core with that of the Suigetsu target curve, analogous to the wiggle matching technique for tree ring sequences, is the approach of our PT technique. Since our first paper of 2007 we stress the need that $^{14}$C records should be aligned as a whole with their shape, not just with piecewise constant or slightly different offsets within and between the plateaus, the key to our MRA estimates.

**2.6/** *Potential role of bioturbational mixing for $^{14}$C plateaus.*

$^{14}$C plateaus in marine sediments of course were checked for potential 'natural' changes in sedimentation rates by means of conventional stratigraphic markers (SST, $\delta^{18}$O, etc.; see discussion on Topic 2.1) always employed as initial stratigraphic guideline. Also, the impact of bioturbational mixing was not overlooked as potential factor influencing $^{14}$C plateaus (e.g., Küssner et al., 2018). Based on various lines of evidence and in view of the rule that PT is only applied to a complete suite of $^{14}$C plateaus each, that is >80% of a $^{14}$C record, bioturbation now was somewhat downgraded as potential factor for the origin of plateaus:

-- For PT, $^{14}$C ages were only measured on monospecific plankton samples, on species that continued in the region studied over glacial to interglacial times (except for mixed samples from ODP Site 1002D; Hughen et al., 2006). This is in contrast to troubling pioneer records of Duplessy et al. (1986) and Bard et al. (1987), who compared stable-isotope and $^{14}$C records of different species either characteristic of interglacial or of glacial times. Accordingly, bioturbational mixing resulted in a divergence of signals reaching up to 30 cm (i.e., ±15 cm) at deglacial times of abrupt climate change.

-- Trauth et al. (1997) gave first precise estimates of bioturbational mixing depth being clearly related to the local flux of nutrients / organic carbon. Low flux rates lead to mixing depths of 2-4 cm depth, high flux rates up to 8-12 cm. Thus, the position of plateau boundaries derived for a suite of plateaus in high-sedimentation rate cores may hardly present an artifact of differential bioturbational mixing, except for times of abrupt major change in nutrient flux.

-- We only applied PT to cores with average sedimentation rates of >10 cm/kyr. In many cores the rates exceed 20-40 cm/kyr and go up to >200 cm/kyr. The high rates contrast with most records $^{14}$C-dated in early days, where bioturbational mixing was particularly relevant at pelagic sedimentation rates of 2-5 cm/kyr.

-- In some cores crucial for paleoceanography (e.g., MD08-3180; PS97-137-1) sediment lamination is definitely precluding any role of bioturbational mixing for $^{14}$C plateaus.

-- PT of Core SHAK6K-05 provides a nice test case (Fig. 1) to compare short-term changes in the abundance of the planktic foraminifer *Globigerina bulloides* with the position and length of paired $^{14}$C plateaus (Ausin et al., 2019 and 2021). In contrast to conjectures of B&H, none of the twelve plateaus up to >15 cm long is linked to any abrupt change in species abundance. The plateaus just display, one-by-one, the reference suite of paired Suigetsu atmospheric plateaus. Local high sedimentation rates of 10-30 cm/kyr probably exceed by far the depth of
ongoing bioturbational mixing (up to ±7 cm; Ausin et al., 2019).

[Figure]

Figure 1. PT of planktic $^{14}C$ record from core SHAK06-5K with atmospheric (atm) $^{14}C$ record
from SUIGETSU. A) Abundance of planktic foraminifera *Globigerinoides bulloides*. B) Planktic
$^{14}C$ ages (blue dots) and oxygen isotope record ($\delta^{18}O$, red dotted line) measured on *G.
bulloides*, $^{14}C$ plateaus (black boxes), mean planktic $^{14}C$ age of each plateau (black numbers),
and MRA (blue numbers) (Ausín et al. [2019] and [2021] modified). $^{14}C$ plateau numbers in B
are deduced by visual correlation with C) atm. $^{14}C$ ages from Lake Suigetsu (blue dots),
corresponding atm. $^{14}C$ plateaus (black boxes), and age control points (cal. ka) plotted versus
U/Th-based model ages of Bronk Ramsey et al. [2012]. YD = Younger Dryas, B/A = Bølling-
Allerød, HS1 and HS2 = Heinrich Stadial 1 and 2, LGM = Last Glacial Maximum.

**2.7/** *Potential conflict of a sharp match of marine $^{14}$C-age plateaus with atmospheric plateaus with the general understanding of the carbon cycle* (see B&H lines 318 ff).

-- B&H employ numerical (box) model experiments showing that both damping and phasing effects in marine surface waters may be in conflict with the main assumption of synchroneity of atmospheric and ocean signals of $^{14}$C production. To meet this concern, we refer, like B&H, to the effect $^{14}$C bomb spike of the early 1960s. The discussion of B&H focuses on changes in the surface ocean and correctly describes the limitations of $^{14}$C variability in this reservoir. Yet they forget that MRA = (Pla - Atm) and that the large variability of atmospheric $^{14}$C, as seen in Miyake events and the bomb spike, means that also MRA can show variations much larger and more rapid than displayed by the box model.

**2.8/** *Plateaus were possibly linked to carbon cycle changes due to abrupt changes in meridional overturning circulation (MOC) like that at the end of the YD and HS1 events.*

-- As already outlined under Topic 2.5, we indeed found clues for the required "special significance" of few plateaus of our $^{14}$C calibration curve, that is, for Plateau YD, 1, and lower 2a, as was discussed in the context of our synthesis Fig. 6.

-- Ocean-induced changes of the atmospheric carbon inventory may indeed result in 'very minor' (in the context of paleoceanographers), up to decade-long regional delays of the $^{14}$C signal between different ocean regions with and without outgassing of deep-water $CO_2$. For outgassing regions B&H mention potential large effects with no delay. Elsewhere we have the same situation as discussed in 2.7. In view of the general broad uncertainty of conventional age estimates for marine sediment records (except for rare ages of marine ash layers; see the introduction of our synthesis) we agree with B&H that also an age control based on PT, though in our view far superior to conventional techniques, may hardly succeed to further constrain these regional age shifts covering less than 100 years.

**2.9/** We missed to give in our synthesis paper (not intended to serve as textbook) a reference to early reconstructions of MRA on the basis of marine volcanic ash layers by Bard (1988) and Bard et al. (1994), a lack revealed by B&H. In turn, we tried to stress that several PT-based age estimates are clearly reproduced by those ages of specific ash layers in the meantime widely-established (here by Siani et al. (2013).

**3.1 - 3.8/ Statistical perspective**

**3.1/** *Identification and correct match of 'true* $^{14}C$ *age plateaus' in sparsely sampled and noisy*

*marine records despite potential influence of MRA changes and bioturbation* (also see 2.3)

-- PT tuning is blamed for a lack of proof by any independent age control. This concern can be clearly rejected: In each marine sediment core PT has been constrained on the basis of initial stratigraphic guidelines derived from various conventional marine stratigraphic techniques such as $\delta^{18}O$ stratigraphy and/or sea surface temperature records tuned to polar ice core stratigraphy (Rae et al., 2014; Sarnthein & Grootes, 2007; Sarnthein et al., 2015).

-- In contrast to suggestions of B&H, the correct pairing of plateaus identified both in marine and atmospheric $^{14}C$ records formed a central topic of discussion for each basic description of

$^{14}C$ plateaus in a marine sediment core published so far (e.g., Sarnthein et al., 2015; Balmer et al., 2016). Again, we should emphasize that the full suite of plateau was considered instead of individual plateaus, as regularly suggested by B&H. In part, we handled the problem by frankly discussing alternative tuning modes (e.g., Küssner et al., 2020). In part, we admitted minor refinements in the mode of tuning of a suite of plateaus in papers published later-on, that is, as soon as additional lines of independent evidence were available.

**3.2/** *Assignment of plateaus in marine* $^{14}C$ *records may be biased by too noisy data sets of*

*marine sediment cores to a likewise noisy atmospheric record of Lake Suigetsu.*

-- B&H argue that the dead carbon fraction (DCF) of the Hulu Cave speleothem record has been stable around a low value of 480 ±55 $^{14}C$ yr and has not masked potential atmospheric plateaus in the Hulu $^{14}C$ record. Their conclusion is based on model tests and on a comparison of Hulu data for individual speleothems with tree-ring $^{14}C$ ages for the Allerød–Younger Dryas and Younger Dryas–Holocene transitions (Southon et al., 2012). Unfortunately, B&H ignore significant centennial-to-millennial-scale variations in a paired $\delta^{13}C$ record (by up to 7 per mil)

over the period 23-10 cal. ka. In part these changes result from changes in climate-controlled soil-organic matter either derived from C3 (more negative $\delta^{13}C$) or from C4 (more positive $\delta^{13}C$)

biomes prevailing in the formation of soil overlying the Hulu speleothem (Dorale & Liu, 2009;

Reimer et al., 2020). In part, however, the change is controlled by the intensity and thickness of differential soil formation that definitely is far more advanced at humid C3 than for semiarid C4

biomes. Hence $\delta^{13}$C changes in part form a rough proxy for the role of changing DCF for $^{14}$C

records in response to ($\delta^{18}$O-derived) climate change (Kong et al., 2005), variations not satisfyingly calibrated yet by Southon et al. (2012). Consequently, we do not see any need either "to smooth Suigetsu-based records" and/or to adjust the marine MRA to a marine average MRA of 480 yr used by the authors.

-- As pondered by B&H, a potential time-directional filtering of atmospheric $^{14}$C signals by the

Hulu speleothem may indeed be revealed by detailed analysis of the outlined $\delta^{13}$C record. It shows major shifts that lag paired $\delta^{18}$O shifts by ~650 to 700 yr (Kong et al., 2005).

-- Prior to 21 cal. ka, some $^{14}$C plateaus of Suigetsu (e.g., Plateau #8, though also reflected in the Hulu record; and #10b) indeed are more difficult to define than other plateaus due to analytical uncertainties amongst three different $^{14}$C laboratories.

-- B&H used the Bayesian spline statistical method for Suigetsu-based $^{14}$C ages for Figs. 1 and

3a. Fig. 1 expands the scale of the Y-axis by more than a factor 2 relative to the X-axis, which necessarily subdues the optical effect of $^{14}$C 'plateaus', but stresses the analytical uncertainties as compared to the 'green line' indicating the 15 plateaus listed in Table 1 of Sarnthein et al.

(2020). Nevertheless, Fig. 1 shows a decent general agreement between the green PT curve and the pink 95% Bayesian spline range. Prior to 20 cal. ka, minor differences may be due to slight revisions of calendar ages listed by Bronk Ramsey et al. (2020) as compared to the ages

Bronk Ramsey et al. published 2012, minor age shifts not properly assessed yet in the definitions given by Sarnthein et al. (2020).

-- In Fig. 2 B&H compare our record of $^{14}$C plateaus with the IntCal20 curve. The smoothed character of IntCal20 beyond 13.9 cal. ka has been generally acknowledged. As explained by

B&H in 3.8, this results from the IntCal aim to provide point wise summaries of the average of data from many different data sets, including carbonate-based marine data and speleothems, for the most reliable calibration of single radiocarbon results. This makes IntCal20 less suitable for exploring the fine structure of the atmospheric $^{14}$C record than the purely atmospheric

Suigetsu record.

**3.3/** *Low $^{14}$C sampling resolution for Lake Suigetsu and, even more so, for marine sediment*

*records as compared to annually resolved the tree ring record may lead to a misalignment of*

*marine records.*

-- The Suigetsu record shows an average sampling resolution of 20 yr at a highly resolved section near 14 cal. ka and one of 40-100 yr between 18 and 29 cal. ka. In turn, PT has been restricted by definition to plateaus longer than 300 yr, hence requires a minimum overall sampling resolution of marine sections better than 100-200 yr and one of 70-100 yr for sediment sections near plateau boundaries achieved by iterative sampling. Enhanced sampling resolution *within* a $^{14}$C plateau has turned out as redundant and waste of effort.

-- In harmony with $\delta^{18}$O and various other high-resolution stratigraphic records that are used as initial stratigraphic guideline, *PT only identifies the whole suites of $^{14}$C age plateaus*. On their basis single plateaus are specified. Also, different modes of PT are tested and discussed for each sediment core on the basis of age records and correlations used by conventional age control widely accepted by paleoceanographers. PT, however, is leading to a much higher resolution of age tie points, in part to minor modifications of conventional age assignments, and most important, to a suite of reasonable estimates of local MRA.

-- After all, the suite of glacial-to-deglacial Suigetsu plateaus can now be successfully reproduced in more than 20 sediment sections from all sectors of the ocean where sufficiently high sedimentation rates occur (some records are still in process of publication). A marked plateau of an Early Holocene tree ring record was reproduced in three neighboring cores from the northern Norwegian Sea.

**3.4/** *How to identify marine $^{14}$C-age plateaus within single cores in the context of a changing*

*and unknown sedimentation rate/ calendar age scale.*

This question has already been answered at length in preceding sections 3.3 and 2.2: PT

identifies suites of plateaus strictly on the basis of initial stratigraphic guidelines based on conventional chronostratigraphic records widely accepted amongst paleoceanographers. PT

has led to major refinements, in part also to modifications of conventional age control due to

$^{14}$C records with centennial-scale resolution.

**3.5/** *Definition of $^{14}$C plateau boundaries: Results of visual inspection versus estimates based on*

*calculation of the 1st derivative of $^{14}$C-age vs depth curve*

Comparative tests (Sarnthein et al., 2015; Fig. 2a) revealed good agreement of plateau boundaries deduced by visual inspection and by calculating the 1st derivative, though differential analytical errors have not been considered in these tests. Optimizing the kernel to the data set of the specific record requires a continued use of visual inspection. Evidence accumulated over the years suggests that $^{14}$C jumps reflected by maximums in the 1st derivative present the best-reproducible evidence to calculate the position of plateau boundaries in a marine sediment record (Ausin et al., 2021). To some degree this finding that forms a backbone of the novel age control induced by PT may indeed support the 'staircase' model of B&H as compared to our 'rung ladder' model, an item discussed in Section 2.3.

**3.6/** *Simulation tests to assess the ability to identify and tune $^{14}$C-age plateaus in the context of 'noisy and sparse' $^{14}$C data. To find the underlying calendar age scale in marine sediments B&H compare a tree ring-based IntCal20 record vs a 'pseudo-Suigetsu' atmospheric record and a pseudo-Cariaco marine record for the period 12-13.9 cal ka.*

**3.7/** *Second series of simulation tests to assess the ability to identify and tune $^{14}$C-age plateaus in pseudo-marine sediment records (similar to that of Cariaco) by tuning to a pseudo-atmospheric record.*

**3.8/** *Differences in the precision of reconstructing past atmospheric $^{14}$C levels -- A basic discussion of Lake Suigetsu and tree ring vs. IntCal20 records.*

A detailed answer to B&H discussion sections **3.6 - 3.8** is found in a companion contribution to this discussion, given by P.M. Grootes and M. Sarnthein, this volume of CPD.

*,,,,,,,,,,,,,,,,,,,,,,,,,,,,,,,,,,,,,,,,,,,,,,,,,,,,,,,,,,,,,,,,,,,,,,,,,,,,,,,,,,,,,,*

REFERENCES

Adolphi, F., Muscheler, R., Friedrich, M., Güttler, D., Wacker, L., Talamo, S., Kromer, B. Radiocarbon calibration uncertainties during the last deglaciation: Insight from now floating tree-ring chronologies. Quaternary Science Reviews, 170, 98-108. doi.org/10.1016/j.quascirev.2017.06.026, 2017

Ausin, B., Haghipour, N., Wacker, L., Voelker, A. H. L., Hodell, D., Magill, C., Looser N., Bernasconi S.M., Eglinton T.I. Radiocarbon age offsets between two surface dwelling planktonic foraminifera species during abrupt climate events in the SW Iberian margin. Paleoceanography and Paleoclimatology, 34, 63-78, doi: 10.1029/2018PA003490, 2019

Ausin, B., M. Sarnthein, N. Haghipour, 2021. Glacial-to-deglacial reservoir and ventilation ages on the southwest Iberian continental margin. – Quaternary Science Reviews, 255, 106818 (1-12 pp.)

S0277-3791(21)00025-1, doi.org/10.1016/j.quascirev.2021.106818

Balmer, S. and Sarnthein, M.: Planktic [14]C plateaus, a result of short-term sedimentation pulses? –
Radiocarbon, 58, DOI:10.1017/RDC.2016.100, 11 pp., 2016.

Balmer, S., Sarnthein, M., Mudelsee, M., and Grootes, P. M.: Refined modeling and 14C plateau
tuning reveal consistent patterns of glacial and deglacial 14C reservoir ages of surface waters in low-
latitude Atlantic. Paleoceanography, 31. doi:10.1002/2016PA002953, 2016.

Bard, E. Correction of accelerator mass spectrometry 14C ages measured on planktonic foraminifera:
Paleoceanographic implications. Paleoceanography, 3, 635-645. doi:10.1029/PA003i006p00635, 1988.

Bard, E., Arnold, M., Duprat, J., Moyes, J. and Duplessy, J.C. Reconstruction of the last deglaciation:
deconvolved records of δ18O profiles, micropaleontological variations and accelerator mass
spectrometric 14C dating. Climate Dynamics, 1, 101-112. doi:10.1007/BF01054479, 1987.

Bard, E., Arnold, M., Mangerud, M., Paterne, M., Labeyrie, L., Duprat, J., Mélières, M.A.,
Sonstegaard, E., Duplessy, J.C. The North Atlantic atmosphere-sea surface 14C gradient during the
Younger Dryas climatic event. Earth and Planetary Science Letters 126, 275-287. doi: 10.1016/0012-
821X(94)90112-0, 1994.

Bard, E. and Heaton, T.J.:  On the tuning of plateaus in atmospheric and oceanic [14]C records to derive
calendar chronologies of deep-sea cores and records of [14]C marine reservoir age changes. Climate of the
Past, Discussion, https://doi.org/10.5194/cp-2020-164, 2021.

Bronk Ramsey, C., Staff, R.A., Bryant, C.L., Brock, F., Kitagawa, H., van der Plicht, J., Schlolaut, G.,
Marshall, M.H., Brauer, A., Lamb, H.F., Payne, R.L., Tarasov, P.E., Haraguchi, T., Gotanda, K., Yonenobu,
H., Yokoyama, Y., Tada, R., Nakagawa, T., A Complete terrestrial radiocarbon record for 11.2 to 52.8 kyr
BP. Science 338:370-374, 2012.

Bronk Ramsey, C., Heaton, T.J., Schlolaut, G., Staff, R.A., Bryant, C.L., Brauer, A., Lamb,
H.F., Marshall, M.H., Nakagawa, T. Reanalysis of the atmospheric radiocarbon calibration
record from Lake Suigetsu, Japan. Radiocarbon 62, 989-999, doi: 10.1017/RDC.2020.18, 2020.

Butzin, M., Heaton, T.J., Köhler, P., and Lohmann, G.: A short note on marine reservoir age
simulations used in INTCAL20. Radiocarbon, oo, 1-7, DOI:10.1017/RDC.2020.9, 2020.

Cheng, H., Edwards, R.L., Southon, J., Matsumoto, K., Feinberg, J.M., Sinha, A., Zhou, W., Li, H., Li, X.
and Xu, Y. Atmospheric 14C/12C changes during the last glacial period from Hulu Cave. Science,
362(6420), 1293–1297, 2018.

de la Fuente, M., Skinner, L., Calvo, E., Pelejero, C., Cacho, I. Increased reservoir ages and poorly
ventilated deep waters inferred in the glacial Eastern Equatorial Pacific. Nat. Commun. 6:7420 doi:
10.1038/ncomms8420, 2015

Dorale, J.A. and Liu, Z. Limitations of Hendy test criteria in judging the paleoclimatic
suitability of speleothems and the need for replications. J. of Cave and Karst Studies, 71, 73-80.
2009

Duplessy, J-C., Arnold, M., Maurice, P., Bard, E., Duprat, J., Moyes, J.: Direct dating of the oxygen-
isotope record of the last deglaciation by [14]C accelerator mass spectrometry. Nature 320, 350-352, 1986.

Gebhardt, H., Sarnthein, M., Kiefer, T., Erlenkeuser, H., Schmieder, F., and Röhl, U.: Paleonutrient and
productivity records from the subarctic North Pacific for Pleistocene glacial terminations I to V.
Paleoceanography 23, PA4212, 1-21, doi:10.1029/2007PA001513, 2008.

Heaton, T.J., Köhler, P., Butzin, M., Bard, E., Reimer, R.W., Austin, W.E.N., Bronk Ramsey, C., Grootes,
P.M., Hughen, K.A., Kromer, B., Reimer, P.J., Adkins, J.F., Burke, A., Cook, M.S., Olsen, J., Skinner, L.C.
Marine20 - the marine radiocarbon age calibration curve (0-55,000 cal BP). Radiocarbon 62, 821-863,
DOI: 10.1017/RDC.2020.68, 2020a.

Hughen, K.A., Southon, J.A., Lehman, S.J., Bertrand, C.J.H., Turnbull, J. Marine-Derived [14]C Calibration
and activity record for the past 50,000 years updated from the Cariaco Basin, Quaternary Science
Reviews 25: 3216-3227, 2006.

Kong, X., Wang, Y., Wu, J., Cheng, H., Edwards, R.L., and Wang, X.: Complicated responses of
stalagmite $\delta^{13}$C to climate change during the last glaciation from Hulu Cave, Nanjing, China, Science in
China Ser. D Earth Sciences, 48, (12), 2174-2181, 2005.

Küssner, K., Sarnthein, M., Lamy, F., and Tiedemann, R.: High-resolution radiocarbon based age
records trace episodes of Zoophycos burrowing, 845 Marine Geology, 403, 48-56,
doi:10.1016/j.margeo.2018.04.01, 2018.

Küssner, K., Sarnthein, M., Michel, E., Mollenhauer, G., Ronge, T.A., Siani, G., and Tiedemann, R.:
Glacial to deglacial reservoir sages of surface waters in the southern South Pacific, PANGAEA,
https://doi.org/10.1594/PANGAEA.922671, 2020.

Rae, J., Sarnthein, M., Foster, G., Ridgwell, A., Grootes, P.M., and Elliott T.: Deep water formation in
the North Pacific and deglacial $CO_2$ rise, Paleoceanography, 29, doi:10.1002/2013PA002570, 645–667,
2014.

Reimer P.J., Austin W.E.N., Bard E., Bayliss A., Blackwell P.G., Bronk Ramsey C., Butzin M., Cheng H.,
Edwards R.L., Friedrich M., Grootes P.M., Guilderson T.P., Hajdas I., Heaton T.J., Hogg A.G., Hughen K.A.,
Kromer B., Manning S.W., Muscheler R., Palmer J.G., Pearson C., van der Plicht H., Reimer R.W., Richards
D., Scott E.M., Southon J.R., Turney C.S.M., Wacker L., Adophi F., Büntgen U., Capano M., Fahrni S.,
Fogtmann-Schulz A., Friedrich R., Kudsk S., Miyake F., Olsen J., Reinig F., Minoru Sakamoto M., Sookdeo
A., Talamo S. The IntCal20 Northern Hemisphere radiocarbon calibration curve (0-55 kcal BP).
Radiocarbon 62, 725-757, doi: 10.1017/RDC.2020.41, 2020.

Sarnthein, M., Grootes, P.M., Kennett, J.P., Nadeau, M.J. 14C Reservoir ages show deglacial changes
in ocean currents, in: Ocean Circulation: Mechanisms and Impacts, Geophysical Monograph Series 173,
edited by: Schmittner, A., Chiang, J., and Hemming, S., American Geophysical Union, Washington, DC,
175-197, doi: 10.1029/173GM0X, 2007.

Sarnthein, M., Balmer, S., Grootes, P.M., and Mudelsee, M.: Planktic and benthic 14C reservoir
ages for three ocean basins, calibrated by a suite of 14C plateaus in the glacial-to-deglacial Suigetsu
atmospheric 14C record, Radiocarbon, 57, 129-151, doi:10.2458/azu_rc.57.17916, 2015.

Sarnthein, M., Küssner, K., Grootes, P.M., Ausin, B., Eglinton, T., Muglia, J., Muscheler, R.,
Schlolaut, G. Plateaus and jumps in the atmospheric radiocarbon record – Potential origin and
value as global age markers for glacial-to-deglacial paleoceanography, a synthesis. Climate of the Past,
16, 2547–2571, doi: 10.5194/cp-16-2547-2020, 2020

Siani, G., Michel, E., De Pol-Holz, R., DeVries, T., Lamy, F., Carel, M., Isguder, G., Dewilde, F.,
Lourantou, A.: Carbon isotope records reveal precise timing of enhanced Southern Ocean upwelling
during the last deglaciation, Nature Communications, 4, 2758, 2013.

Skinner, L.C., Fallon, S., Waelbroeck, C., Michel, E., and Barker, S.: Ventilation of the deep Southern
Ocean and deglacial $CO_2$ rise, Science, 328, 1147–1151, 2010.

Southon, J., Noronha, A.L., Cheng, H, Edwards, R.L., and Wang, Y.: A high-resolution record of
atmospheric [14]C based on Hulu Cave speleothem H82, Quaternary Science Reviews, 33:32-41, 2012.

Svetlik, I., Jull, A., Molnár, M., Povinec, P., Kolář, T., Demján, P., Pachnerova Brabcova K, Brychova V,
Dreslerová D, Rybníček M, Simek, P. The Best possible time resolution: How precise could a radiocarbon
dating method be? Radiocarbon, 61(6), 1729-1740. doi:10.1017/RDC.2019.134, 2019

Trauth, M.H., Sarnthein, M. and M. Arnold, M.: Bioturbational mixing depth and carbon flux at the
seafloor. - Paleoceanography, 12, 517-526., 1997.

Voelker, A., Sarnthein, M., Grootes, P.M., Erlenkeuser, H., Laj, C., Mazaud,, A., Nadeau, M., and
Schleicher, M.: Correlation of marine [14]C ages from the Nordic Seas with the GISP2 Isotope record:
Implications for radiocarbon calibration beyond 25 kyr - In: Proc. 16th Internat. [14]C Conference;
Radiocarbon, 40, 517-534, 1988.

Waelbroeck, C. et al., Consistently dated Atlantic sediment cores over the last 40 thousand years.
Scientific Data 6:165, https://doi.org/10.1038/s41597-019-0173-8. 2019

Wang, L.J., Sarnthein, M., Erlenkeuser, H., Grimalt, J., Grootes, P., Heilig, S., Ivanova, E., Kienast, M.,
Pelejero, C., and Pflaumann, U.: East Asian monsoon climate during the late Pleistocene: High-resolution
sediment records from the South China Sea, Marine Geology, 156, 245-284, 1999.

,,,,,,,,,,,,,,,,,,,,,,,,,,,,,,,,,,,,,,,,,,,,,,,,,,,,,,,,,,,,,,,,,,,,,,,,,,,,,,,,,,,,,,,,,,,,

---

## Community Comment (CC3)

**Comments on**
**Edouard Bard and Timothy J. Heaton (B&H)**
**On the tuning of plateaus in atmospheric and oceanic [14]C records to derive calendar**
**chronologies of deep-sea cores and records of [14]C marine reservoir age changes.**
**Climate of the Past. Discussions**

Pieter M. Grootes[1] and Michael Sarnthein[2],

Institute for Eco System Research[1] and Institute of Geosciences[2],

Christian Albrecht's University Kiel, 24118 Kiel, Germany

In the fall 2019, Edouard Bard and Tim Heaton (B&H) got access to the discussion version (opened in CPD at 25-10-2019) of our paper "Plateaus and jumps in the atmospheric radiocarbon record – Potential origin and value as global age markers for glacial-to-deglacial paleoceanography, a synthesis" (Sarnthein et al., Clim. Past 16, 2020 (SA2020)). A letter to the Editor of CPD on 31-01-2020 stated they had written an extended comment to the paper but had submitted it as a research paper since it "*includes substantial material of broad interest to the community using radiocarbon in marine sediments for geochronology and paleoceanography*". This comment is now subject of our discussion. Its aim is to demonstrate that Plateau Tuning (PT) is fraught with problems and should not be used.

We thank B&H for the time and efforts they spent formulating the problems they see with the technique of [14]C plateau tuning. Their detailed arguments and reasoning extend far down to basic processes that may control an atmospheric and sedimentary [14]C record and thereby provide a base for a factual discussion of PT. Both basics and details are important when evaluating potential major-to-minor pitfalls of PT but can rarely be discussed at meetings or workshops (B&H *lines 65-69*). As stated in their letter the paper 'includes substantial material of broad interest' and many of their potential pitfalls are worth considering. Our response may help to clear various misconceptions and further explain crucial aspects of the PT method. This is important since all of us aim to find the best-possible techniques to generate proper age control of ocean sediment records and to make an optimum use of the wealth of environmental information they contain. Below we summarize two points where B&H misconstrued PT and we advocate a different conclusion from their Fig. 3 , 5, and 6.  Then we address their specific chapters and text.

Summary

Many of the 17 objections raised by B&H are based on two simple points:

(1) *The difficulty of reliably identifying a single $^{14}C$-concentration plateau in a noisy $^{14}C$ sediment record and then finding its correct partner in the noisy record of atmospheric $^{14}C$ concentrations*

This is the subject of eight objections (2.1; 2.2; 2.5; 2.6; 3.3; 3.4 straight and part of 2.3 and 3.1). These objections are based on *lines 40-41*, the first lines of the B&H Introduction: In line 40 the term *'a suite of'* is missing between *'tuning'* and *'hypothesized'* and in line 41 *'those that'* should be replaced by *'a suite of plateaus'*.

The problem of how to identify a plateau has been extensively considered in the development of PT. Sarnthein et al., 2007 clearly mention they *'identified a reference suite of prominent atmospheric $^{14}C$ "plateaus" '*, based at the time on Cariaco ODP 1002 and on U/Th dated corals and Bahama speleothem, and the identification of *'analogous series of $^{14}C$ plateaus in several other marine sediment cores …'*. This emphasis on suites of plateaus and suites of tie-points has been part of every PT paper including SA 2020, discussed by B&H. As B&H point out: Identifying a single plateau is very hard. Further details on meeting this set of questions are given in the companion response text of Sarnthein & Grootes (S&G).

(2) *The focus on $^{14}C$ concentration changes in the surface ocean (pla = planktic $^{14}C$ concentration) instead of on marine reservoir age (MRA = pla- Atm, where Atm is the contemporaneous atmospheric $^{14}C$ concentration).*
This leads to objections 2.3; 2.7; 2.8; 3.1 and 3.2 that basically repeats 2.3.

In a simple carbon cycle box-model (e.g. Siegenthaler et al., 1980) with a deep ocean that contains about 60 times more carbon and 50 times more radiocarbon than the atmosphere, most of the variability in $^{14}C$ concentration will be in the atmosphere and in its closely-connected thin ocean-atmosphere exchange layer. The focus of B&H on the surface ocean is logical, because it is easily accessible and its plankton provides our paleoenvironmental record, but MRA is the difference in $^{14}C$ concentration between atmosphere and surface ocean (*pla-Atm*). B&H use the Bard et al., 1997, 12-box model to calculate an attenuated and somewhat delayed response of the surface ocean to, especially

rapid, atmospheric [14]C changes. This leads them to reject the PT derived MRA changes as "too large, too frequent, too abrupt". Their modelling addresses, however, only one facet of MRA, and a strongly attenuated *pla* signal will generate an MRA (= *pla-Atm*) signal with little attenuation. This is borne out by the effects on MRA of the [14]C bomb spike and Miyake events mentioned in their text. A box model, moreover, does not consider local variations in near-surface ocean mixing and ocean-atmosphere exchange, that can lead locally to large and rapid changes in *pla* and thus MRA for an unchanged atmosphere.

Fig. 3a shows the translation of the PT suite of plateaus, defined by SA2020, in the [14]C-age/ calendar-age domain, into the $\Delta^{14}$C/calendar age domain and compares the translated plateau step curve with the Bayesian-spline generated Suigetsu atmospheric record.  The statistically sound zig-zags of the  Bayesian Suigetsu curve reveal a generally satisfactory agreement with the (green) plateaus (sections of (faster) decreasing $\Delta^{14}$C vs. decreasing cal. age) and jumps (sections of slower decrease or increase), despite the fact that the plateaus defined by Sarnthein et al. (2015 and 2020) were based on the 2012 Suigetsu data and did not consider the most recent age corrections of Bronk Ramsey et al., (2020). The zig-zag curve does not pin the plateau slope to zero and offers another look at the position of inflection points, so far defined by the beginning and end of zero-slope plateaus. This will be further explored for PT.

The modelling exercise of 3.6 and 3.7 demonstrates how the statistical scatter of sampling and imperfect measurements may distort and mask underlying real signals. Yet, contrary to the stated conclusion, it offers hope in showing that at least two out of the three 'plateau' features of the underlying short record can indeed be found in the modelled examples. It also makes a clear point that such identification of the 'true' fine structure of a [14]C record is a serious research project requiring consideration of a broad range of conventional age tie points and oceanographic information and a long sequence of plateaus in order to produce reliable results. And, even then, it still needs to be checked against other independent records.

In the following we first comment on B&H Chapter 1. and then address the objections of Chapters 2 and 3.

**1/ Introduction**

*Line 40-41*: A small but crucial element is missing in the introduction of PT. The technique tunes **a suite** of $^{14}$C age plateaus in a sediment record against **a suite** of plateaus defined in the atmosphere (as represented by the Suigetsu record). Such a suite covers usually 5000 to 10 000 years and is crucially needed to define a proper match and deal with local disturbances that may distort, eliminate, or mimic single plateaus.

*Line 58*: contains the quote *'ocean models still poorly reproduce" their reconstruction ..."'*

The quote is correct but the paragraph misrepresents the SA2020 paper. Ventilation ages are obtained via the difference in $^{14}$C concentration between benthic foraminifera and the atmosphere (*Be-Atm*). The deep ocean contains 60 times more C and 50 times more $^{14}$C than the atmosphere, so in a carbon cycle mass balance model the $^{14}$C concentration in the atmosphere fluctuates about 50 times more than in the deep ocean. Most of the variability in *Be-Atm* will thus be in the atmosphere. The changes in *Be* related to the deep ocean circulation and the release or drawdown of $CO_2$ will not be as large as the ventilation age variability appears to suggest. Core specific results often deviate from the predictions of global- or basin-wide ocean models, but this is often not due to bad data or a bad model but rather to a model not tuned to the specific core site conditions. One may say that progress in model development is prompted by boundary conditions and results not properly covered by existing models (Lohmann et al., 2020).

*Lines 64-65*: Umling and Thunell (2017) show in their Fig. 4 a close agreement between time scales obtained by tuning to Greenland ice cores, Hulu speleothems and $^{14}$C plateaus using PT. Discussed in 2.4 *lines 214-223*

*Line 72*: *'the compiled records based on PT leads to perplexing outcomes (no coherence with either production changes or with ocean modelling results).'*

The wording is suggestive and misleading. Chances are small for finding clear correlations between $^{14}$C fluctuations and $^{14}$C production (except for Laschamp and Mono Lake) and/or ocean circulation/climate. A few correlations appear to be emerging, but we are still far away from coherence in the poorly known and highly variable deglacial.

A more neutral formulation could have been: *'The compiled records do not yet allow a clear attribution of the observed $^{14}$C variations to $^{14}$C production or carbon cycle changes'* as pointed out by the authors themselves.

**2/ Paleoclimatic and paleoceanographic perspective**

**2.1** "*Non-laminated ocean sediments are not suitable for wiggle matching*".

The base for this objection lies in the first introductory sentence (*lines 40-41*). PT uses indeed wiggle matching in a non-laminated sediment. The pattern of yearly ring widths of the tree is replaced by the pattern of $^{14}$C concentration variations along a sediment core. Clearly a single plateau cannot yield a pattern. A suite of plateaus, correlated on the strength of its pattern, provides an 'elastic' time scale to the sediment depth record. The stratigraphic and sedimentological procedures and correlations used are presented in the answer to 2.1 by S&G and need not be repeated here.

**2.2** '*PT assumes that the marine $^{14}$C reservoir age is strictly constant during the age plateaus*' and since this cannot be, '*PT structures are severely under-constrained in $^{14}$C and calendar ages*'

This objection has the same root as 2.1, namely that single plateaus are correlated. Indeed, single plateaus **can not** be reliably correlated, there are too many; the whole suite of plateaus is required! We use B&H Fig. 3 for a brief explanation of PT to counter the objections. Fig 3a shows Suigetsu $^{14}$C concentrations as $\Delta^{14}$C (a surrogate for the atmosphere), varying irregularly from 14 to 30 cal kyr BP on centennial time scales. These concentrations can be converted into a curve of apparent atmospheric $^{14}$C ages against calendar age such as Fig.1 of B&H and of SA2020. The same can be done for $^{14}$C concentrations of planktic carbonate in a sediment core plotted against depth. If an approximate time scale can be obtained for the sediment core based on conventional stratigraphic techniques and correlations (see also comment S&G), then an optimum match between local sediment and global atmosphere can be sought for the **full** $^{14}$C age-pattern of irregularly alternating low slopes ('plateaus') and high slopes ('jumps').

**Note**: There is no assumption for MRA estimates! MRA = $(pla - Atm)$ for each plateau pair.

The one crucial assumption is the one also used in IntCal to extract atmospheric $^{14}$C concentrations from marine carbonate $^{14}$C records, namely that there is a close correspondence between $^{14}$C in the atmosphere and in the top layer of the local ocean.

The objections listed in *lines 100-105* are thus based on misunderstanding.

The problems listed in *lines 105-114* are real and are routinely encountered in PT. Abrupt changes in local oceanography or sediment disturbance can disturb, destroy, or mimic

a plateau. Matching suites of plateaus will bring this out, although disturbance will make it more difficult to find the right match amongst alternative tuning possibilities. Plotting the planktic $^{14}$C concentrations (or ages) against depth means the match will transfer the standard atmospheric time scale to the sediment core. This often reveals variable sedimentation rates. Checking whether those rates are physically realistic is one of PT's many quality controls, used to decide between alternative tunings. Matching the sediment record $^{14}$C suite with the atmospheric master $^{14}$C suite (Suigetsu) provides: 1). A series of time markers in the sediment at the inflection points of the wiggly curve (approximated by line segments as seen in Fig. 3a and Fig. 1) and 2). A series of difference values *(pla − Atm)* between the $^{14}$C ages of matched plateaus in core and atmosphere, the estimates of MRA.

[Figure]

Figure 3

Fig. 3a of B&H shows the translation into the $\Delta^{14}$C/calendar age domain of the PT suite of plateaus, defined by SA2020 on basis of Bronk Ramsey et al., 2012 in the $^{14}$C-age/ calendar-age domain. The pattern of horizontal plateaus and steep jumps transforms into the green curve with sections of steeper decrease in $\Delta^{14}$C (the former plateaus) and sections with less decrease or even an increase in $\Delta^{14}$C (former jumps) with time going forward. The green pattern compares reasonably well with the pink Bayesian spline of the 2020 version of Suigetsu $\Delta^{14}$C (Bronk Ramsey et al., 2020). Fig.3b shows the same for IntCal20. At >20 cal kyr BP, conversion of cal. ages of hypothesized Suigetsu plateaus to recent age estimates of Bronk Ramsey et al. (2020) will further improve the alignment.

[Figure]

Figure 1. SA2020, shows in blue the 2012- Suigetsu $^{14}$C data with plateaus >300 cal. yrs long defined for low-slope sections and, for comparison in red, the same for the Hulu record, shifted by +3000 $^{14}$C yrs.

[Figure]

Figure 2. SA2020, shows the 2012 Suigetsu $^{14}$C values (700-yr offset) and IntCal13-tree-ring values with low-slope sections >300 cal. yrs indicated by plateau bands.

**2.3** This section discusses two key objections. First *'weak evidence for many of these hypothesized $^{14}$C-age plateaus' (lines 132-133)* and second their physical impossibility *'in their model, $^{14}$C ages must jump, often instantaneously, from one plateau to the next' (line 135)*

B&H Fig. 1–3 provide in combination with Fig.1-2 and Table 1 of SA2020 a valuable basis for the discussion of these objections. Fig. 3 shows a nice picture of fluctuating

atmospheric $^{14}$C concentrations, expressed as $\Delta^{14}$C over the period 14-30 cal kyr BP, based on the primary 2020 Suigetsu $^{14}$C data points of Bronk Ramsey et al., 2020, (Fig. 3a) and on the latest IntCal20 synthesis (Fig. 3b). The smoothed character of IntCal20 compared with Suigetsu is obvious. Unfortunately, B&H did not reconstruct the full record up to 10 ka, as presented in Figs. 1 by SA2020. Their more detailed Fig. 2 highlights the missing section 10 to 14 cal kyr BP, where Suigetsu and tree-ring-based IntCal 13 overlap (IntCal20 was not available yet). As can be seen, the tree ring pattern of alternating sections with more and with less slope is matched, though with significantly more scatter, by the Suigetsu data. Six 'plateaus' defined in Suigetsu match seven IntCal13 'plateaus' whereby plateau 1a of Suigetsu combines two short, close-together 'plateaus' of IntCal13. This similarity provides a basis for trusting Suigetsu as a good indicator of atmospheric $^{14}$C beyond 14 cal kyr BP and, thereby, confirms the smoothed character of IntCal20 beyond the 14 kyr BP tree ring calibration (further discussion in S&G). IntCal20 provides the most reliable $^{14}$C calibration we currently have, but for certain specific purposes like PT the single atmospheric Suigetsu record, although stand-alone and noisy, will be more suitable. As such our use of Suigetsu follows the encouragement in the conclusions of IntCal13.

Fig. 1 and 2 of B&H show $^{14}$C age versus calendar age for Suigetsu and IntCal20 in the usual IntCal20 format. As stated by B&H in *lines 129-131* the evidence for many of the older plateaus in IntCal20 Fig. 2 is dubious. This is not unexpected considering the smoothed character of IntCal20. The following sentence *'They are not replicated in either our statistically- robust Lake Suigetsu-only curve (Fig. 1) or in the IntCal20 curve (Fig.2).'* is, however, not supported by Fig. 1. Inspecting the green plateaus, selected from the noisy Suigetsu data set by visual inspection as well as by a first-derivative kernel technique, shows that 13 of the plateaus correspond with calendar age broadening in the Bayesian-spline-generated pink band; solely plateaus 5b and 10b are not reflected in the pink band, although the local scatter of Suigetsu data makes their selection understandable.  Some of the discrepancies between the 'green' Suigetsu plateau record of SA2020 and the Bayesian spline may reflect that the plateau record has not incorporated yet the recent age corrections of Bronk Ramsey et al. (2020).

B&H have spent considerable effort to document in the second section (*lines 134-145*) the physical impossibility of the PT model. They conclude that the strictly horizontal plateaus cover 82 % of the time between 14 and 29 cal kyr BP while in the jumps *'during the remaining*

*18 % of the time, the radiocarbon clock was running almost 5 times too fast.'* Although this sounds ridiculous and physically impossible, it is not as crazy as it seems and it points to one of the big problems in radiocarbon dating. We do not know the initial concentration of the sample/atmosphere. Since the radiocarbon-decay clock always runs at a steady pace, the varying measured radiocarbon concentrations indicate either varying sample ages or varying initial radiocarbon concentrations. B&H mention (*lines 151-155*) that during the Myake AD 774-75 event (Miyake et al., 2012) the atmospheric $^{14}$C concentration increased by about 1.2 %. This provides an example of the radiocarbon 'clock' 'running' forward almost 100 years in that single year. The increase was followed by delayed decline in atmospheric $^{14}$C attributed in IntCal20 to the time needed to mix the $^{14}$C into the ocean which produces a saw-tooth $^{14}$C concentration pattern.

The green plateau pattern of B&H Fig. 1 translated in the $^{14}$C concentration domain produces a saw-tooth pattern that fits the fluctuating Suigetsu atmospheric $^{14}$C concentrations in Fig. 3a reasonably well. It is, however, clear that the assumption of plateau slope zero, i.e. constant $^{14}$C age with changing calendar age, is a simplifying approximation; some of the downslope sections in Fig. 3a are steeper, some less steep. Close comparison of the green curve with the pink Bayesian-spline envelope in Fig. 3a also indicates that deciding on the length of a plateau may be even more difficult than selecting its position. Mainly the statistically more uncertain section beyond ~22 cal kyr BP shows deviations with 'undershoots' for plateaus 11, 10a, and 8 while plateau 10 b is not seen in the pink band. Figures 1-3 of B&H thus demonstrate that IntCal20 is ill-suited to investigate $^{14}$C fine-structure in environmental records while a Bayesian spline representation of Suigetsu preserves fine structure and yields promising results for PT studies.

**2.4**. B&H object that primary sediment records and their tuning to the atmosphere are not provided in the synthesis paper Sarnthein et al., 2020. The tuning is bad and results in large changes in sedimentation rates and hiatuses.

B&H complain it was *'necessary to dig into the literature to see the marine core records'* (*line 169*) and provide a list of the records they recovered. These data have been available, some as early as 2007, and it is disappointing that B&H access them only now, considering the serious objections they mention. It would have been great to have a scientific in-depth discussion earlier, preferably in the discussion phase of the synthesis paper, or

earlier. Although B&H don't provide examples of their tuning objections, a major problem appears to be the fluctuating sedimentation rates that result from the introduction of multiple time markers, i.e. plateau boundaries, in the sediment records. Replacing 'constant sedimentation rates' that result from a lack of data with a pattern of significantly varying sedimentation rates can be disturbing but may also provide rewarding insights. Sedimentary and stratigraphic aspects of tuning these records are further discussed by S&G.

The reference in *lines 64-65*, and *212-224* to Umling and Thunell., 2017, using PT refers to 'puzzling results' but fails to mention the close agreement between the ages they obtained by three independent techniques, including PT, and shown in their Fig. 4. Their Fig. 3c suggests a naming problem for the youngest, 12.8-13.1 cal kyr BP, of their suite of five matched plateaus. This is the 'no name' plateau of SA2020, instead of the 'YD' plateau, which reduces the hiatus from 1200 to ~600 years. Its timing roughly corresponds to the Inter-Allerød-Cold-Period in Greenland/Northwest Europe.

Searching for the best match between the plateau suites of $^{14}$C concentrations in sediment and atmosphere is a process of balancing the physics of ocean, atmosphere, and sediment by exploring different options. It has yielded surprising and sometimes controversial results, including hiatuses, and certainly warrants a critical discussion such as advocated by B&H. The collection of some twenty long records, though each individually on statistically shaky grounds, is by now providing consistent patterns indicating real past oceanic and ocean-atmosphere dynamics.

**2.5**. PT focus is on plateaus instead of '*matching the entire $^{14}$C record with the target curve*'

This objection has the same root as 2.1 and 2.2, namely the misconception that PT correlates single plateaus, and is based on the first introductory sentence (*lines 40-41*). Thus we fully agree with B&H that the full records should be compared. The choice of a horizontal (=zero) plateau slope is simply made for the $^{14}$C scatter bands with insufficient data to define a real slope. The pattern of horizontal and sloping line segments in the $^{14}$C age/cal age domain facilitates visual comparison but has as a consequence that MRA values, obtained from the difference between oceanic and atmospheric $^{14}$C plateaus, are constant over a plateau and change in steps from one plateau to another. In reality MRA variability will be more distributed. As made clear in the discussion of 2.3, the pattern of plateaus and steep(er) connections translates well into the saw-tooth curve of $\Delta^{14}$C vs calendar age of Fig. 3a.

Bayesian spline fits, like shown in Fig. 3a, may offer PT another statistical method of data evaluation and a way to approximate the real slope of non-zero-slope plateaus. Approaching the real world with subtle and varying differences in slope between oceanic and atmospheric records will, however, remain a lofty goal that will require (impossibly) denser data.

**2.6**. B&H warn that marine age plateaus may result from slumps, sedimentation rate changes, and bioturbation around abundance maxima.

This objection has the same root as 2.1, 2.2, and 2.5. Again, the incomplete description of PT in the first introductory sentence (*lines 40-41*) is the base for admonishing SA2020, that correlating single plateaus is dangerous. Each of the processes mentioned will indeed disturb the regular age-depth pattern in a sediment core and might cause an age plateau. For this reason PT always uses a suite of plateaus so the loss or addition of a single plateau will be evident. The effect on PT of bioturbation in combination with a varying abundance of foraminifers in the sediment is discussed in our companion discussion text S&G.

**2.7**. Ocean modelling considerations argue against the PT results.

As already stated in the comment on *line 58* of the introduction, most of the carbon and $^{14}$C reside in the deep ocean. A mass balance model of the carbon cycle thus requires that C and $^{14}$C changes in the deep ocean are relatively minor and that most of the variability is located near the very surface of the ocean and in the atmosphere.

The modelling and discussion of *lines 284-336* of B&H focus on the $^{14}$C concentrations of surface ocean reservoirs (*pla*) of complete ocean basins each in response to an atmospheric $^{14}$C production signal. The 12-box model of Bard et al., 1997, introduced in lines 284-294, calculated the changes in $^{14}$C distribution over the various reservoirs of the carbon cycle in response to a reduction of the global thermohaline circulation from today's 20 Sverdrup (Sv) to a postulated glacial 10 Sv (Fig. 4, $\Delta^{14}$C values / $\Delta^{14}$C in brackets). In both cases the atmosphere is the benchmark and its $\Delta^{14}$C has been set to zero.  In reality the $\Delta^{14}$C of the 10 Sv atmosphere had increased by 35 ‰ relative to the standard atmosphere of 20 Sv.

 The modelled graphs 4a,b of B&H quantify the attenuation of sinusoidal atmospheric $^{14}$C signals of different frequency entering the surface ocean (4a) and a 500-year atmospheric sinusoidal signal moving down into the ocean (4b). The graphs show that the response of surface ocean $^{14}$C concentration *pla* to $^{14}$C concentration changes in *Atm* will be attenuated

and depend on the rate of change of the atmospheric $^{14}$C signal as well as on the location, here the ocean basin. B&H do not consider MRA, the difference between troposphere and surface ocean in Fig. 4b, only *pla*. For a large change in *Atm*, due to $^{14}$C production or remote ocean outgassing, the calculated local change in *pla* may be small, but the change in MRA may be large and, for strong attenuation, approach in shape and size the *Atm* signal. Fig. 4 thus does not prove that large and rapid MRA signals are physically unrealistic.

The discussion of the bomb spike in *lines 318-336* is correct but provides another example of the focus on *pla* instead of MRA = (*pla-Atm*). The bomb $^{14}$C signal and its penetration into the ocean have been well documented. Atmospheric $\Delta^{14}$C values were -20‰ in 1954, +400‰ in 1962 and ~+900‰ in August 1963, all relative to the standard atmosphere. If surface ocean and atmosphere were in equilibrium in 1954 with a MRA of 400 years, then 1954 $\Delta^{14}$C *(pla-Atm)* was -48 ‰. If *pla* did not change significantly from 1954 to 1963, then (*pla-Atm*) was -468 ‰ in 1962 and -968 ‰ in August 1963, $\Delta^{14}$C values all based on the standard atmosphere. Converting these $\Delta^{14}$C differences to the classical MRA: 'time needed for the atmospheric $^{14}$C concentration to decay to the $^{14}$C concentration of the surface ocean' that is anchor the system in the atmosphere of the time so $\Delta^{14}$C$_{atm}$ is per definition zero, we obtain a MRA of 3270 $^{14}$C years in 1962 and one of 5720 years in August 1963. This MRA variability, following definitions, is of course the result of our bomb $^{14}$C input into the atmosphere. Yet, for a Miyake event of 12‰ the *(pla-Atm)* will change by that much and MRA will increase by about 100 years over one year in a natural system.

Far more than for OGCMs, a weak point of the box model approach to probe the physical feasibility of MRA records obtained for individual sediment cores is its limited spatial resolution, limited to ocean basins. Changes in *pla* that result from local changes in ocean upwelling and mixing, as well as changes in ocean-atmosphere exchange cannot be box modelled. These changes are common, can be large and, with a well mixed, constant atmospheric $^{14}$C concentration *Atm*, produce large spatio-temporal changes in MRA as further discussed by S&G.

Large changes in MRA with significant spatial variability that are revealed by PT are thus not incompatible with earth system reality as B&H claim based on the modelled results in Fig. 4. Instead, they reveal new detailed information about the interplay of ocean dynamics and the carbon cycle and provide the challenge to attribute those changes to different parts of the system and thus find out more about them.

**2.8**. B&H admonish that $\Delta^{14}$C wiggles may result from production as well as from carbon cycle changes on generally centennial instead of millennial time scales.

This is a continuation of the 2.7 discussion of the effect of $^{14}$C production signals on surface ocean $^{14}$C concentrations. The consideration of ocean out gassing splits the problem in ocean-atmosphere of active source regions and of the passive rest of the world. For a comparison of MRA values produced via PT with model results again the focus has to be on the ocean-atmosphere difference. In the real world $^{14}$C production variations and carbon cycle changes will occur simultaneously to different degrees. Separating their contributions to PT-derived MRA records remains a challenging modelling task for high-resolution OGCM that are able to incorporate local boundary conditions that cannot be considered in box models.

**2.9**. Sarnthein et al should have mentioned *'the two seminal papers'* regarding the use of volcanic ash layers for MRA reconstruction by Bard 1988 and Bard et al., 1994.

SA2020 give an overview of PT, its foundation, its use, and new data and insights it generated. Thus agreement between ages obtained by PT and from volcanic ashes off Chile and New Zealand was discussed. A review of the full field of MRA reconstruction, in which the Bard papers certainly would have featured, was not intended.

**3/ Statistical perspective**

**3.1** Identifying plateaus in the $^{14}$C records of atmospheric Suigetsu and ocean sediment cores is unreliable. This also applies to correlating ocean and atmosphere plateaus to obtain MRA from *pla-Atm* and ventilation ages from *Be-Atm*.

The discussion of *lines 370-385* rephrases the discussion of section 2. As already argued in 2.3 there is a decent correspondence between Suigetsu plateaus and the Bayesian spline pattern in B&H Fig. 1, also seen in the $\Delta^{14}$C domain in Fig. 3a. This answers the first part of the objection above. The question of correlating plateaus has been the main topic of objections in section 2. The problems of correlating single plateaus were extensively discussed by B&H and their arguments are definitely valid. These problems have, however, been considered in PT. They are the reason that only a match of *full suites of plateaus* can be considered, an emphasis overlooked and not mentioned in the introductory *lines 40-41*.

**3.2** This section questions the smoothed character of the Hulu record in *lines 392-422* and repeats the discussion of the real existence of atmospheric $^{14}$C plateaus of section 2.3 in the statistics domain in *lines 423-454*.

B&H misrepresent SA2020, in *lines 392-399*. This may again be based on the misconception of *lines 40-41* that single plateaus are correlated instead of the full records. The Hulu record in Fig. 1 of SA2020, provides a simple graph of Suigetsu data of Bronk Ramsey et al., 2012, and Hulu data of Southon et al., 2012. Not only does Hulu show much smaller age uncertainties than Suigetsu but it also has a smoother structure. Several Suigetsu plateaus can also be seen in Hulu but quite a few cannot be identified in the smoothed record. No theory here, just facts.

B&H argue that the carbonate-based Hulu $^{14}$C record experienced only limited and fairly constant smoothing and, therefore, better indicates atmospheric $^{14}$C concentrations than the terrestrial macrofossil based Suigetsu, used by SA2020. The low 'dead carbon' fraction (DCF) of 480 ± 55 yr, used in IntCal20, is based on a reevaluation of the individual Hulu speleothems against the tree ring $^{14}$C record for the period 10.7 to 13.9 cal kyr BP (Reimer et al., 2020). Its scatter could be the result of random scatter (null hypothesis). Since climate over this Allerød-Younger Dryas-Preboreal period was highly variable, it was concluded that Hulu DCF was insensitive to climate and could be used for the full deglacial-glacial record. The low DCF value was ascribed to the 'sandstone ceiling and open system conditions with the soil above' the section of the cave containing the speleothems (B&H *line 409*).This was recently modified to state that the original overlying limestone has been largely replaced with iron oxides (Reimer et al., 2020).

Including soil organic carbon as a potentially significant source of speleothem carbon complicates life because it will not contribute only 'dead carbon'. DCF is a calculated value indicating how much 'dead' carbon would be needed to obtain the drop in $^{14}$C concentration observed in the speleothem. The speleothem carbon must have been a mixture of recent carbon from the atmosphere and root respiration, and older, not necessarily 'dead' carbon, derived from exchange with iron oxides and decomposing soil organic matter and, maybe, a remnant limestone. The soil organic carbon and iron oxides form a dynamic organic pool in continuous exchange with organic carbon transported by rain water down from the surface. This mixed carbon source makes the isotopic composition of speleothem carbon dependent

on local vegetation, precipitation, and temperature influencing the balance between the different sources. The organic carbon adsorbed on iron oxides and the soil organic carbon, create an integrating memory of preceding $^{13}$C and atmospheric $^{14}$C concentration variability. Indications for significant centennial-to-millennial scale variations in carbon isotopic composition were found by Kong et al., 2005. Their Hulu speleothem $\delta^{13}$C record shows short-term variations of up to 7‰, correlated with climate change as indicated by $\delta^{18}$O, over the period 10-23 cal kyr BP. As both the $^{13}$C and the $^{14}$C isotopic composition of Hulu speleothems will be influenced by mixing processes there is, unfortunately, no easy relationship between the two that can be used for isotopic corrections.

B&H state that the 480-yr DCF and the transport and mixing processes indicate a situation similar to the typical MRA of a low-to-mid latitude surface ocean and that some smoothing of atmospheric $^{14}$C variations, as modelled in 2.7, is to be expected for the Hulu record. As discussed in 2.7, the attenuated recording of an atmospheric $^{14}$C signal in the Hulu speleothem (compare B&H Fig. 4) indicates a change over time in the $^{14}$C concentration difference DCF between speleothem and atmosphere in analogy to MRA = *pla-Atm* .

The second part largely repeats the objection of 2.3. As we discussed there, B&H fig. 1 and 3a actually show a convincing match between the Bayesian-spline generated Suigetsu curves and our selected plateaus. The argument that fluctuations in the section of IntCal20 considered could represent random scatter, i.e. the null hypothesis, can statistically not be rejected with 95 % probability. This means there is indeed no statistical proof that the observed fluctuations in the record are real. Yet, absence of statistical proof is not a statistical proof of absence. To establish the physical reality of $^{14}$C 'events' statistically buried in the atmospheric and oceanic $^{14}$C records we have to resort to comparing many data sets.  The probability that random fluctuations will create patterns that match over a defined interval, decreases with the length of the suite of plateaus and with every new record added to the collection. By now, PT has employed many plateaus of considerable length and their age information and the resulting MRAs and ventilation ages provide new insights.

**3.3** 'How can one be sure of choosing the right plateau in a noisy world'.

This objection correctly lists how difficult it is to extract information from environmental archives. Yet, it is based on the same misconception of trying to match single plateaus, created in *lines 40-41*. The PT protocol has been developed to meet these

challenges, first by matching only suites of plateaus, then by considering all lines of customary stratigraphic information and physical feasibility and by comparing alternative modes of PT and, finally, by evaluating and comparing many records.

**3.4** How does one define a $^{14}$C plateau.

This continues the focus on single plateaus, created by *lines 40-41*, and ignores what has been written about defining suites of plateaus in the age-depth domain and providing them with a rough time estimate using the customary stratigraphic and correlating tools. The implausible numbers presented appear to be just that, misinterpretations. No defined examples are presented.

**3.5** Plateau identification by visual inspection is subjective.

The trained eye is a wonderful instrument, yet, the argument of subjective identification is indeed correct. The use of the first-derivative-kernel- technique, developed by Mudelsee, provided a general mathematical confirmation of the subjective choices. In general, plateaus are first identified visually as $^{14}$C-age/calendar-age line segments with low, near-zero slope; mathematical check and confirmation come later.

**3.6** '*To asses objectively the ability to reliably identify and tune $^{14}$C age plateaus*'
Plateau simulation by random scatter around a straight line. Test of pseudo-atmospheric record

This is a nice and detailed modelling test. The choice of the period 12 – 13.9 cal kyr BP, based on the use of Cariaco as example of the sediment record, is limiting because it covers only 1900 yrs and only three fairly short plateaus (Fig. 2 Sarnthein et al., 2020). A plateau match from 10 -13.9 could have used the same procedure of sampling and adding noise to the IntCal20 tree ring record for a hypothetical Cariaco record and would have been an even better test covering five plateaus instead of three. The first-derivative modelling of B&H seems to correspond reasonably well to what Mudelsee described in the supplement of Sarnthein et al., 2015. There also the source code used has been given. Mudelsee states he used a modest undersmoothing to bring out details but aimed for a minimum plateau length of 300 yr. B&H appear to smooth less and obtain mini plateaus that we rejected as too uncertain on the basis of our sampling resolution. The choice of a gradient of 1, that is one $^{14}$C

year per calendar year instead of 0.5 seems better as it approaches natural [14]C decay and the IntCal curve for this time interval. However, this choice should not change the results; just make them more difficult to read in the figures. B&H Fig. 5a-d demonstrates that noise can indeed change our realization of the underlying record and, consequently, modify a first derivative plot. Yet, the three plateaus, selected in Suigetsu and IntCal13 tree rings of Fig. 2 in Sarnthein et al., 2020, can still be seen as major structures. The test of B&H would have been easier to judge if the three plateaus had been indicated for comparison on the X-axis of Fig. 5.

[Figure]

B&H Fig. 6. In 6b pseudo-marine [14]C records A (purple dots) and B (yellow dots) with first derivatives on depth scale. In 6d first derivatives of sediment record A and B (purple and yellow) and of matching atmospheric record on calendar age scale. In blue on X-axis the plateaus of the underlying tree ring record.

**3.7** Plateau simulation by random scatter around a straight line: Test of pseudo-marine record.

Like 3.6 this exercise may promise to 'realistically scramble' the atmospheric [14]C record in such a way as it might be preserved in a 'nice' sediment core like Cariaco. The crucial record to study is Fig.6b. B&H (*line 631-641)* start a visual description of the pseudo sediment [14]C data and identify observations #1-5 in pseudo 'core A' as potential 'plateau' (*lines 632-635*). They don't identify anything else so, logically, the conclusion of their exercise is that there can be no reliable identification.

If we continue the B&H discussion of Fig. 6b above, then pseudo-core A and B both show nearly constant [14]C ages, a plateau, for observation points #1-6 (=pink and yellow dots) , with B extending back to #7. Then [14]C ages increase till observation #10 in A, #11 in B, and show little change for observations #11 through #16 (A) or #17 (B) which indicates a potential second plateau. Beyond that we have an irregular increase of [14]C ages. If we transfer these choices in Fig. 6b to a time scale in 6c and 6d, it is clear that not only the 12-12.5 cal kyr BP

'YD' plateau could be identified in pseudo-cores A and B but that also the presence of the 'no name' plateau at 12.8-13.1 was seen in both. The third plateau is not reflected. The first derivative has low values where a group of $^{14}$C-age data points has similar values. This works for the 12-12.5 plateau. The 12.8-13.1 plateau shows some scatter in $^{14}$C ages and, accordingly, the first derivative fluctuates but still gives some low values. The fluctuations of the first derivative indicate it reacts to individual data points, which indicates the kernel is undersmoothing. Having the 'estimated gradient' Y-axis in 6c and 6d, like in Fig. 5, and the three plateaus from Sarnthein et al., 2020, indicated on the X-axis in Fig. 6a-c would facilitate the comparison.

As B&H state in the discussion of this experiment, they tried to mimic the plateau tuning of the papers by Sarnthein et al. Yet, the details of the kernel set-up determine the smoothing and their kernel optimization may have been slightly different. If one compares the pseudo planktic record with the pseudo atmosphere in Fig. 6d, the 12-12.5 and 12.8-13.1 plateaus have been identified in this statistical exercise. Accordingly, the underlying structure ('plateaus') of the IntCal tree ring record indeed was detected. The conclusion in B&H *lines 644-650* is thus incorrect, while not supported by their Fig. 6. On the contrary**,** Fig. 6 shows that even in a short segment with only three plateaus two out of three features of the underlying fine structure can be recovered and correlated The modelling exercise also illustrates that, 'even under favourable conditions', it is a difficult task to extract low-amplitude signals from a noisy record. In real records of planktic $^{14}$C that have been subjected to sediment disturbance and potential changes in ocean mixing, the fine structure extracted from each individual record must be checked on its physical feasibility. It is clear that long suites of plateaus, such as used in PT, need to be matched with the atmosphere and with other similar records before they can be accepted. Since the beginning in 2007 many records have been obtained that give a growing confidence that PT provides a means for extracting new information regarding the variations in internal ocean dynamics and ocean-atmosphere exchange over last glacial-deglacial-Holocene times from open ocean sites otherwise lacking clear chronostratigraphic markers for detailed age control.

**3.8** Contention that *'Lake Suigetsu alone provides a more precise reconstruction of atmospheric $^{14}$C levels from 55-13.9 cal kyr BP than IntCal20 synthesis'*.

This section provides a beautifully clear description of what IntCal20 is (and is not) in defence against an attack that never took place. The smoothed character of IntCal beyond tree ring calibration has long been recognized. With the new Bayesian-spline approach IntCal 20 is a lot less smooth than its predecessor IntCal13. Also, the introduction of a modelled time varying reservoir age correction for the different marine records has been a big step forward. Yet, the current IntCal 20 version is the best choice we have for $^{14}$C calibration but it is a work in progress and still has a smoothed character.

As stated in the conclusions of IntCal13: '*We encourage researchers to use the available data to synthesize curves in their own way and potentially contribute new and improved strategies to the construction of the $^{14}$C calibration curve.*' To explore the $^{14}$C fine structure of the ocean sediment record you need the best-available atmospheric fine structure as global benchmark. PT therefore uses the Suigetsu sediment record, based on purely terrestrial macrofossils, as atmospheric benchmark instead of the allegedly more secure but also smoothed IntCal20, resulting from a statistically sound synthesis of all available data sets. The objection that Suigetsu is only one record and maybe noisy is correct but fades against the fact that Suigetsu directly represents the atmosphere. PT follows the encouragement of IntCal13, quoted above, and uses the Suigetsu record as a research tool to extract new information from the marine records that can, we hope, contribute to the next, improved version of IntCalXX.

**References**

Bard, E., Raisbeck, G., Yiou, F., Jouzel, J. Solar modulation of cosmogenic nuclide production over the last millennium: comparison between $^{14}$C and $^{10}$Be records. *Earth and Planetary Science Letters* 150, 453-462. doi: 10.1016/S0012-821X(97)00082-4, 1997

Bronk Ramsey. C., Heaton, T.J., Schlolaut, G., Staff, R.A., Bryant, C.L., Brauer, A., Lamb, H.F., Marshall, M.H., Nakagawa, T. Reanalysis of the atmospheric radiocarbon calibration record from Lake Suigetsu, Japan. *Radiocarbon* 62, 989-999, doi: 10.1017/RDC.2020.18, 2020.

Bronk Ramsey, C., Staff, R..A., Bryant, C.L., Brock, F., Kitagawa, H., van der Plicht, J., Schlolaut, G., Marshall, M.H., Brauer, A., Lamb, H.F., Payne, R.L., Tarasov, P.E., Haraguchi, T., Gotanda,

K., Yonenobu, H., Yokoyama, Y., Tada, R., Nakagawa, T., A complete terrestrial radiocarbon record for 11.2 to 52.8 kyr BP. *Science* 338: 370-374, 2012.

Kong, X., Wang, Y., Wu, J., Cheng, H., Edwards, R.L., and Wang, X., Complicated responses of stalagmite $\delta^{13}C$ to climate change during the last glaciation from Hulu Cave, Nanjing, China, *Science in China Ser. D Earth Sciences*, 48, (12), 2174-2181, 2005.

Lohman, G., Butzin, M., Eissner, N., Shi, X., Stepanek, C., Abrupt climate and weather changes across time scales. *Paleoceanography and Paleoclimatology*, submitted 2021

Miyake, F., Nagaya, K., Masuda, K., Nakamura, T., A signature of cosmic-ray increase in AD 774-775 from tree rings in Japan. *Nature* 486, 240-242, doi: 10.1038/nature11123, 2012

Reimer, P.J., Austin, W.E.N., Bard, E., Bayliss, A., Blackwell, P.G., Bronk Ramsey, C., Butzin, M., Cheng, H., Edwards, R.L., Friedrich, M., Grootes, P.M., Guilderson, T.P., Hajdas, I., Heaton, T.J., Hogg., A.G., Hughen, K.A., Kromer, B., Manning, S.W., Muscheler, R., Palmer, J.G., Pearson, C., van der Plicht, H., Reimer, R.W., Richards, D., Scott, E.M., Southon, J.R., Turney, C.S.M., Wacker, L., Adolphi, F., Büntgen, U., Capano, M., Fahrni, S., Fogtmann-Schulz, A., Friedrich, R., Kudsk, S., Miyake, F., Olsen, J., Reinig, F., Minuro Sakamoto, M., Sookdeo, A., Talamo, S., The IntCal20 Northern Hemisphere radiocarbon calibration curve (0-55 kcalBP). *Radiocarbon* 62, 724-757, doi: 10.1017/RDC.2020.41. 2020

Sarnthein, M., Balmer, S., Grootes, P.M., and Mudelsee, M., Planktic and benthic [14]C reservoir ages for three ocean basins, calibrated by a suite of [14]C plateaus in the glacial-to-deglacial Suigetsu atmospheric [14]C record, *Radiocarbon*, 57, 129-151, doi: 10.2458/azu_rc.57.17916, 2015

Sarnthein, M., Grootes, P.M., Kennett, J.P., Nadeau, M.J., [14]C Reservoir ages show deglacial changes in ocean currents. In: Ocean Circulation: Mechanisms and Impacts, *Geophysical Monograph Series* 173, edited by: Schmittner, A., Chiang, J., and Hemming, S., American Geophysical Union, Washington, DC. 175-179. doi: 10.1029/173GMOX, 2007.

Siegenthaler, U., Heimann, M., and Oeschger, H., $^{14}$C variations caused by changes in the global carbon cycle. *Radiocarbon* 22, 177-191, 1980.

Southon, J., Noronha, A.L., Cheng, H., Edwards, R.L., and Wang, Y., A high-resolution record of atmospheric $^{14}$C based on Hulu Cave speleothem H82, *Quaternary Science Reviews*, 33, 32-41, 2012.

Umling, N.E. and Thunell, R.C., Synchronous deglacial thermcline and deep-water ventilation in the eastern equatorial Pacific. *Nat. Commun*. 8. 14203 doi: 10.1038/ncomms 14203 , 2017

---

## Community Comment (CC4)

**Comparison of visually and mathematically constructed $^{14}$C plateaus, based on Suigetsu data and INTCAL20 curve.**

Bernhard Weninger, Universität zu Köln, Institute for Prehistoric Archaeology,
Weyertal 125, D-50923 Köln, Germany

This communication is focused on the presently open question whether – or not – the sequence of $^{14}$C-plateaus defined by Sarnthein et al., 2020, for the Lake Suigetsu data record can likewise be identified (or not) in the INTCAL20 curve. Despite the extensive critique of the Plateau Tuning (PT) method by Bard and Heaton (2021; B&H) and detailed responses by Sarnthein and Grootes (2021; S&G), Grootes and Sarnthein (2021; G&S), it appears to me that a further open question exists, namely the (seemingly) reasonable claim by Paula Reimer (2021), that the INTCAL20 curve is '*simply too noisy*', whereas Suigetsu is '*without sufficient resolution*' to allow for a '*robust*' identification of plateaus - in either record. If confirmed, this would not necessarily inhibit an application of INTCAL20 for PT, for reasons put forward by S&G and G&S. The 'noise'-argument, however, still presents a number of caveats that persist. In particular, since INTCAL20 incorporates Suigetsu with carbonate based marine and cave records, it appears more 'secure', less 'noisy', but 'smoothed'. As apparently accepted by all participants of this discussion, in consequence the application of PT presently seems to be restricted to the use of a 'plateau master curve' solely based on the Suigetsu record (Sarnthein and Grootes, 2021).

Notwithstanding the validity of many arguments that underly the views of S&G and B&H, by chance the very same question – "*How to define a set of plateaus for a smoothed calibration curve*?" – has recently been addressed in detail both for the Holocene by myself (Weninger, 2020) and the Last Glacial in a joint paper (Weninger and Edinborough, 2020; W&E). In brief, we developed a mathematical method for an automated derivation of the requested 'calibration-curve' plateaus. The method is based (in a sense) on amplifying the otherwise difficult to recognise shape-properties of INTCAL20-curve, whereby the curve-shape analysis is replaced by amplitude-analysis of an equivalent summed probability distribution (SPD) of calibrated $^{14}$C-ages. In technical terms, based on INTCAL20 for the period 9-25 ka cal BP, a dataset with 9500 evenly distributed Gaussian $^{14}$C-ages is derived, each assigned a standard deviation σ = 50 BP. The Gaussian $^{14}$C-ages are summed on the $^{14}$C-scale, and the resulting $^{14}$C-histogram is back-calibrated from the $^{14}$C-domain to the calendar time scale. The resulting SPD has widely varying amplitude. It shows a characteristic sequence of peaks and troughs, the dating and amplitude of which are a strong function both of the curve slope and of its 'wiggles'. The peak values are scaled to a maximum 'probability' p(rel) = 100 %. This maximum height is reached for plateaus 'YD' and '1' labelled by Sarnthein et al. (2020). Finally, the requested plateau boundaries are derived by a crucial gauge method. In mathematical terms, both the SPD-construction and the gauge method are related to a description of $^{14}$C-calibration as Fourier Transform. The approach leads to corresponding concepts of $^{14}$C-dating probability, that are derived from Quantum Theory. To avoid misunderstandings, the SPD method must not be regarded as a Fourier Transformation. (The method does *no*t redraw the calibration curve as sum of sines and cosines, but as sum of transformed Gaussians. An optical analogy is laser amplification). In a nut-shell, SPD-shape and assigned rectangular plateaus represent the same entity from the perspective of Quantum theory. Suffice to say, the approach of using a SPD record as proxy for the INTCAL20 curve shape is a non-Bayesian probability definition.

For the present discussion it may be of particular interest that a simple close-up view of the shape of the INTCAL20 calibration curve *per se* does not allow any recognition of the inborn plateau structure. The plateaus, however, exist indeed, are strong, and are easily *made visible* by means of the SPD-peaks, as detailed in W&E.

[Figure]

**Fig. 1. Shape of INTCAL20 record compared to plateau boundary ages based on Suigetsu data.**

SUIGETSU: Plateau names and blue-shaded areas, according to Sarnthein and Grootes (2020: Tab.1)
Method: visual inspection of Suigetsu data.
INTCAL20: Summed Calibrated Probability Distribution (SPD) of 9500 model Gaussian [14]C-ages (identical σ=50 BP).
Method: In descriptive terms, the input uniform sample sequence is folded by INTCAL20 to produce a characteristic sequence of peaks and troughs. The SPD-peaks represent the flat regions of the calibration curve.The troughs represent the steep regions. In mathematical terms, the SPD is a Fourier Transform of INTCAL20 (W&E).

As illustrated in Fig. 1, the SPD peaks clearly serve as proxy for plateaus potentially existing in the INTCAL20 record. Of course the plateaus could also be - *qua method* - wrongly established in both records. But the method is promising. Many SPD-peaks are entirely consistent with the independently established plateaus of Sarnthein et al., (2020) (blue shades). With increasing cal. age, these are the peaks/plateaus named Preboreal, TopYD, YD, nn, 1a, 1, 7, and 8 (lump of three peaks). Other plateaus (6b, 6a, 5b, 5a, 4, 3, 2a and 2b) certainly only show (what archaeologists might term) 'restricted similarities', although with interesting deviations. Beyond the question at which ages the two methods lead to '*well-fitting*' or '*not well-fitting*' plateaus, it is of further interest that plateaus may exist at ages previously unrecognised (e.g. ~17.2 ka, ~19.7 ka cal BP). In different words, we now have the chance not only to differentiate between 'plateaus' and 'noise', but may further refine the existing plateau definitions. We can furthermore learn how to quantify (from the new perspective of SPD) some possibly underlying physical and oceanographic variables that have controlled the origin of plateaus, perhaps in combination with further studies towards the newly developed concepts of probability theory that are derived from Quantum physics.

Conclusion
Given the overall good agreement between the visually-defined Suigetsu plateau-sequence and the mathematically-derived INTCAL20-based plateau-sequence (Fig. 1), the critique on the PT-method by B&H may appear unnecessarily restrictive, especially when seen in relation to actually promising further PT-research, now underway.

References
    Bard, E. and Heaton, T. J.: On the tuning of plateaus in atmospheric and oceanic [14]C records to derive calendar chronologies of deep-sea cores and records of [14]C marine reservoir age changes, Clim. Past Discuss. [preprint], https://doi.org/10.5194/cp-2020-164, in review, 2021.
    Grootes, P.M. and Sarnthein, M., 2021. Comments on Eduard Bard and Timothy J. Heaton (B&H). Climate of the Past. Discussions, submitted 17 Feb 2021.
    Reimer, P. J., Austin, W. E. N., Bard, E., et al.: The INTCAL20 northern hemisphere radiocarbon calibration curve (0–55 cal kBP), Radiocarbon, 62, 725–757, https://doi.org/10.1017/RDC.2020.41, 2020.

Michael Sarnthein, Kevin Küssner, Pieter M. Grootes, Blanca Ausin, Timothy Eglinton, Juan Muglia, Raimund Muscheler, and Gordon Schlolaut: Plateaus and jumps in the atmospheric radiocarbon record – potential origin and value as global age markers for glacial-to-deglacial paleoceanography, a synthesis. Clim. Past, 16, 2547–2571, https://doi.org/10.5194/cp-16-2547-2020, 2020a.

Michael Sarnthein and Pieter M. Grootes., 2021. Response to the Preprint of E.Bard and T.J.Heaton (B&H), Climate of the Past. Discussions, submitted 16 Feb 2021

Weninger, Bernhard. 2020. Barcode seriation and concepts of Gauge Theory. The [14]C-Chronology of Starčevo, LBK, and early Vinča. Quaternary International Volumes 560–561, 20 September 2020, Pages 20-37. https://doi.org/10.1016/j.quaint.2020.04.031

Weninger, Bernhard. and Edinborough, Kevan. Bayesian [14]C-rationality, Heisenberg Uncertainty, and Fourier Transform. Documenta Praehistorica 47:536-559, 2020.

---

## Author Response (AR1)

Chaire de l'évolution du climat et de l'océan

Aix-en-Provence, June 15th, 2021

Dear Editor, Dear André,

Please find enclosed our revised version of the paper *"On the tuning of plateaus in atmospheric and oceanic $^{14}C$ records to derive calendar chronologies of deep-sea cores and records of $^{14}C$ marine reservoir age changes".*

In the previous phase of the review process, we provided detailed answers to the points raised in the RCs and CCs. When revising the manuscript, we took into account your recommendations:

The request by Paula Reimer (Referee #1) led us to perform and provide further statistical calculations to demonstrate the effects of the threshold value of the gradient ($^{14}C$ yr/cal yr), and of the kernel bandwidth used to detect and define the age plateaus. This led to the inclusion of an additional Figure 6 and additional text (lines 613-648, 697-746, 803-809, 819-836).

We followed the suggestion by the anonymous referee (Referee # 2) and added a final section entitled "conclusion and outlook" (lines 886-959).

The last paragraphs of this final section (lines 924-953) provide conditions and tests that are necessary to allow wiggle matching of $^{14}C$ plateaus and other $^{14}C$ structures in the data records. The last paragraph (lines 954-959) mentions the usefulness of new ocean modeling combined with wiggle matching.

In addition, additional sentences were included throughout the paper to discuss and clarify relevant points raised in the CCs.

In particular, the technique proposed by Weninger & Edinborough (2020) has been mentioned and briefly discussed (lines 597-603).

For the additions listed above, it was necessary to cite 20 new papers, which were added in the list of references, notably the papers by Arz et al. (1999), Ausin et al. (2021), Lamy et al. (2015), Siani et al. (2013) and Weninger & Edinborough (2020).

All these important changes are highlighted in yellow on the marked copy including brief comments about the link with specific RCs or CCs.

We hope that the changes we have carried out will qualify this manuscript for publication in *Climate of the Past*, and we thank you very much for your attention.

Edouard Bard    & Tim Heaton